# ON THE EXPRESSIVITY OF OBJECTIVE-SPECIFICATION FORMALISMS IN REINFORCEMENT LEARNING

**Rohan Subramani**[a,b,*]**, Marcus Williams**[a,*]**, Max Heitmann**[a,c,*]**, Halfdan Holm**[a]**, Charlie Griffin**[a,c]**, Joar Skalse**[a,c]

[a]AI Safety Hub, [b]Columbia University, [c]University of Oxford
*Equal contribution. Correspondence to Rohan Subramani at rs4126@columbia.edu.

## ABSTRACT

Most algorithms in reinforcement learning (RL) require that the objective is formalised with a Markovian reward function. However, it is well-known that certain tasks cannot be expressed by means of an objective in the Markov rewards formalism, motivating the study of alternative objective-specification formalisms in RL such as Linear Temporal Logic and Multi-Objective Reinforcement Learning. To date, there has not yet been any thorough analysis of how these formalisms relate to each other in terms of their expressivity. We fill this gap in the existing literature by providing a comprehensive comparison of 17 salient objective-specification formalisms. We place these formalisms in a preorder based on their expressive power, and present this preorder as a Hasse diagram. We find a variety of limitations for the different formalisms, and argue that no formalism is both dominantly expressive and straightforward to optimise with current techniques. For example, we prove that each of Regularised RL, (Outer) Nonlinear Markov Rewards, Reward Machines, Linear Temporal Logic, and Limit Average Rewards can express a task that the others cannot. The significance of our results is twofold. First, we identify important expressivity limitations to consider when specifying objectives for policy optimization. Second, our results highlight the need for future research which adapts reward learning to work with a greater variety of formalisms, since many existing reward learning methods assume that the desired objective takes a Markovian form. Our work contributes towards a more cohesive understanding of the costs and benefits of different RL objective-specification formalisms.

## 1 INTRODUCTION

There are many ways of specifying objectives in Reinforcement Learning (RL). The most common method is to maximise the expected time-discounted sum of scalar Markovian rewards.[1] While this method has achieved wide-ranging success, recent work has identified practical objectives that cannot be specified using standard Markov rewards (Skalse & Abate (2023); Abel et al. (2022); Bowling et al. (2022)). Numerous other formalisms have been proposed and utilised for specifying objectives in practice, including Multi-Objective RL (Hayes et al. (2022); Roijers et al. (2013); Coello Coello et al. (2002)), Maximum Entropy RL (Hazan et al. (2019); Mutti et al. (2023); Ziebart et al. (2008)), Linear Temporal Logic (Littman et al. (2017); Lahijanian et al. (2011); Ding et al. (2011)), and Reward Machines (Icarte et al. (2018); Toro Icarte et al. (2022); Camacho et al. (2019)). Additionally, there are various abstract formalisms (such as arbitrary functions from trajectories to the real numbers), which are generally intractable to optimise but which capture intuitive classes of objectives.

In this paper, we comprehensively compare the expressivities of 17 formalisms, which are listed and defined in Table 1. We say that a formalism A can express another formalism B if in all environments, A can express all objectives that B can express. We formalise this notion in Section 2. Our main results are summarised in Figure 1, a Hasse diagram displaying the relative expressivities of all the formalisms we consider. We find that in many cases, there is no simple answer to the question of which of two formalisms is preferable to use for policy optimisation in practice because each can

---

[1]A reward function is Markovian if it depends only on the most recent transition.

express objectives that the other cannot. While there are formalisms at the top of our diagram that can express every formalism below them, we suspect that none of these are tractable to optimise in general. Therefore, we advise RL practitioners to familiarise themselves with a variety of formalisms and think carefully about which to select depending on the desired use case. Our analysis describes some of the strengths and limitations of each formalism, which can be used to inform this selection.

Our results are relevant not only for policy optimisation but also for reward learning. Reward learning is often used when it is difficult to hardcode a reward function for a task but easy to evaluate attempts at that task. For instance, (Christiano et al. (2017)) train a reward model which incentivises an RL agent to do backflips in a simulated environment. Notably, the structure of reward models in many prior works have implicitly assumed that the desired task is expressible with Markov rewards, since reward models output real numbers given a transition as input (Christiano et al., 2017; Hadfield-Menell et al., 2016; Skalse et al., 2023). Our findings highlight the limitations of Markov rewards and the importance of advancing reward learning methods for the other formalisms we discuss.

In Section 2, we briefly introduce core concepts from RL relevant to our paper, make the nature of our formalism comparisons precise, and define the objective-specification formalisms that we are comparing in this work. We provide our primary results in Section 3. These are summarised in Figure 1, a Hasse diagram that depicts all the expressivity relationships among the formalisms. We also present many of the theorems and propositions on which the diagram is based; the remaining statements and all proofs are found in Appendix B. In Section 4, we discuss the implications of our findings in the context of prior research, along with limitations and directions for future work.

## 2 PRELIMINARIES

### 2.1 BASIC DEFINITIONS

Many of the concepts in this paper are derived from the study of Markov Decision Processes (MDPs) (Sutton & Barto (2018)). An MDP is a 6-tuple $(\mathcal{S}, \mathcal{A}, \mathcal{T}, \mathcal{I}, \mathcal{R}, \gamma)$ where $\mathcal{S}$ is a set of states, $\mathcal{A}$ is a set of actions, $\mathcal{T} : \mathcal{S} \times \mathcal{A} \to \Delta\mathcal{S}$ is a transition function and $\mathcal{I} \in \Delta\mathcal{S}$ is the initial distribution over states. In this work, we assume $\mathcal{S}$ and $\mathcal{A}$ are finite and only consider stochastic stationary policies $\pi : \mathcal{S} \to \Delta\mathcal{A}$, solutions to the MDP which stochastically output an action in any state.[2] A (rewardless) trajectory is an infinite sequence $\xi = (s_0, a_0, s_1, a_1, \cdots)$ such that $a_i \in \mathcal{A}$, $s_i \in \mathcal{S}$ for all $i$, where $s_0 \sim \mathcal{I}$, $a_0 \sim \pi(s_0)$, $s_1 \sim \mathcal{T}(s_0, a_0)$ and so on; we denote the set of (infinite) trajectories as $\Xi = \mathcal{S} \times (\mathcal{A} \times \mathcal{S})^\omega$. The reward function $\mathcal{R} : \mathcal{S} \times \mathcal{A} \times \mathcal{S} \to \mathbb{R}$ gives a reward at each time step $r_t = \mathcal{R}(s_t, a_t, s_{t+1})$, and $\gamma \in [0, 1)$ gives a discount factor.

The return of a trajectory $\xi$ for $\mathcal{R}$ and $\gamma$ is $G_{R,\gamma}(\xi) = \sum_{t=0}^{\infty} \gamma^t \mathcal{R}(s_t, a_t, s_{t+1})$ and the expected return of a policy is taken over trajectories sampled using the environment and the policy: $J^E_{\mathcal{O}_{MR}}(\pi) = \mathbb{E}_{\xi \sim \pi, \mathcal{T}, \mathcal{I}}[G(\xi)]$.

Since we consider various ways to define an objective, we separate our decision process into an *environment* and an *objective specification*.

**Definition 2.1** $(E, \Pi^E, Envs)$**.** We refer to the 4-tuple $(\mathcal{S}, \mathcal{A}, \mathcal{T}, \mathcal{I})$ as the environment and denote it $E$. We denote the set of finite environments as $Envs$.[3] A given environment determines the stationary-policy space $\Pi^E = \{\pi \mid \pi : S \to \Delta(A)\}$.

Using a Markovian Reward $\mathcal{R}$ and geometric discounting factor $\gamma$ is one way to specify an objective. However, in this work we want to compare and contrast ways to specify objectives. To do this, we introduce the notion of an *objective specification*.

**Definition 2.2** $(\mathcal{O}, \succeq^E_{\mathcal{O}})$**.** Given an environment $E$, an *objective specification* is a tuple $\mathcal{O}$ that allows us to rank policies in $\Pi^E$. Formally, $\mathcal{O}$ defines $\succeq^E_{\mathcal{O}}$ a total preorder over $\Pi^E$. Total preorders are transitive and strongly connected.

---

[2] We assume that the state and action spaces are finite and only consider stationary policies in order to make our findings more relevant to common RL algorithms. For example, Q-learning only has a convergence guarantee for finite state and action spaces and only considers stationary policies.

[3] We can formally define $Envs$, the set of environments as $Envs := \{(\mathcal{S}, \mathcal{A}, \mathcal{T}, \mathcal{I}) \mid S, A \text{ are any two finite sets}, T : \mathcal{S} \times \mathcal{A} \to \Delta\mathcal{S}, \mathcal{I} \in \Delta(\mathcal{S})\}$.

**Definition 2.3.** (MR, $\mathcal{O}_{MR}$, $\succeq_{\mathcal{O}}^E$) For environment $E$, an objective specification in the Markovian reward formalism is a pair $\mathcal{O}_{MR} = (\mathcal{R}, \gamma)$ with $\mathcal{R} : \mathcal{S} \times \mathcal{A} \times \mathcal{S} \to \mathbb{R}$ and $\gamma \in [0, 1)$. The ordering over $\Pi^E$ induced by $\mathcal{O}_{MR}$ is given by: $\pi_1 \succeq_{\mathcal{O}_{MR}}^E \pi_2 \iff J_{(\mathcal{R},\gamma)}(\pi_1) \geq J_{(\mathcal{R},\gamma)}(\pi_2)$.

To compare different objective-specification formalisms, it will be important to consider the set of policy orderings possible in a given formalism. Since the set of valid objective specifications and orderings depends on the environment, we define a function $Ord$.

**Definition 2.4** (Objective specification formalism $X$, $Ord_X$ and $Ord_{MR}$)**.** An objective-specification formalism $X$ is valid if it defines a function from environments to orderings over policies in that environment. Given an objective-specification formalism $X$, we denote the set of possible orderings for a given environment as $Ord_X(E)$ where $Ord_X(E) \subseteq \mathcal{P}(\Pi^E \times \Pi^E)$. For example, $Ord_{MR}(E) = \{\succeq_{\mathcal{O}_{MR}}^E \mid R : \mathcal{S} \times \mathcal{A} \times S \to \mathbb{R}$ and $\gamma \in [0, 1)\}$.

Finally, we define a partial order expressivity relation over objective specification formalisms.

**Definition 2.5** ( $\succeq_{EXPR}$)**.** . We define an order over objective specification formalisms $\succeq_{EXPR}$. For any two objective specification formalisms $X$ and $Y$, $X \succeq_{EXPR} Y$ if and only if $X$ can express all policy orderings that $Y$ can express, in all environments. Formally:

$$X \succeq_{EXPR} Y \iff \forall E \in Envs, \, Ord_X(E) \supseteq Ord_Y(E)$$

Note that $\succeq_{EXPR}$ is reflexive and transitive.

Like Abel et al. (2022) and Skalse & Abate (2023), we focus on the ability of formalisms to express policy orderings, as opposed to alternatives such as their ability to induce a desired optimal policy. One key reason we define expressivity in this way is that it is sometimes infeasible to train an agent to find an optimal policy in practice; in such cases, an objective specification is more likely to provide an effective training signal if it expresses a desired policy ordering than if it merely induces a desired optimal policy. We motivate the choice to focus on policy orderings further in Appendix A.1.[4]

## 2.2 FORMALISM DEFINITIONS

In Table 1, we present the definitions of all the formalisms we consider in this work. These definitions are supplemented with a few additional details in Section 2.2.1. Past work related to many of these formalisms is discussed in Section 4. For formalisms which have ambiguous or varying definitions in past literature, we attempt to select the definitions that best allow for meaningful expressivity comparisons. In some cases, the appeal of these formalisms also becomes clearer in the context of our results and proofs.

In Appendix A.2, we argue that objective specifications in all of the formalisms in Table 1 induce total preorders on the set of policies.

### 2.2.1 ADDITIONAL MACHINERY

Here, we provide a few additional definitions that the formalisms in Table 1 depend upon. The first is for Linear Temporal Logic (LTL) formulas, which are required for specifying objectives with the LTL formalism.

**Definition 2.6.** *Linear Temporal Logic Formula.* An LTL formula is built up from a set of atomic propositions; the logic connectives: negation ($\neg$), disjunction ($\vee$), conjunction ($\wedge$) and material implication ($\to$); and the temporal modal operators: next ($\bigcirc$), always ($\square$), eventually ($\diamond$) and until ($\mathcal{U}$). We take the set of atomic propositions to be $\mathcal{S} \times \mathcal{A} \times \mathcal{S}$, the set of transitions. An LTL formula $\varphi$ is either true or false for a trajectory $\xi \in \Xi = \mathcal{S} \times (\mathcal{A} \times \mathcal{S})^\omega$; we say $\varphi(\xi) = 1$ if the formula evaluates to true in $\xi$ and $\varphi(\xi) = 0$ if the formula evaluates to false (Littman et al. (2017); Manna & Pnueli (1992); Baier & Katoen (2008)).

---

[4]The definitions in Section 2.1 use detailed indexing to clearly convey that some symbols correspond to specific environments and / or objective specifications. In other sections (particularly Appendix B), we often drop these indices when the meaning of the symbols is evident.

| Formalism Name | Objective Tuple | Types and definitions | Policy ordering method. For scalar J functions, $\pi_1 \succeq^E_O \pi_2 \iff J(\pi_1) \geq J(\pi_2)$ |
|---|---|---|---|
| Markov Rewards (MR) | $(\mathcal{R}, \gamma)$ | $\mathcal{R} : \mathcal{S} \times \mathcal{A} \times \mathcal{S} \to \mathbb{R}$ 
 $\gamma \in [0, 1)$ | $J(\pi) := \mathbb{E}_\xi \left[ \sum\limits_{t=0}^\infty \gamma^t \mathcal{R}(s_t, a_t, s_{t+1}) \right]$ |
| Limit Average Reward (LAR) | $(\mathcal{R})$ | $\mathcal{R} : \mathcal{S} \times \mathcal{A} \times \mathcal{S} \to \mathbb{R}$ | $J(\pi) := \lim\limits_{N \to \infty} \left[ \frac{1}{N} \mathbb{E}_\xi \left[ \sum\limits_{t=0}^{N-1} \mathcal{R}(s_t, a_t, s_{t+1}) \right] \right]$ |
| Linear Temporal Logic (LTL) | $(\varphi)$ | $\varphi : \Xi \to \{0, 1\}$ is an LTL formula 
 (See Definition 2.6) | $J(\pi) := \mathbb{E}_\xi [\varphi(\xi)]$ |
| Reward Machines (RM) | $(U, u_0, \delta_U, \delta_\mathcal{R}, \gamma)$ | $U$ is a finite set of machine states 
 $u_0 \in U$ is the start state 
 $\delta_U : U \times \mathcal{S} \times \mathcal{A} \times \mathcal{S} \to U$ is a transition function for machine states 
 $\delta_\mathcal{R} : U \times U \to [\mathcal{S} \times \mathcal{A} \times \mathcal{S} \to \mathbb{R}]$ is a transition function that outputs a reward function given a machine transition 
 $\gamma \in [0, 1)$ | $J(\pi) := \mathbb{E}_\xi \left[ \sum\limits_{t=0}^\infty \gamma^t \mathcal{R}_t(s_t, a_t, s_{t+1}) \right]$, where 
 $\mathcal{R}_t = \delta_\mathcal{R}(u_t, u_{t+1})$ |
| Inner Nonlinear Markov Rewards (INMR) | $(\mathcal{R}, f, \gamma)$ | $\mathcal{R} : \mathcal{S} \times \mathcal{A} \times \mathcal{S} \to \mathbb{R}$ 
 $f : \mathbb{R} \to \mathbb{R}$ 
 $\gamma \in [0, 1)$ | $J(\pi) := \mathbb{E}_\xi \left[ f \left( \sum\limits_{t=0}^\infty \gamma^t \mathcal{R}(s_t, a_t, s_{t+1}) \right) \right]$ |
| Inner Multi-Objective RL (IMORL) | $(k, \mathcal{R}, f, \gamma)$ | $k \in \mathbb{N}$ 
 $\mathcal{R} : \mathcal{S} \times \mathcal{A} \times \mathcal{S} \to \mathbb{R}^k$ is a k-dimensional reward function 
 $f : \mathbb{R}^k \to \mathbb{R}$ 
 $\gamma \in [0, 1)$ | $J(\pi) := \mathbb{E}_\xi [f(G_1(\xi), ..., G_k(\xi))]$, where 
 $G_i(\xi) := \sum\limits_{t=0}^\infty \gamma^t \mathcal{R}_i(s_t, a_t, s_{t+1})$ and 
 $\mathcal{R}_i$ is the ith dimension of $\mathcal{R}$ |
| Functions from Trajectories to Reals (FTR) | $(f)$ | $f : \Xi \to \mathbb{R}$ | $J(\pi) := \mathbb{E}_\xi [f(\xi)]$ |
| Regularised RL (RRL) | $(\mathcal{R}, \alpha, F, \gamma)$ | $\mathcal{R} : \mathcal{S} \times \mathcal{A} \times \mathcal{S} \to \mathbb{R}$ 
 $\alpha \in \mathbb{R}$ 
 $F : \Delta(A) \to \mathbb{R}$ 
 $\gamma \in [0, 1)$ | $J(\pi) := \mathbb{E}_\xi \left[ \sum\limits_{t=0}^\infty \gamma^t \left( \mathcal{R}(s_t, a_t, s_{t+1}) - \alpha F[\pi(s_t)] \right) \right]$ |
| Outer Nonlinear Markov Rewards (ONMR) | $(\mathcal{R}, f, \gamma)$ | $\mathcal{R} : \mathcal{S} \times \mathcal{A} \times \mathcal{S} \to \mathbb{R}$ 
 $f : \mathbb{R} \to \mathbb{R}$ 
 $\gamma \in [0, 1)$ | $J(\pi) := f \left( \mathbb{E}_\xi \left[ \sum\limits_{t=0}^\infty \gamma^t \mathcal{R}(s_t, a_t, s_{t+1}) \right] \right)$ |
| Outer Multi-Objective RL (OMORL) | $(k, \mathcal{R}, f, \gamma)$ | $k \in \mathbb{N}$ 
 $\mathcal{R} : \mathcal{S} \times \mathcal{A} \times \mathcal{S} \to \mathbb{R}^k$ is a k-dimensional reward function 
 $f : \mathbb{R}^k \to \mathbb{R}, \gamma \in [0, 1)$ | $J(\pi) := f(J_1(\pi), ..., J_k(\pi))$, where 
 $J_i(\pi) := \mathbb{E}_\xi \left[ \sum\limits_{t=0}^\infty \gamma^t \mathcal{R}_i(s_t, a_t, s_{t+1}) \right]$ |
| Functions from Occupancy Measures to Reals (FOMR) | $(f, \gamma)$ | $f : \vec{m}(\Pi) \to \mathbb{R}, \gamma \in [0, 1)$ | $J(\pi) := f(\vec{m}(\pi))$ 
 (See Definition 2.7) |
| Functions from Trajectory Lotteries to Reals (FTLR) | $(f)$ | $f : L_{\Pi_{S,A}} \to \mathbb{R}$ 
 (See Definition 2.8) | $J(\pi) := f(L_\pi)$ |
| Functions from Policies to Reals (FPR) | $(J)$ | $J : \Pi_{S,A} \to \mathbb{R}$ | $J(\pi)$ (arbitrary) |
| Occupancy Measure Orderings (OMO) | $(\gamma, \succeq_m)$ | $\succeq_m$ is a total preorder on 
 $\vec{m}(\Pi) = \{\vec{m}(\pi) : \pi \in \Pi_{S,A}\}$ 
 $\gamma \in [0, 1)$ 
 (See Definition 2.7) | $\pi_1 \succeq \pi_2 \iff \vec{m}(\pi_1) \succeq_m \vec{m}(\pi_2)$ |
| Trajectory Lottery Orderings (TLO) | $(\succeq_L)$ | $\succeq_L$ is a total preorder on $L_{\Pi_{S,A}}$ 
 (See Definition 2.8) | $\pi_1 \succeq \pi_2 \iff L_{\pi_1} \succeq_L L_{\pi_2}$ |
| Generalised Outer Multi-Objective RL (GOMORL) | $(k, \mathcal{R}, \gamma, \succeq_J)$ | $k \in \mathbb{N}$ 
 $\mathcal{R} : \mathcal{S} \times \mathcal{A} \times \mathcal{S} \to \mathbb{R}^k$ is a k-dimensional reward function 
 $\gamma \in [0, 1)$ 
 $\succeq_J$ is a total preorder on $\mathbb{R}^k$ | $\vec{J}(\pi) := \langle J_1(\pi), ..., J_k(\pi) \rangle$, where 
 $J_i(\pi) := \mathbb{E}_\xi \left[ \sum\limits_{t=0}^\infty \gamma^t \mathcal{R}_i(s_t, a_t, s_{t+1}) \right]$ 
 $\pi_1 \succeq \pi_2 \iff \vec{J}(\pi_1) \succeq_J \vec{J}(\pi_2)$ |
| Policy Orderings (PO) | $(\succeq)$ | $\succeq$ is a total preorder on $\Pi_{S,A}$ | $\pi_1 \succeq \pi_2$ (directly specified) |

Table 1: This table provides the objective tuple along with definitions for all 17 formalisms considered in this paper. It also explicitly states the method by which each formalism orders policies. All expectations are taken over trajectories in the environment sampled using the policy.

Next, we define occupancy measures, which are a central object for the formalisms of Functions from Occupancy Measures to Reals and Occupancy Measure Orderings.

**Definition 2.7.** *Occupancy Measures.* Given a policy $\pi$, an environment $E = (\mathcal{S}, \mathcal{A}, \mathcal{T}, \mathcal{I})$, and a discount factor $\gamma$, the occupancy measure $\vec{m}(\pi)$ is a vector in $\mathbb{R}^{|S||A||S|}$, where:

$$\vec{m}(\pi)[s, a, s'] := \sum_{t=0}^\infty \gamma^t \mathbb{P}_{\xi \sim \pi, \mathcal{T}, \mathcal{I}}[s_t = s, a_t = a, s_{t+1} = s']$$

As the name suggests, $\vec{m}(\pi)$ measures the extent to which a policy "occupies" each transition, in expectation, across the entirety of a trajectory.

For the formalisms of Functions from Trajectory Lotteries to Reals and Trajectory Lottery Orderings, we must define trajectory lotteries. To avoid concerns related to non-measurable sets that arise with generic distributions over the set of all trajectories, we define trajectory lotteries using an infinite sequence of lotteries over finite trajectory segments.

**Definition 2.8.** *Trajectory Lotteries.* Let $\Xi_k$ be the set of all initial trajectory segments of length $2k + 1$. We write $[\xi]_k$ for the first $2k + 1$ elements of $\xi$. Define $L_{k,\pi} \in \Delta(\Xi_k) : L_{k,\pi}(\xi_k = (s_0, a_0, ..., s_k)) = P_{\xi \sim \pi, T, I}([\xi]_k = \xi_k)$. A trajectory lottery $L_\pi$ is then defined as the infinite sequence of $L_{k,\pi}$ values: $L_\pi := (L_{0,\pi}, L_{1,\pi}, L_{2,\pi}, ...)$. The set of trajectory lotteries generated by any policy in an environment $E = (\mathcal{S}, \mathcal{A}, \mathcal{T}, \mathcal{I})$ is defined to be $L_{\Pi_{S,A}} := \{L_\pi | \pi \in \Pi_{S,A}\}$.

## 3 RESULTS

In this section, we present our expressivity results. For each of the 289 *ordered* pairs of formalisms $(X, Y)$, we prove either that $X \succeq_{EXPR} Y$ or that $X \not\succeq_{EXPR} Y$.[5] Figure 1 depicts the expressivity relationships between all formalisms. Note that the *absence* of a sequence of arrows from one formalism to another is as significant as the presence of a sequence of arrows: the presence of a sequence of arrows means that the first formalism can express all policy orderings that the second can express, and the absence of a sequence of arrows means that there exists a policy ordering that the second formalism can express and the first cannot express. A considerable part of our contribution in this work is to demonstrate the expressive limitations of these formalisms, and these limitations are represented by the absence of arrows. In this section we will state and provide intuition for many of the nontrivial positive and negative expressivity results that serve as the basis for the Hasse diagram. The remaining results, and all proofs, can be found in Appendix B.

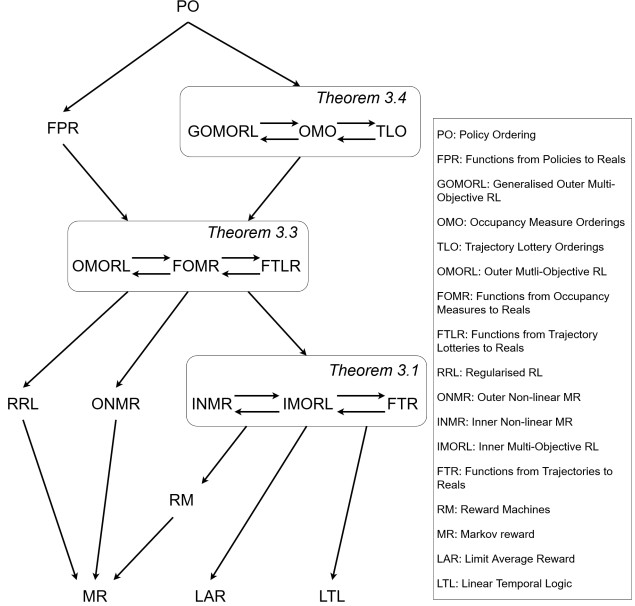

Figure 1: This Hasse diagram displays all expressivity relationships between our formalisms. An arrow or chain of arrows from one formalism to another indicates that the first formalism can express all policy orderings that the second formalism can express, in all environments. Arrows going both directions mean that the formalisms have the same expressivity. If there is no chain of arrows from one formalism to another, there are policy orderings that the latter can express and the former cannot.

### 3.1 POSITIVE RESULTS

Several of the positive connections in the Hasse diagram are relatively trivial; these are discussed in Appendix B.1. Here, we focus on the most substantive results. Proofs for the results in this subsection are available in Appendix B.2 and referenced beside each result.

**Theorem 3.1.** *Inner Nonlinear Markov Rewards (INMR), Inner Multi-Objective RL (IMORL), and Functions from Trajectories to Reals (FTR) are equally expressive. Proof: B.21*

It is straightforward to see that IMORL can express all policy orderings that INMR can express and that FTR can express all policy orderings that IMORL can express (and we show these results

---

[5]We study 17 formalisms, so there are $17^2 = 289$ ordered pairs of formalisms. This includes the ordered pairs of a formalism with itself; trivially, any formalism can express itself. The expressivity results for some other ordered pairs of formalisms are also fairly straightforward, and many results follow from the transitivity of the expressivity relation. Table 2 in the appendix depicts all proof dependencies for these results.

explicitly in Appendix B.1). Therefore, the following proposition is sufficient to conclude that INMR, IMORL, and FTR are equally expressive:

**Proposition 3.2.** *Any policy ordering expressible with Functions from Trajectories to Reals (FTR) can be expressed with Inner Nonlinear Markov Rewards (INMR). Proof: B.21*

Briefly, our proof demonstrates that it is possible to express an arbitrary function from trajectories to reals $f_{FTR}$ with an INMR specification $(R, f_{INMR}, \gamma)$ by selecting $R$ and $\gamma$ so that the trajectory return function $G$ is injective, and then setting $f_{INMR}$ equal $f_{FTR} \circ G^{-1}$. The full proof is in Appendix B.2. This proof notably relies on allowing $f_{INMR}$ to be an arbitrary function that need not satisfy properties such as differentiability, continuity, or monotonicity. Since many possible nonequivalent constraints might be of interest, we choose to define a very general version of INMR; we discuss the implications of this choice in the next section, and in more detail in Appendix A.4.

**Theorem 3.3.** *Outer Multi-Objective Reinforcement Learning (OMORL), Functions from Occupancy Measures to Reals (FOMR), and Functions from Trajectory Lotteries to Reals (FTLR) are equally expressive. Proof: B.24*

We show that a) two policies have the same occupancy measure if and only if they generate the same trajectory lottery, and b) it is always possible to specify an OMORL objective such that two policies have the same policy evaluation vector $(\vec{J}(\pi) := \langle J_1(\pi), ..., J_k(\pi) \rangle)$ if and only if they have the same occupancy measure. Therefore, the expressivity of a function from policy evaluation vectors to reals is exactly the same as a function from occupancy measures to reals and a function from trajectory lotteries to reals. It is worth noting that in order to possess this expressive power, OMORL requires access to $|\mathcal{S}||\mathcal{A}||\mathcal{S}|$ reward functions for any environment $E = (\mathcal{S}, \mathcal{A}, \mathcal{T}, \mathcal{I})$. It may be infeasible to specify and optimise an objective with so many reward functions in practice; in Section 4 and Appendix A.4, we discuss the possibility of restricting the number of reward functions a multi-reward objective can include.

Our third theorem is very similar to Theorem 3.3:

**Theorem 3.4.** *Generalised Outer Multi-Objective RL (GOMORL), Occupancy Measures Orderings (OMO), and Trajectory Lottery Orderings (TLO) are equally expressive. Proof: B.25*

The proof of this theorem utilises the same lemmas as the proof of Theorem 3.3. Total preorders over policy evaluation vectors, occupancy measure, and trajectory lotteries are all equally expressive, because two policies share a trajectory lottery if and only if they share an occupancy measure if and only if they share a policy evaluation vector (with appropriately selected reward functions).

## 3.2 NEGATIVE RESULTS

All proofs for the results in this subsection can be found in Appendix B.3.

**Proposition 3.5.** *There exists a policy ordering that Linear Temporal Logic (LTL) and Limit Average Rewards (LAR) can express but Reward Machines (RM) and Markov Rewards (MR) cannot express. Proof: B.26*

RM specifications can only induce policy evaluation functions that are continuous (in the sense that infinitesimal changes to a policy can only lead to infinitesimal changes to the policy evaluation). Note that it must also be true that Markov Rewards (MR) can only induce continuous policy evaluation functions, because as shown in Appendix B.1, RM can express any policy evaluation function that MR can express. LTL and LAR can induce discontinuous policy evaluation functions.

**Proposition 3.6.** *There exists a policy ordering that Markov Rewards (MR) and Limit Average Rewards (LAR) can express but Linear Temporal Logic (LTL) cannot express. Proof: B.24*

An LTL specification can only assign a value of $\varphi(\xi) = 0$ or $\varphi(\xi) = 1$ to a trajectory, since an LTL formula $\varphi$ is either true or false for any given trajectory $\xi$. MR and LAR, on the other hand, can give any scalar-valued rewards. In certain cases, this prevents LTL from differentiating between trajectories — and lotteries over trajectories generated by policies — that MR and LAR can distinguish. For instance, LTL cannot induce any strict ordering of 3 deterministic policies in a deterministic environment.

**Proposition 3.7.** *There exists a policy ordering that Linear Temporal Logic (LTL) and Markov Rewards (MR) can express but Limit Average Rewards (LAR) cannot express. Proof: B.29*

LAR cannot distinguish between policies that generate the same trajectory lottery after a finite amount of time, even if the lotteries they generate over initial trajectory segments are different. LTL and MR can distinguish between policies based on differences that appear only early on in trajectories.

**Proposition 3.8.** *There exists a policy ordering that Outer Nonlinear Markov Rewards (ONMR) can express but Functions from Trajectories to Reals (FTR) cannot express. Proof: B.34*

ONMR can express the objective of making a trajectory occur with at least some desired threshold probability, while being indifferent to achieving probabilities higher than the threshold. For example, this can be achieved (for a particular environment and reward specification) with the wrapper function $f_{ONMR}(x) = \begin{cases} 1 & \text{if } x \geq 0.5, \\ 0 & \text{otherwise} \end{cases}$. By contrast, for an FTR specification ($f_{FTR}$) to incentivise a trajectory, $f_{FTR}$ must assign a higher value to that trajectory than others. However, this would incentivise maximizing the probability of that trajectory, not merely meeting a threshold.

**Proposition 3.9.** *There exists a policy ordering that Functions from Policies to Reals (FPR) can express but Generalised Outer Multi-Objective RL (GOMORL) cannot express. Proof: B.36*

A function from policies to reals can assign different values to any two distinct policies, even if they only differ on states that both policies have probability zero of ever visiting. GOMORL cannot express preferences between policies that are identical on all states visited with nonzero probability.

**Proposition 3.10.** *There exists a policy ordering that Generalised Outer Multi-Objective RL (GO-MORL) can express but Functions from Policies to Reals (FPR) cannot express. Proof: B.37*

There exist lexicographic preference orderings of GOMORL policy evaluation vectors that cannot be represented by any real-valued function (Steen & Seebach (2012)). These orderings of policy evaluation vectors can be used to express tasks in GOMORL that are inexpressible in FPR.

## 4  DISCUSSION AND RELATED WORK

Previous work has attempted to settle the reward hypothesis, which states that "all of what we mean by goals and purposes can be well thought of as maximization of the expected value of the cumulative sum of a received scalar signal (called reward)" (Sutton (2004)). Related to this is the von Neumann-Morgenstern (VNM) utility theorem, which states that an agent's preferences between lotteries over a finite set of outcomes can be represented as ordered according to the expected value of a scalar utility function of the outcomes if and only if these preferences satisfy four axioms of rationality (the VNM axioms) (Von Neumann & Morgenstern (1944)). Consequently, if one's preferences cannot be represented by a utility function, the preferences must violate at least one of these axioms.

The VNM theorem is typically considered in the context of single-decision problems, while RL is applied to sequential decision-making. Without modification, the VNM theorem can be applied to RL settings by taking the outcome space to be a finite set of (full) trajectories. It then states that an agent's preferences about lotteries over a finite number of trajectories can be expressed as a function from trajectories to the reals (i.e., in the FTR formalism) if and only if those preferences satisfy the VNM axioms. Close relatives of the FTR formalism as defined in this work have been studied in Chatterji et al. (2022) and Pacchiano et al. (2021), though in these works the formalism is restricted to trajectories of finite length (among other differences). Other recent work has proven stronger statements about the required *form* of the trajectory return function under the supposition of additional axioms for preferences over temporally extended outcomes that supplement the VNM axioms.

Pitis (2019), Shakerinava & Ravanbakhsh (2022), and Bowling et al. (2022) all identify axioms from which it is possible to prove that preferences regarding temporally extended outcomes can be expressed using Markovian rewards. All three papers identify slightly different axioms that, when supplemented with the VNM axioms, enable this proof. One caveat is that all three papers show that preferences can be represented with a Markovian reward function along with a *transition-dependent* discount factor, rather than the traditional constant discount factor in MDPs.

The VNM theorem and the work that has adapted it to sequential decision-making settings present axioms that must be violated for an objective to be inexpressible with Markov rewards or a function from trajectories to reals. Our work demonstrates several objectives that Markov rewards and functions from trajectories to reals cannot express, which raises the question: are there reasonable

objectives that violate the axioms, or are the objectives we present unreasonable? We elaborate on this tension in Appendix A.5, particularly for violations of the VNM axioms. In our work we also assume a conventional constant discount factor, but future work could compare the expressive power of formalisms that allow transition-dependent discount factors with the formalisms we consider.

Abel et al. (2022) formalise a task in three different ways. One is a set of acceptable policies, another is an ordering over policies and the third is an ordering over trajectories. They find that for each notion of a task there exists an instance that no Markov reward function can capture. The second formalisation, policy orderings, is the one that we use, and we adapt their proof that Markov Rewards cannot express an "XOR" policy ordering to display the expressive limitations of several formalisms.

Multi-Objective RL (MORL) has received considerable attention in the literature. Silver et al. (2021) argues that maximization of a scalar reward is enough to drive most behaviour in natural and artificial intelligence, while Vamplew et al. (2022) argue that multi-objective rewards are necessary. In particular, Vamplew et al. (2022) argue that scalar reward is insufficient to develop and align Artificial General Intelligence. Miura (2023) shows that multi-dimensional reward functions are also not universally expressive; they state necessary and sufficient conditions under which a task defined as a set of acceptable policies can be expressed with a multi-dimensional reward function. They use a definition of MORL where a policy is acceptable if each dimension of reward is above a given lower bound — we generalise this notion for one of our formulations of MORL . Algorithms for solving multi-objective problems have been developed and used in several applications (Hayes et al. (2022); Jalalimanesh et al. (2017); Castelletti et al. (2013)).

Skalse & Abate (2023) demonstrate that some multi-objective, risk-sensitive, and modal tasks cannot be expressed with Markovian rewards. We analyse several formalisms related to this paper: Generalised Outer MORL (GOMORL) is based on this paper's definition of MORL, and our Inner Nonlinear Markov Rewards (INMR) is partially motivated as a generalisation for risk-sensitive tasks. We extend these definitions to study what we call Outer MORL (OMORL), Inner MORL (IMORL), and Outer Nonlinear Markov Rewards (ONMR), which enables a robust comparison of the expressive power of wrapper functions that are applied before and after taking an expected value over trajectories. Skalse & Abate (2023) also use occupancy measures in a proof to demonstrate limitations of Markov Rewards, and we find occupancy measures a sufficiently useful tool for reasoning about formalism expressivity that we include Functions from Occupancy Measures to Reals and Occupancy Measure Orderings as formalisms in our analysis. Our extensions are further justified by their relationships to other prior work: Convex RL (e.g. Zahavy et al. (2023); Mutti et al. (2023)), also referred to as RL with general utilities (Zhang et al., 2020), is equivalent to Functions from Occupancy Measures to Reals (FOMR) except with a convexity constraint on the functions, and RL with vectorial rewards (Cheung, 2019) is a special case of IMORL in the time-average reward setting.

Prior work has studied Reward Machines (RM), Limit Average Rewards (LAR), and Linear Temporal Logic (LTL) as well. Icarte et al. (2018) point out that reward machines can trivially express all objectives that Markov Rewards can express and provide rewards that depend on a finite number of features of the history so far, rather than just the most recent transition. This paper also applies RM specifications to toy tasks. Mahadevan (1996) investigates algorithms for learning goals specified with Limit Average Rewards (LAR), but we have not found any work on the expressivity of LAR.

Littman et al. (2017) study LTL, which was previously utilised in control theory (Bacchus et al., 1970; Puterman, 1994), in an RL setting. The authors mention that many common tasks can be expressed with LTL, and are sometimes easier to express with LTL than with Markov Rewards. Related to LTL is Signal Temporal Logic (STL), an extension of LTL to the continuous time domain, which has a number of desirable properties (Balakrishnan & Deshmukh, 2019; Wang et al., 2023). However, in this work we focus on objective specifications in discrete environments and using discrete time, for which STL is equivalent to LTL.

When attempting to reverse engineer an agent's utility function from its behavior, Maximum Entropy Inverse RL makes an assumption that an agent's distribution over actions has maximum entropy (Ziebart et al. (2008)). Our formalism of Regularised RL relates to this concept by allowing any function of the distributions over actions to be included in the RL objective.

## 4.1 Limitations and Future Work

In this paper, we focus on stationary policies, and define objectives as orderings of stationary policies. In principle, however, policies can be history-dependent; that is, an agent can select actions based on the history rather than just the most recent state. We make the choice to focus on stationary policies because common RL algorithms (such as Q-learning (Watkins & Dayan (1992))) derive exclusively stationary policies from state-action value functions. Many, but not all, of our expressivity results carry over to the history-dependent setting. We hope to see future research extend our study to history-dependent policies, and we discuss this topic in more detail in Appendix A.3.

A few of our results rely heavily on the technical details of the formalisms in ways that may reduce their practical significance. For instance, Theorem 3.1 relies on the ability of Inner Nonlinear Markov Rewards to utilise an arbitrary wrapper function for the trajectory return, and Theorems 3.3 and 3.4 require allowing multi-objective formalisms access to an arbitrary number of reward functions. One direction for future work is to replicate our analysis under various restricting conditions; we discuss some conditions that may be of interest in Appendix A.4.

Another limitation of our results is that we do not assess the important tradeoff between expressivity and tractability. An important area for future work might be to identify which of the more expressive formalisms considered here could be implemented in practice and provide guidance for balancing expressivity and tractability in various practical settings. The research community can address this question gradually by developing new algorithms to solve a range of problems with a variety of formalisms. Understanding the expressivity-tractability tradeoff might also involve considering restricting conditions for the formalisms, since the conditions required to make a formalism tractable to optimise could reduce its expressivity.

Until the tractability of these formalisms is studied more carefully, it is difficult to draw a clear conclusion about which formalisms are best to use in practice. A more immediate takeaway from our results is that many of the formalisms are expressively incommensurable; for example, Limit Average Reward, Linear Temporal Logic, Reward Machines, Regularised RL, and Outer Nonlinear Markov Rewards can each express policy orderings that none of the others can express. This highlights the importance of carefully selecting a formalism for any task of interest; our results elucidate strengths and limitations of different formalisms that can inform these selections.

Future research can also explore reward learning with formalisms other than Markov Rewards. There have been preliminary efforts in the direction of reward learning for Reward Machines (Icarte et al. (2019)), Linear Temporal Logic (Neider & Gavran (2018)), and Limit Average Reward (Wu et al. (2023)), but given the range of objectives that we show cannot be specified with Markov Rewards, it may be important to further develop reward learning alternatives that allow us to reliably express the objectives we desire across a variety of domains.

## 5 Conclusion

This paper provides a complete map of the relative expressivities of seventeen formalisms for specifying objectives in RL. We discovered meaningful expressive limitations to all of these formalisms,[6] including many of the alternatives and extensions to Markov rewards that have been discussed in the existing literature. We also related practical formalisms to a number of theoretical constructs, such as trajectory lotteries and occupancy measures, which help to fill out a richer intuitive picture of where each formalism stands in the expressivity hierarchy. We hope our work will serve as a reference point for future discussions of these methods for specifying temporally extended objectives, as well as provide an impetus for future work that contextualises these results in light of the tradeoff between expressivity and tractability that appears in both policy optimisation and reward learning.

## Reproducibility Statement

All of our results are theoretical. All proofs can be found in Appendix B. For our definitions of environments, objectives, total order and formalisms see Section 2. Any further assumptions have been stated in the proofs.

---

[6]With the exception of Policy Orderings, which by our definition can express all objectives.

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

| Can ROW express COLUMN? | MR | LAR | LTL | RM | INMR | IMORL | FTR | RRL | ONMR | OMORL | FOMR | FTLR | FPR | OMO | TLO | GOMORL | PO |
|---|---|---|---|---|---|---|---|---|---|---|---|---|---|---|---|---|---|
| MR | | 26 | 26 | 31 | 26+13+21 | 26+13+21 | 26+13 | 1+15+35 | 1+15+34 | 1+15+34+7 | 26+13+8+24 | 26+13+8 | 26+13+8+9 | 26+13+8+10+25 | 26+13+8+10 | 1+15+34+7+11 | 26+12 |
| LAR | 29 | | 29 | 29+1 | 29+1+15+21 | 29+1+15+21 | 29+1+15 | 13+35 | 13+34 | 13+34+7 | 13+34+7+24 | 13+34+7+24 | 13+34+7+24+9 | 13+34+7+11+25 | 13+34+7+11 | 13+34+7+11 | 29+12 |
| LTL | 30 | 30 | | 30+15 | 30+1+15+21 | 30+1+15+21 | 30+1+15 | 14+35 | 14+34 | 14+34+7 | 14+34+7+24 | 14+34+7+24 | 14+34+7+24+9 | 14+34+7+11+25 | 14+34+7+11 | 14+34+7+11 | 30+12 |
| RM | 1 | 26 | 26 | | 26+13+21 | 26+13+21 | 26+13 | 15+35 | 15+34 | 15+34+7 | 15+34+7+24 | 15+34+7+24 | 15+34+7+24+9 | 15+34+7+11+25 | 15+34+7+24+10 | 15+34+7+11 | 15+34+12 |
| INMR | 2 | 21+13 | 21+14 | 21+15 | | 21 | 21 | 21+35 | 21+34 | 21+34+7 | 21+34+7+24 | 21+34+7+24 | 21+34+7+24+9 | 21+34+7+11+25 | 21+34+7+24+10 | 21+34+7+11 | 21+34+12 |
| IMORL | 21+2 | 21+13 | 21+14 | 21+15 | 21 | | 21 | 21+35 | 21+34 | 21+34+7 | 21+34+7+24 | 21+34+7+24 | 21+34+7+24+9 | 21+34+7+11+25 | 21+34+7+24+10 | 21+34+7+11 | 21+34+12 |
| FTR | 15+1 | 13 | 14 | 15 | 21 | 21 | | 35 | 34 | 34+7 | 34+7+24 | 34+7+24 | 34+7+24+9 | 34+7+11+25 | 34+7+24+10 | 34+7+11 | 34+12 |
| RRL | 5 | 39 | 31 | 31 | 31+15+21 | 31+15+21 | 31+15 | | 31 | 31+7 | 31+7+24 | 31+7+24 | 31+7+24+9 | 31+7+11+25 | 31+7+24+10 | 31+7+11+25 | 31+12 |
| ONMR | 6 | 32 | 33 | 40 | 40+15+21 | 40+15+21 | 40+15 | 38 | | 34+8+24 | 34+8+24 | 34+8 | 34+8+9 | 34+8+10+25 | 34+8+10 | 34+8+10+25 | 32+12 |
| OMORL | 7+6 | 24+8+13 | 24+8+14 | 24+8+15 | 24+8+21 | 24+8+21 | 24+8 | 24+P2 | 7 | | 24 | 24 | 11+36 | 24+9+37+25 | 24+9+37+25 | 24+9+37 | 11+36+12 |
| FOMR | 24+7+6 | 24+8+13 | 24+8+14 | 24+8+15 | 24+8+21 | 24+8+21 | 24+8 | 24+P2 | 24+7 | 24 | | 24 | 24+11+36 | 24+9+37+25 | 24+9+37+25 | 24+9+37 | 24+9+37+12 |
| FTLR | 8+15+1 | 8+13 | 8+14 | 8+15 | 8+21 | 8+21 | 8 | P2 | 24+7 | 24 | 24 | | 24+11+36 | 9+37+25 | 9+37+25 | 9+37 | 9+37+12 |
| FPR | 9+8+15+1 | 9+8+13 | 9+8+14 | 9+8+15 | 9+8+21 | 9+8+21 | 9+8 | 9+P2 | 9+24+7 | 9+24 | 9+24 | 9 | | 37+25 | 37+25 | 37 | 37+12 |
| OMO | 25+11+7+6 | 25+10+8+13 | 25+10+8+14 | 25+10+8+15 | 25+10+8+21 | 25+10+8+21 | 25+10+8 | 25+10+P2 | 25+11+7 | 25+11 | 25+10+24 | 25+10+24 | 25+36 | | 25 | 25 | 25+36+12 |
| TLO | 10+8+15+1 | 10+8+13 | 10+8+14 | 10+8+15 | 10+8+21 | 10+8+21 | 10+8 | 10+P2 | 25+11+7 | 25+11 | 10+24 | 10 | 25+36 | 25 | | 25 | 25+36+12 |
| GOMORL | 11+7+6 | 25+10+8+13 | 25+10+8+14 | 25+10+8+15 | 25+10+8+21 | 25+10+8+21 | 25+10+8 | 25+10+P2 | 11+7 | 11 | 11+24 | 11+24 | 36 | 25 | 25 | | 36+12 |
| PO | 12 | 12 | 12 | 12 | 12 | 12 | 12 | 12 | 12 | 12 | 12 | 12 | 12 | 12 | 12 | 12 | |

Table 2: Comprehensive table of expressivity comparisons. A green cell indicates that the formalism in the ROW can express the formalism in the COLUMN, and a red cell indicates that it cannot. The numbers in the cells are the numbers of the propositions which prove the result in question.

# A  APPENDIX: FURTHER DISCUSSION

## A.1  POLICY ORDERINGS AS THE MEASURE OF EXPRESSIVITY

We opted to compare the expressivities of different objective-specification formalisms in terms of the policy orderings they can induce because the policy ordering is the most fine-grained unit of analysis (that we are aware of) for this purpose. We explain why we believe it would be less informative to consider only a formalism's ability to induce a desired optimal policy in Section 2.1. Another alternative is to consider a formalism's ability to induce a desired set of acceptable policies (SOAP), as proposed in Abel et al. (2022). SOAPs are constructed by choosing a cutoff value to divide the policies into two classes (i.e., the acceptable and the unacceptable), and discarding all information about how the policies are ordered within each class. Thus, it is straightforward to see that policy orderings contain strictly more information than do SOAPs, and so any difference in expressivity that is captured by a SOAP task definition is also captured by the policy ordering task definition. The latter is therefore the one that is maximally sensitive to expressive differences.

Additionally, our choice to use the total preorder over policies as the unit of analysis is more continuous with the theoretical framework standardly employed in the formal theory of individual decision-making. In that context, the central question takes the form: Which preference orderings can be represented by a utility function (of a given sort)? For each of the formalisms considered in our paper, we ask a similar question: Which policy orderings can be represented by an objective specification in the formalism? We thought it would be valuable to maintain the parallels between these two questions in our paper (and discuss the connection explicitly in Section 4 and Appendix A.5).

## A.2  INDUCING TOTAL PREORDERS

Most of the formalisms in our work induce a scalar policy evaluation function, which in turn produces a total preorder on the set of policies because the "$\geq$" relation on $\mathbb{R}$ is a total preorder. The four formalisms that do not induce a scalar policy evaluation function (Occupancy Measure Orderings, Trajectory Lottery Orderings, Generalised Outer Multi-Objective RL, and Policy Orderings) all include a total preorder on some set in their objective specifications, and the policy orderings they produce inherit transitivity and strong connectedness from the specified total preorder.

## A.3  HISTORY-DEPENDENT POLICIES

As mentioned in Section 4.1, one limitation of our work is that an analysis of formalisms' ability to express orderings over the set of stationary policies may not match a similar analysis that considers orderings over the set of history-dependent policies. It is worth noting that many of our expressivity results directly carry over to history-dependent policies. Firstly, if formalism A can express a stationary policy ordering that is not expressible by formalism B, then clearly A can express an ordering over history-dependent policies that B cannot. So every negative expressivity result (i.e., every red box in Table 2) carries over directly. Further, several of our positive proofs do not assume that we are considering only stationary policies. For example, all trivial results in Appendix B.1 apply for history-dependent policies as well. Unfortunately, not all of our results carry over to the history-dependent setting: it is fairly straightforward to show that a reward machine can provide different amounts of reward to two history-dependent policies that have the same occupancy measure, so Occupancy Measure Orderings (OMO) cannot express all history-dependent policy orderings that Reward Machines (RM) can express. This is a significant departure from our findings, in which OMO is near the top of the Hasse diagram and RM is near the bottom. We consider it an interesting direction for future research to round out a comprehensive comparison of expressivity for history-dependent policies, especially given that some formalisms (such as Functions from Trajectories to Reals and Reward Machines) seem particularly well-suited to expressing tasks where history-dependent policies are essential. To support such an endeavor, we include an incomplete table of expressivity relationships for history-dependent policies below (Table 3).

## A.4  EXPRESSIVITY UNDER RESTRICTING CONDITIONS

One concern about the practical utility of some of our results as they stand is that they are largely blind to the ease or difficulty of using the theoretically available capabilities of the more expressive

formalisms for training models on objectives that cannot be captured by less expressive formalisms. Some of our proofs lean heavily on technicalities of the formal definitions we offer, with little regard for whether those technicalities can reasonably be trusted to lie available to RL algorithms in practice. For example, the proof that INMR can express FTR relies on the fact that the Markovian trajectory return can be chosen to be an injective function into the reals. In practice, however, computers can only store numbers with finite precision, so we cannot assume that arbitrarily close real numbers will always be distinguishable to an RL algorithm.

An obvious suggestion for how this problem might be solved is to attempt to redefine each formalism in such a way that unreasonable technicalities are eliminated. Unfortunately, this strategy runs into a familiar problem: virtually any formal definition of an informal concept can be "gamed" to flaunt the intended spirit of that concept. For this reason, we believe an alternative strategy might be more fruitful: Replicate these results under various carefully chosen restricting conditions, whose purpose is to ensure that our theoretically sound results can be trusted to hold up in practice.

We offer the following (highly incomplete) list of potential restricting conditions:

1. Introduce a cutoff to the precision of the inputs available to the functions wrapping the trajectory return(s) in INMR and IMORL, and wrapping the policy value(s) in ONMR and OMORL. This cutoff could correspond to the precision with which modern-day computers can store real numbers.

2. Introduce various other constraints on these wrapper functions. For instance, we might require that they be continuous, differentiable, monotonic, analytic, or computable, among other options.

3. Introduce an upper limit to the number of reward functions that may be employed in IMORL, OMORL, and GOMORL. In particular, do not assume that this number may reach (or exceed) the number of transitions in an environment $|S \times A \times S|$.

### A.5 CONNECTION TO THE VNM THEOREM

It is interesting to consider our results in light of the Von Neumann-Morgenstern (VNM) Utility Theorem. In particular, subject to a few assumptions, we have the following:
*Any trajectory lottery ordering that cannot be induced by FTR must violate at least one of the VNM axioms.*

To apply the VNM axioms, we need to restrict to an environment $E$ in which only a finite number of trajectories are possible, $\Xi_E = \{\xi_1, ..., \xi_n\}$. Then, the VNM axioms read as follows:

1. *Total Preorder*: $\succeq$ over $\Delta(\Xi_E)$ is complete and transitive.
2. *Continuity*: For all $L_{\pi_1}, L_{\pi_2}, L_{\pi_3} \in \Delta(\Xi_E)$ such that $L_{\pi_1} \succ L_{\pi_2}$ and $L_{\pi_2} \succ L_{\pi_3}$, there exist $\alpha, \beta \in (0, 1)$ such that $L_{\pi_1} \alpha L_{\pi_3} \succ L_{\pi_2}$ and $L_{\pi_2} \succ L_{\pi_1} \beta L_{\pi_3}$.
3. *Independence*: For all $L_{\pi_1}, L_{\pi_2}, L_{\pi_3} \in \Delta(\Xi_E)$ and for all $\alpha \in (0, 1)$:
   $L_{\pi_1} \succ L_{\pi_2} \iff L_{\pi_1} \alpha L_{\pi_3} \succ L_{\pi_2} \alpha L_{\pi_3}$.

(Here $L\alpha M = \alpha L + (1 - \alpha)M$ is the mixing operation.)

**VNM Theorem**: $\succeq$ over $\Delta(\Xi_E)$ respects *Total Preorder*, *Continuity*, and *Independence* **if and only if** there exists a function $u : \Xi_E \to \mathbb{R}$ such that for all $L_{\pi_1}, L_{\pi_2} \in \Delta(\Xi_E)$,

$$L_{\pi_1} \succeq L_{\pi_2} \iff \mathbb{E}_{\xi \sim \pi_1, T, I}[u(\xi)] \geq \mathbb{E}_{\xi \sim \pi_2, T, I}[u(\xi)]$$

Under the assumption that our policy preferences are determined by our preferences over trajectory-lotteries (such that $L_{\pi_1} \succeq L_{\pi_2} \implies \pi_1 \succeq \pi_2$), the RHS simply states the condition that there exists an FTR objective specification, given by $f_{FTR} = u$, which induces this policy ordering. Thus, **in an environment in which there are only finitely many possible trajectories, any policy ordering that FTR cannot induce must violate either *Continuity* or *Independence*** (*Total Preorder* is guaranteed to hold by definition). Insofar as it is reasonable to evaluate policies based on the trajectory-lotteries that they give rise to, and insofar as our preferences over trajectory lotteries ought to conform to *Continuity* and *Independence*, we might justifiably wonder whether there is much to be gained from moving up the Hasse diagram beyond FTR.

| Can ROW express COLUMN? | MR | LAR | LTL | RM | INMR | IMORL | FTR | RRL | ONMR | OMORL | FOMR | FTLR | FPR | OMO | TLO | GOMORL | PO |
|---|---|---|---|---|---|---|---|---|---|---|---|---|---|---|---|---|---|
| MR | | 26 | 26 | 31 | 26 + 13 + 21 | 26 + 13 + 21 | 26 + 13 | 1 + 15 + 35 | 1 + 15 + 34 | 1 + 15 + 34 + 7 | 26 + 13 + 8 + 24 | 26 + 13 + 8 | 26 + 13 + 8 + 9 | 26 + 13 + 8 + 10 + 25 | 26 + 13 + 8 + 10 | 1 + 15 + 34 + 7 + 11 | 26 + 12 |
| LAR | 29 | | 29 | 29 + 1 | 29 + 1 + 15 + 21 | 29 + 1 + 15 + 21 | 29 + 1 + 15 | 13 + 35 | 13 + 34 | 13 + 34 + 7 | 13 + 34 + 7 + 24 | 13 + 34 + 7 + 24 | 13 + 34 + 7 + 24 + 9 | 13 + 34 + 7 + 11 + 25 | 13 + 34 + 7 + 11 | 13 + 34 + 7 + 11 | 29 + 12 |
| LTL | 30 | 30 | | 30 + 15 | 30 + 1 + 15 + 21 | 30 + 1 + 15 + 21 | 30 + 1 + 15 | 14 + 35 | 14 + 34 | 14 + 34 + 7 | 14 + 34 + 7 + 24 | 14 + 34 + 7 + 24 | 14 + 34 + 7 + 24 + 9 | 14 + 34 + 7 + 11 + 25 | 14 + 34 + 7 + 11 | 14 + 34 + 7 + 11 | 30 + 12 |
| RM | 1 | 26 | 26 | | 26 + 13 + 21 | 26 + 13 + 21 | 26 + 13 | 15 + 35 | 15 + 34 | 15 + 34 + 7 | 15 + 34 + 7 + 24 | 15 + 34 + 7 + 24 | 15 + 34 + 7 + 24 + 9 | 15 + 34 + 7 + 11 + 25 | 15 + 34 + 7 + 24 + 10 | 15 + 34 + 7 + 11 | 15 + 34 + 12 |
| INMR | 2 | 21 + 13 | 21 + 14 | 21 + 15 | | 21 | 21 | 21 + 35 | 21 + 34 | 21 + 34 + 7 | 21 + 34 + 7 + 24 | 21 + 34 + 7 + 24 | 21 + 34 + 7 + 24 + 9 | 21 + 34 + 7 + 11 + 25 | 21 + 34 + 7 + 24 + 10 | 21 + 34 + 7 + 11 | 21 + 34 + 12 |
| IMORL | 21 + 2 | 21 + 13 | 21 + 14 | 21 + 15 | 21 | | 21 | 21 + 35 | 21 + 34 | 21 + 34 + 7 | 21 + 34 + 7 + 24 | 21 + 34 + 7 + 24 | 21 + 34 + 7 + 24 + 9 | 21 + 34 + 7 + 11 + 25 | 21 + 34 + 7 + 24 + 10 | 21 + 34 + 7 + 11 | 21 + 34 + 12 |
| FTR | 15 + 1 | 13 | 14 | 15 | 21 | 21 | | 35 | 34 | 34 + 7 | 34 + 7 + 24 | 34 + 7 + 24 | 34 + 7 + 24 + 9 | 34 + 7 + 11 + 25 | 34 + 7 + 24 + 10 | 34 + 7 + 11 | 34 + 12 |
| RRL | 5 | 39 | 31 | 31 | 31 + 15 + 21 | 31 + 15 + 21 | 31 + 15 | | 31 | 31 + 7 | 31 + 7 + 24 | 31 + 7 + 24 | 31 + 7 + 24 + 9 | 31 + 7 + 11 + 25 | 31 + 7 + 24 + 10 | 31 + 7 + 11 + 25 | 31 + 12 |
| ONMR | 6 | 32 | 33 | 40 | 40 + 15 + 21 | 40 + 15 + 21 | 40 + 15 | 38 | | 34 + 8 + 24 | 34 + 8 + 24 | 34 + 8 | 34 + 8 + 9 | 34 + 8 + 10 + 25 | 34 + 8 + 10 | 34 + 8 + 10 + 25 | 32 + 12 |
| OMORL | 7 + 6 | | | | | | | | 7 | | 24 | | 11 + 36 | 24 + 9 + 37 + 25 | 24 + 9 + 37 + 25 | 24 + 9 + 37 | 11 + 36 + 12 |
| FOMR | 24 + 7 + 6 | | | | | | | | | | | 24 + 11 + 36 | 24 + 9 + 37 + 25 | 24 + 9 + 37 + 25 | 24 + 9 + 37 + 25 | 24 + 9 + 37 | 24 + 9 + 37 + 12 |
| FTLR | 8 + 15 + 1 | 8 + 13 | 8 + 14 | 8 + 15 | 8 + 21 | 8 + 21 | 8 | P2 | 24 + 7 | 24 | 24 | | 24 + 11 + 36 | 9 + 37 + 25 | 9 + 37 + 25 | 9 + 37 | 9 + 37 + 12 |
| FPR | 9 + 8 + 15 + 1 | 9 + 8 + 13 | 9 + 8 + 14 | 9 + 8 + 15 | 9 + 8 + 21 | 9 + 8 + 21 | 9 + 8 | 9 + P2 | 9 + 24 + 7 | 9 + 24 | 9 + 24 | 9 | | 37 + 25 | 37 + 25 | 37 | 37 + 12 |
| OMO | 25 + 11 + 7 + 6 | | | A.3 | A.3 + 21 + 15 | A.3 + 21 + 15 | A.3 + 15 | | | | | A.3 + 15 + 8 | 25 + 36 | | | A.3 + 15 + 8 + 10 | 25 + 36 + 12 |
| TLO | 10 + 8 + 15 + 1 | 10 + 8 + 13 | 10 + 8 + 14 | 10 + 8 + 15 | 10 + 8 + 21 | 10 + 8 + 21 | 10 + 8 | 10 + P2 | 25 + 11 + 7 | 25 + 11 | 10 + 24 | 10 | 25 + 36 | 25 | | 25 | 25 + 36 + 12 |
| GOMORL | 11 + 7 + 6 | | | | | | | | | 11 + 7 | 11 | 11 + 24 | | 36 | 25 | | 36 + 12 |
| PO | 12 | 12 | 12 | 12 | 12 | 12 | 12 | 12 | 12 | 12 | 12 | 12 | 12 | 12 | 12 | 12 | |

Table 3: Incomplete table of expressivity comparisons using history-dependent policies. A green cell indicates that the formalism in the ROW can express all history-dependent policy orderings that the formalism in the COLUMN can express, a red cell indicates that the formalism in the ROW cannot do this, and a white cell indicates that the result is not yet known. The numbers in the cells are the numbers of the propositions which prove the result in question.

# B   APPENDIX: THEOREMS AND PROOFS

## B.1   TRIVIAL RESULTS OF EXPRESSIVITY($X \succeq Y$)

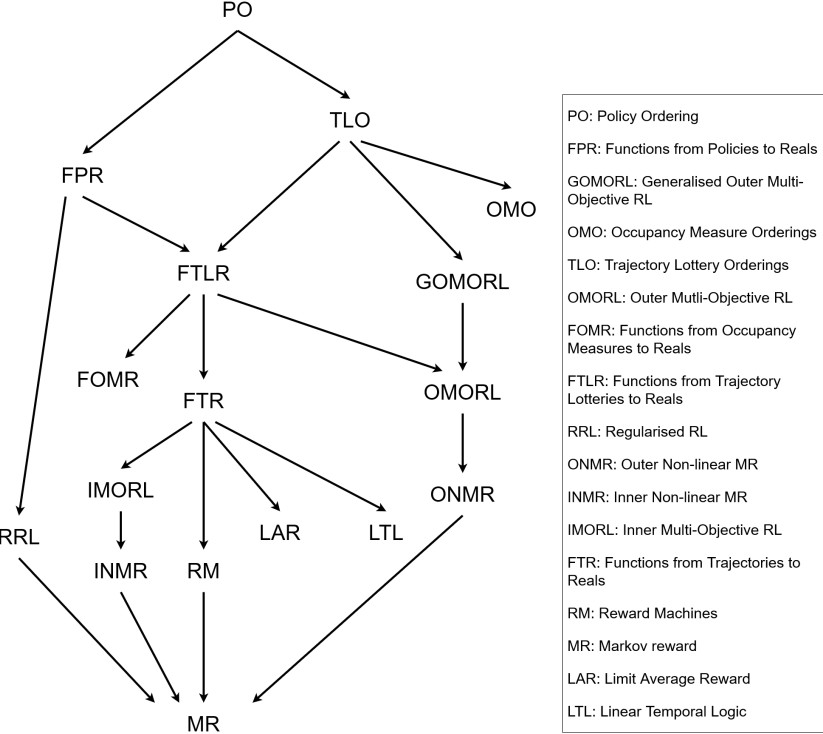

| | |
|---|---|
| PO: Policy Ordering | |
| FPR: Functions from Policies to Reals | |
| GOMORL: Generalised Outer Multi-Objective RL | |
| OMO: Occupancy Measure Orderings | |
| TLO: Trajectory Lottery Orderings | |
| OMORL: Outer Mutli-Objective RL | |
| FOMR: Functions from Occupancy Measures to Reals | |
| FTLR: Functions from Trajectory Lotteries to Reals | |
| RRL: Regularised RL | |
| ONMR: Outer Non-linear MR | |
| INMR: Inner Non-linear MR | |
| IMORL: Inner Multi-Objective RL | |
| FTR: Functions from Trajectories to Reals | |
| RM: Reward Machines | |
| MR: Markov reward | |
| LAR: Limit Average Reward | |
| LTL: Linear Temporal Logic | |

Figure 2: This diagram displays straightforward inclusions of formalisms that are not necessarily strict. Unlike in Figure 1, the absence of a sequence of arrows between two formalisms does not mean anything here. This diagram is simply a useful guide to some basic positive results of expressivity proven in this section.

The propositions in this section are proven formally, but each proof is preceded by an intuitive argument. It will likely be helpful to have Table 1 readily available for reference while reading these results and proofs.

**Proposition B.1** ($RM \succeq_{EXPR} MR$). *Any policy ordering expressible with Markov Rewards can be expressed with Reward Machines.*

**Intuition:** A reward machine can use different reward functions depending on aspects of the history it has seen so far. However, it can also use a constant reward function, in which case it is equivalent to a Markov Rewards specification.

*Proof.* Let $(\mathcal{R}_{MR}, \gamma_{MR})$ be an arbitrary MR objective specification in an arbitrary environment. Then the following RM specification $(U, u_0, \delta_U, \delta_{\mathcal{R}}, \gamma)$ expresses the same policy ordering:

- $U := \{u_0\}$

- $u_0 := u_0$

- $\forall s \in S, \delta_U(u_0, s) := u_0$, where $S$ is the set of states in the environment

- $\delta_{\mathcal{R}}(u_0, u_0) := \mathcal{R}_{MR}$

- $\gamma := \gamma_{MR}$

Since the specified reward machine always utilises the same reward function and discount factor as the MR specification, it yields the same exact policy evaluation function.

$$J_{RM}(\pi) = \mathbb{E}_{\xi \sim \pi, T, I, \delta_U, u_0} \left[ \sum_{t=0}^{\infty} \gamma^t \mathcal{R}_t(s_t, a_t, s_{t+1}) \right], \text{ where } \mathcal{R}_t = \delta_{\mathcal{R}}(u_t, u_{t+1})$$

$$= \mathbb{E}_{\xi \sim \pi, T, I} \left[ \sum_{t=0}^{\infty} \gamma_{MR}^t \mathcal{R}_{MR}(s_t, a_t, s_{t+1}) \right], \text{ since } u_t = u_0 \forall t \text{ and } \delta_{\mathcal{R}}(u_0, u_0) = \mathcal{R}_{MR}$$

$$= J_{MR}(\pi)$$

Both RM and MR derive policy orderings directly from the policy evaluation functions, so this means the two specifications induce the same policy ordering. Therefore, for any MR specification in any environment, it is possible to construct an RM specification which expresses the same policy ordering. $\square$

**Proposition B.2** ($INMR \succeq_{EXPR} MR$). *Any Markovian Reward specification can be captured by an Inner Nonlinear Markovian Reward specification. ($\forall E \in Envs, \ Ord_{MR}(E) \subseteq Ord_{INMR}(E)$)*

**Intuition:** If the wrapper function in INMR is set to be the identity function, then that function becomes idle, and the INMR policy evaluation function reduces to the MR policy evaluation function:

$$J_{INMR}(\pi) := \overset{E,\pi}{\underset{\xi}{\mathbb{E}}} [f(G(\xi))] = \overset{E,\pi}{\underset{\xi}{\mathbb{E}}} [G(\xi)] =: J_{MR}(\pi)$$

*Proof.* We must show that for any environment $E$ and Markovian Reward specification $O_{MR} = (\mathcal{R}, \gamma)$, there is an Inner-nonlinear specification $O_{INMR} = (\tilde{\mathcal{R}}, \tilde{f}, \tilde{\gamma})$ such that $\preceq_{E,O_{INMR}}$ is identical to $\preceq_{E,O_{MR}}$. We construct $O_{INMR}$ as follows:

- $\tilde{\mathcal{R}} := \mathcal{R}$

- $\tilde{f} : \mathbb{R} \to \mathbb{R}$ is the identity ($\tilde{f}(x) = x$).

- $\tilde{\gamma} := \gamma$

We can see immediately that $J_{E,O_{MR}}(\pi) = J_{E,O_{INMR}}(\pi)$ for all $\pi \in \Pi_E$, and thus these two specifications induce the same policy ordering. $\square$

**Proposition B.3** ($IMORL \succeq_{EXPR} INMR$). *Any policy ordering expressible with an Inner Nonlinear Markov Reward specification can be expressed with an Inner Multi-Objective RL specification.*

**Intuition:** IMORL can always choose to use the same reward function, wrapper function, and discount factor as INMR, and can always choose not to use more than one reward function.

*Proof.* Let $(\mathcal{R}_{INMR}, f_{INMR}, \gamma_{INMR})$ be an arbitrary INMR objective specification in an arbitrary environment. Construct the IMORL specification $(k, \mathcal{R}, f, \gamma)$ as follows:

- $k = 1$

- $\mathcal{R} = \mathcal{R}_{INMR}$ (since $k = 1$, $\mathbb{R}^k = \mathbb{R}$ and $\mathcal{R}$ is a function from $\mathcal{S} \times \mathcal{A} \times \mathcal{S}$ to $\mathbb{R}$)

- $f = f_{INMR}$

- $\gamma = \gamma_{INMR}$

We can see immediately that $J_{IMORL}(\pi) = J_{INMR}(\pi)$ for all $\pi \in \Pi_E$, and thus these two specifications induce the same policy ordering. $\square$

**Proposition B.4** ($FTR \succeq_{EXPR} IMORL$). *Any policy ordering expressible with an Inner Multi-Objective RL specification can be expressed with a Function from Trajectories to Reals.*

**Intuition:** The expression $f(G_1(\xi), ..., G_k(\xi))$ in the IMORL policy evaluation function is a function from trajectories to reals.

*Proof.* Let $(k, \mathcal{R}, f_{IMORL}, \gamma)$ be an arbitrary IMORL objective specification in an arbitrary environment. Then the following FTR specification ($f_{FTR}$) expresses the same policy ordering:

- $f_{FTR}(\xi) = f_{IMORL}\left(\sum\limits_{t=0}^{\infty} \gamma^t \mathcal{R}_1(s_t, a_t, s_{t+1}), ..., \sum\limits_{t=0}^{\infty} \gamma^t \mathcal{R}_k(s_t, a_t, s_{t+1})\right)$

Here, $\mathcal{R}_i(s, a, s') := \mathcal{R}(s, a, s')[i]$. This is a function from trajectories to reals because a trajectory $\xi = (s_0, a_0, s_1, a_1, ...)$ uniquely defines $(s_t, a_t, s_{t+1})$ for all $t$. $\qquad\square$

**Proposition B.5** ($RRL \succeq_{EXPR} MR$). *Any policy ordering expressible with Markov Rewards can be expressed with Regularised RL. ($\forall E \in Envs, \; Ord_{MR}(E) \subseteq Ord_{RRL}(E)$)*

**Intuition:** The additional term in the RRL objective, $\alpha F[\pi(s)]$, can be set to zero by selecting $\alpha = 0$. This makes the RRL objective identical to the MR objective.

*Proof.* We must show that, for any environment $E$ and Markovian Reward specification $O_{MR} = (\mathcal{R}, \gamma)$, there is a Regularised RL Specification $O_{RRL} = (\tilde{\mathcal{R}}, \tilde{\alpha}, \tilde{F}, \tilde{\gamma})$ such that $\preceq_{E, O_{RRL}}$ is identical to $\preceq_{E, O_{MR}}$. We construct $O_{RRL}$ as follows:

- $\tilde{\mathcal{R}} := \mathcal{R}$
- $\tilde{F} : \Delta A \to \mathbb{R}$ is any function.
- $\tilde{\alpha} := 0$
- $\tilde{\gamma} := \gamma$

We can see immediately that $J_{E, O_{MR}}(\pi) = J_{E, O_{RRL}}(\pi)$ for all $\pi \in \Pi_E$, and so these two specifications induce the same policy ordering. $\qquad\square$

**Proposition B.6** ($ONMR \succeq_{EXPR} MR$). *Any policy ordering expressible with a Markov Reward specification can be expressed with an Outer Nonlinear Markov Reward specification.*

**Intuition:** ONMR can specify the same reward function and discount factor as MR, then select the identity as a wrapper function so that the function does not affect anything.

*Proof.* Let $(\mathcal{R}_{MR}, \gamma_{MR})$ be an arbitrary MR objective specification in an arbitrary environment. Construct an ONMR specification $(\mathcal{R}_{ONMR}, f, \gamma_{ONMR})$ as follows:

- $\mathcal{R}_{ONMR} = \mathcal{R}_{MR}$
- $f : \mathbb{R} \to \mathbb{R}$ is the identity function, $f(x) = x$
- $\gamma_{ONMR} = \gamma_{MR}$

We can see immediately that $J_{E, O_{ONMR}}(\pi) = J_{E, O_{MR}}(\pi)$ for all $\pi \in \Pi_E$, and so these two specifications induce the same policy ordering. $\qquad\square$

**Proposition B.7** ($OMORL \succeq_{EXPR} ONMR$). *Any policy ordering expressible with an Outer Nonlinear Markov Reward specification can be expressed with an Outer Multi-Objective RL specification.*

**Intuition:** The OMORL specification can use the same reward function, discount factor, and wrapper function as ONMR, and can simply not use more than one reward function.

*Proof.* Let $(\mathcal{R}_{ONMR}, f_{ONMR}, \gamma_{ONMR})$ be an arbitrary ONMR objective specification in an arbitrary environment. Construct an OMORL specification $(k, \mathcal{R}, f, \gamma)$ as follows:

- $k = 1$

- $\mathcal{R} = \mathcal{R}_{ONMR}$ (since $k = 1$, $\mathbb{R}^k = \mathbb{R}$ and $\mathcal{R}$ is a function from $\mathcal{S} \times \mathcal{A} \times \mathcal{S}$ to $\mathbb{R}^k = \mathbb{R}$)

- $f = f_{ONMR}$

- $\gamma = \gamma_{ONMR}$

We can see immediately that $J_{E,O_{OMORL}}(\pi) = J_{E,O_{ONMR}}(\pi)$ for all $\pi \in \Pi_E$, and so these two specifications induce the same policy ordering. $\square$

**Proposition B.8** ($FTLR \succeq_{EXPR} FTR$). *Any policy ordering expressible with Functions from Trajectories to Reals can be expressed with Functions from Trajectory Lotteries to Reals.*

**Intuition:** One way to evaluate a trajectory lottery is to assign values to each individual trajectory and then use the probabilities from the lottery to take an expectation. A function from trajectory lotteries to the reals that does this is equivalent to an FTR specification that assigns the same values to individual trajectories.

*Proof.* Let $(f_{FTR})$ be an arbitrary FTR objective specification in an arbitrary environment. Then the following FTLR specification $(f_{FTLR})$ expresses the same policy ordering:

- $f_{FTLR}(L_\pi) = \mathbb{E}_\xi^{E,\pi}[f_{FTR}(\xi)]$

Here, $L_\pi$ is the lottery over trajectories that is produced by policy $\pi$ in the environment. Since $J_{FTLR}(\pi) := f_{FTLR}(L_\pi)$, the equivalence of $J_{FTLR}$ and $J_{FTR}$ is immediate: $J_{FTLR}(\pi) := f_{FTLR}(L_\pi) = \mathbb{E}_\xi^{E,\pi}[f_{FTR}(\xi)] =: J_{FTR}(\pi)$. $\square$

**Proposition B.9** ($FPR \succeq_{EXPR} FTLR$). *Any policy ordering expressible with Functions from Trajectory Lotteries to Reals can be expressed with Functions from Policies to Reals.*

**Intuition:** An FTLR specification assigns a value to each policy based on the trajectory lottery it generates. This is evidently expressible as a policy evaluation function.

*Proof.* Let $(f_{FTLR})$ be an arbitrary FTLR objective specification in an arbitrary environment. Then the following FPR specification $(J_{FPR})$ expresses the same policy ordering:

- $J_{FPR}(\pi) := f_{FTLR}(L_\pi)$

Here, $L_\pi$ is the lottery over trajectories that is produced by policy $\pi$ in the environment. Since $J_{FTLR}(\pi) := f_{FTLR}(L_\pi)$, the equivalence of $J_{FTLR}$ and $J_{FPR}$ is immediate. $\square$

**Proposition B.10** ($TLO \succeq_{EXPR} FTLR$). *Any policy ordering expressible with a Function from Trajectory Lotteries to Reals can be expressed with a Trajectory Lottery Ordering.*

**Intuition:** A function from trajectory lotteries to the reals induces a total preorder because the "$\geq$" relation on $\mathbb{R}$ is a total preorder. This same total preorder can always be expressed directly over the trajectory lotteries as well.

*Proof.* Let $(f_{FTLR})$ be an arbitrary FTLR objective specification in an arbitrary environment. Then the following TLO specification $(\succeq_L)$ expresses the same policy ordering:

- $L_{\pi_1} \succeq_L L_{\pi_2} \iff f_{FTLR}(L_{\pi_1}) \geq f_{FTLR}(L_{\pi_2})$

Here, $L_\pi$ is the lottery over trajectories that is produced by policy $\pi$ in the environment. This $\succeq_L$ is a valid total preorder on $L_\Pi$ because it is transitive and strongly connected.

Now:

$$\begin{aligned}
\pi_1 \succeq_{O_{TLO}} \pi_2 &\iff L_{\pi_1} \succeq_L L_{\pi_2} \\
&\iff f_{FTLR}(L_{\pi_1}) \geq f_{FTLR}(L_{\pi_2}) \\
&\iff J_{FTLR}(\pi_1) \geq J_{FTLR}(\pi_2) \\
&\iff \pi_1 \succeq_{O_{FTLR}} \pi_2
\end{aligned}$$

$\square$

**Proposition B.11** ($GOMORL \succeq_{EXPR} OMORL$). *Any policy ordering expressible with an Outer Multi-Objective RL specification can be expressed with a Generalised Outer Multi-Objective RL specification.*

**Intuition:** An OMORL specification orders policies according to the values assigned by a function from policy-evaluation vectors $\vec{J}(\pi)$ to the reals. A GOMORL specification can directly order the policy evaluation vectors the same way. That is, $f(\vec{J_1}) \geq f(\vec{J_2}) \iff \vec{J_1} \succeq_J \vec{J_2}$.

*Proof.* Let $(k_{OMORL}, \mathcal{R}_{OMORL}, f, \gamma_{OMORL})$ be an arbitrary OMORL objective specification in an arbitrary environment. Then the following GOMORL specification $(k_{GOMORL}, \mathcal{R}_{GOMORL}, \gamma_{GOMORL}, \succeq_J)$ expresses the same policy ordering:

- $k_{GOMORL} = k_{OMORL} = k$

- $\mathcal{R}_{GOMORL} = \mathcal{R}_{OMORL}$

- $\gamma_{GOMORL} = \gamma_{OMORL}$

- $\forall \vec{J_1}, \vec{J_2} \in \mathbb{R}^k : f(\vec{J_1}) \geq f(\vec{J_2}) \iff \vec{J_1} \succeq_J \vec{J_2}$

With this specification:

$$\begin{aligned}
\pi_1 \succeq_{O_{OMORL}} \pi_2 &\iff f(\vec{J}(\pi_1)) \geq f(\vec{J}(\pi_2)) \\
&\iff \vec{J}(\pi_1) \succeq_J \vec{J}(\pi_2) \\
&\iff \pi_1 \succeq_{O_{GOMORL}} \pi_2
\end{aligned}$$

$\square$

**Proposition B.12** ($PO \succeq_{EXPR} F$, for any objective specification formalism $F$).

By definition, any policy ordering in any environment is expressible with a Policy Ordering specification ($\succeq_\pi$).

**Proposition B.13** ($FTR \succeq_{EXPR} LAR$). *Any policy ordering expressible with a Limit Average Reward specification can be expressed with a Function from Trajectories to Reals.*

**Intuition:** The limit average reward is a value assigned to a trajectory. A function from trajectories to reals can assign the limit average reward of a particular reward function to each trajectory.

*Proof.* Let $(\mathcal{R})$ be an arbitrary LAR objective specification in an arbitrary environment. Then the following FTR specification $(f_{FTR})$ expresses the same policy ordering:

$$f_{FTR}(\xi) = \lim_{N \to \infty} \left[ \frac{1}{N} \sum_{t=0}^{N-1} \mathcal{R}(s_t, a_t, s_{t+1}) \right]$$

Then:

$$J_{LAR}(\pi) = \lim_{N \to \infty} \left[ \mathbb{E}_{\pi, T, I} \left[ \frac{1}{N} \sum_{t=0}^{N-1} R(s_t, a_t, s_{t+1}) \right] \right]$$

We have bounded rewards, i.e. $\forall (s_t, a_t, s_{t+1}),\ R(s_t, a_t, s_{t+1}) < R_{max}$, where $R_{max}$ is the maximum reward assigned to any transition by the reward function. This means that the average reward converges: $\lim_{N \to \infty} \left[ \frac{1}{N} \sum_{t=0}^{N-1} R(s_t, a_t, s_{t+1}) \right] = X < \infty$. These two facts and Lebesgue's Dominated Convergence Theorem imply that we can move the expectation outside the limit. So:

$$J_{LAR}(\pi) = \mathbb{E}_{\pi,T,I} \left[ \lim_{N \to \infty} \left[ \frac{1}{N} \sum_{t=0}^{N-1} R(s_t, a_t, s_{t+1}) \right] \right]$$

The expression inside the expectation is equal to $f_{FTR}(\xi)$, therefore:

$$J_{LAR}(\pi) = \mathbb{E}_{\pi,T,I} \left[ f_{FTR}(\xi) \right] = J_{FTR}(\pi)$$

$\square$

**Proposition B.14** ($FTR \succeq_{EXPR} LTL$). *Any policy ordering expressible with Linear Temporal Logic can be expressed with Functions from Trajectories to Reals.*

**Intuition:** An LTL formula is a function that assigns the value 0 to trajectories in which the formula is false and the value 1 to trajectories in which the formula is true. This is a special case of a function from trajectories to reals.

*Proof.* Let $(\varphi)$ be an arbitrary LTL objective specification in an arbitrary environment. Then the following FTR specification ($f_{FTR}$) expresses the same policy ordering:

- $f_{FTR}(\xi) := \varphi(\xi)$

(An LTL formula assigns a truth value of 0 or 1 to any given trajectory, so a formula is a function from trajectories to $\{0, 1\}$. This is a special case of a function from trajectories to reals.) $\square$

**Proposition B.15** ($FTR \succeq_{EXPR} RM$). *Any policy ordering expressible with Reward Machines (RM) can be expressed with Functions from Trajectories to Reals (FTR).*

**Intuition:** Reward machines give step-by-step rewards in a trajectory based on aspects of the history so far. The behavior of a reward machine for an entire trajectory is fixed by the states and actions in the trajectory, so the discounted sum of rewards that a reward machine yields is a function from trajectories to reals.

*Proof.* Let $(U, u_0, \delta_U, \delta_{\mathcal{R}}, \gamma)$ be an arbitrary RM objective specification in an arbitrary environment. Then the following FTR specification ($f_{FTR}$) expresses the same policy ordering:

- $f_{FTR}(\xi) := \sum_{t=0}^{\infty} \gamma^t \left( \delta_{\mathcal{R}}(u_t, u_{t+1})(s_t, a_t, s_{t+1}) \right)$

Here, $u_t$ is the state of the specified reward machine at time step $t$. This value is well-defined given $U, u_0, \delta_U$, and $\xi$, because $u_0$ is the starting machine state, and given a machine state at any time $t$, $u_{t+1} = \delta_U(u_t, s_t, a_t, s_{t+1})$. $\xi$ specifies $s_t$ and $a_t$ for all $t$, and given $u_0$ and $\delta_U$, the machine state at any time step can be derived using this rule iteratively starting from $u_0$.

Since $u_t$ is well-defined for all $t$, so is $\delta_{\mathcal{R}}(u_t, u_{t+1})$, and so is $f_{FTR}(\xi)$.

This FTR specification induces the same policy ordering as the RM specification above:

$$\begin{aligned}
J_{FTR}(\pi) &:= \overset{E,\pi}{\underset{\xi}{\mathbb{E}}} \left[ f_{FTR}(\xi) \right] \\
&= \overset{E,\pi,u_0,\delta_U}{\underset{\xi}{\mathbb{E}}} \left[ \sum_{t=0}^{\infty} \gamma^t \left( \delta_{\mathcal{R}}(u_t, u_{t+1})(s_t, a_t, s_{t+1}) \right) \right] \\
&= J_{RM}(\pi)
\end{aligned}$$

Both FTR and RM derive policy orderings directly from the policy evaluation functions, so this means the two specifications induce the same policy ordering. □

**Proposition B.16** ($FTLR \succeq_{EXPR} RRL$). *Any policy ordering expressible with Regularised RL (RRL) can be expressed with Functions from Trajectory Lotteries to Reals (FTLR).*

**Intuition:** Two policies generate the same trajectory lottery if and only if they are identical on all states that either policy ever visits with nonzero probability. If two policies are identical on all states visited with nonzero probability, then an RRL specification must assign them the same value. Therefore, a well-defined function can take a trajectory lottery as input and output the value assigned by an RRL specification to all policies that generate the given trajectory lottery.

*Proof.* Let $(\mathcal{R}, \alpha, F, \gamma)$ be an arbitrary RRL objective specification in an arbitrary environment. Then the following FTLR specification ($f_{FTLR}$) expresses the same policy ordering:

- $f_{FTLR}(L_\pi) := J_{RRL}(\pi) := \mathbb{E}_\xi^{E,\pi} \left[ \sum_{t=0}^{\infty} \gamma^t \left( \mathcal{R}(S_t, A_t, S_{t+1}) - \alpha F[\pi(S_t)] \right) \right]$

Here, $L_\pi$ is the lottery over trajectories that is produced by policy $\pi$ in the environment. Since $J_{FTLR}(\pi) := f_{FTLR}(L_\pi)$, the equality of $J_{FTLR}$ and $J_{RRL}$ on all policies is immediate. However, we must verify that this function from trajectory lotteries to reals is well-defined, i.e. that $L_{\pi_1} = L_{\pi_2} \implies J_{RRL}(\pi_1) = J_{RRL}(\pi_2)$.

Since rewards in Regularised RL are given step-by-step, the policy evaluation function can be rewritten as follows:

$$J_{RRL}(\pi) := \mathbb{E}_\xi^{E,\pi} \left[ \sum_{t=0}^{\infty} \gamma^t \left( \mathcal{R}(S_t, A_t, S_{t+1}) - \alpha F[\pi(S_t)] \right) \right]$$

$$J_{RRL}(\pi) := \sum_{t=0}^{\infty} \sum_{(s,a,s') \in \mathcal{S} \times \mathcal{A} \times \mathcal{S}} \mathbb{P}_\xi^{E,\pi}[S_t = s, A_t = a, S_{t+1} = s'] \left( \gamma^t (\mathcal{R}(s, a, s') - \alpha F[\pi(s)]) \right)$$

**Lemma B.17.** *If $L_{\pi_1} = L_{\pi_2}$, then for all $t$ and for all $(s, a, s') \in \mathcal{S} \times \mathcal{A} \times \mathcal{S}$:*

$$\mathbb{P}_\xi^{E,\pi_1}[S_t = s, A_t = a, S_{t+1} = s'] = \mathbb{P}_\xi^{E,\pi_2}[S_t = s, A_t = a, S_{t+1} = s']$$

*Proof of Lemma.* First recall the definition of a trajectory lottery:

Let $\Xi_k$ be the set of all initial trajectory segments of length $2(k+1)$. We write $[\xi]_k$ for the first $2(k+1)$ elements of $\xi$. Define $L_{k,\pi} \in \Delta(\Xi_k) : L_{k,\pi}(\xi_k = \langle s_0, a_0, ..., s_k, a_k \rangle) = P_{\xi \sim \pi, T, I}([\xi]_k = \xi_k)$. A trajectory lottery $L_\pi$ is then defined as the infinite sequence of $L_{k,\pi}$, $L_\pi := (L_{0,\pi}, L_{1,\pi}, L_{2,\pi}, ...)$.

Now, let $\Xi_{t+1,(s,a,s')}$ be the set of trajectory segments of length $2(t + 2)$ which have $(s, a, s')$ as the most recently completed transition. We can then see that for a given $t$ and $\pi$:

$$\mathbb{P}_\xi^{E,\pi}[S_t = s, A_t = a, S_{t+1} = s'] = \sum_{\xi_{t+1,(s,a,s')} \in \Xi_{t+1,(s,a,s')}} L_{t+1,\pi}(\xi_{t+1,(s,a,s')})$$

Equivalently, in words, the probability that the transition $(s, a, s')$ is taken from time step $t$ to $t + 1$ is equal to the sum of the probabilities of all the trajectory segments of length $t + 2$ that have the transition $(s, a, s')$ from time step $t$ to $t + 1$. (The reason we look at trajectory segments of length $t + 2$ is that time indexing starts at 0, so we need $t + 2$ steps to get to the step indexed by $t + 1$.)

The right-hand side of this equation is fully determined by $L_\pi$, so the left-hand side is also fully determined by $L_\pi$. This completes the proof of Lemma B.17. □

**Corollary B.17.1.** *If $L_{\pi_1} = L_{\pi_2}$, then for all $t$ and for all $s \in S$, $\mathbb{P}_\xi^{E,\pi_1}[S_t = s] = \mathbb{P}_\xi^{E,\pi_2}[S_t = s]$.*

This follows straightforwardly from Lemma B.17 by marginalising over $a$ and $s'$.

**Corollary B.17.2.** *If $L_{\pi_1} = L_{\pi_2}$, then for all $t$ and for all $(s, a) \in S \times A$, $\mathbb{P}_\xi^{E,\pi_1}[S_t = s, A_t = a] = \mathbb{P}_\xi^{E,\pi_2}[S_t = s, A_t = a]$.*

This follows straightforwardly from Lemma B.17 by marginalising over $s'$.

**Corollary B.17.3.** *If $L_{\pi_1} = L_{\pi_2}$, then $\pi_1(s) = \pi_2(s)$ for all states $s$ that either policy ever visits with nonzero probability.*

*Proof of B.17.3.* First, note that if either $\pi_1$ or $\pi_2$ ever visits a state $s^*$ with nonzero probability, then by B.17.1, there exists $t^*$ such that $\mathbb{P}_\xi^{E,\pi_1}[S_{t^*} = s^*] = \mathbb{P}_\xi^{E,\pi_2}[S_{t^*} = s^*] > 0$. If $\pi_1(s^*) \neq \pi_2(s^*)$, there must be an action $a^*$ such that $\pi_1(a^*|s^*) > \pi_2(a^*|s^*)$. But by B.17.2, we know that since $L_{\pi_1} = L_{\pi_2}$, $\mathbb{P}_\xi^{E,\pi_1}[S_t = s^*, A_t = a^*] = \mathbb{P}_\xi^{E,\pi_2}[S_t = s^*, A_t = a^*]$. Thus:

$$\mathbb{P}_\xi^{E,\pi_1}[S_t = s^*, A_t = a^*] = \mathbb{P}_\xi^{E,\pi_2}[S_t = s^*, A_t = a^*]$$
$$\mathbb{P}_\xi^{E,\pi_1}[S_t = s^*]\pi_1(a^*|s^*) = \mathbb{P}_\xi^{E,\pi_2}[S_t = s^*]\pi_2(a^*|s^*)$$
$$\mathbb{P}_\xi^{E,\pi_1}[S_t = s^*]\pi_1(a^*|s^*) = \mathbb{P}_\xi^{E,\pi_1}[S_t = s^*]\pi_2(a^*|s^*) \qquad \text{(by Corollary B.17.1)}$$
$$\pi_1(a^*|s^*) = \pi_2(a^*|s^*)$$

So the two policies must agree on $s^*$ after all. $\qquad\qquad\square$

We require one more lemma to show that $L_{\pi_1} = L_{\pi_2} \implies J_{RRL}(\pi_1) = J_{RRL}(\pi_2)$.

**Lemma B.18.** *If $L_{\pi_1} = L_{\pi_2}$, then for all $t$ and for all $(s, a, s') \in \mathcal{S} \times \mathcal{A} \times \mathcal{S}$:*
$\mathbb{P}_\xi^{E,\pi_1}[S_t = s, A_t = a, S_{t+1} = s'] \left(\gamma^t(\mathcal{R}(s, a, s') - \alpha F[\pi_1(s)])\right) = \mathbb{P}_\xi^{E,\pi_2}[S_t = s, A_t = a, S_{t+1} = s'] \left(\gamma^t(\mathcal{R}(s, a, s') - \alpha F[\pi_2(s)])\right)$ *for any RRL specification $(\mathcal{R}, \alpha, F, \gamma)$.*

*Proof of B.18.* Lemma B.17 states that if $L_{\pi_1} = L_{\pi_2}$, then for all $t$ and for all $(s, a, s') \in \mathcal{S} \times \mathcal{A} \times \mathcal{S}$:

$$\mathbb{P}_\xi^{E,\pi_1}[S_t = s, A_t = a, S_{t+1} = s'] = \mathbb{P}_\xi^{E,\pi_2}[S_t = s, A_t = a, S_{t+1} = s']$$

So it remains to be shown that $\mathbb{P}_\xi^{E,\pi_1}[S_t = s, A_t = a, S_{t+1} = s'] \left(\gamma^t(\mathcal{R}(s, a, s') - \alpha F[\pi_1(s)])\right) = \mathbb{P}_\xi^{E,\pi_1}[S_t = s, A_t = a, S_{t+1} = s'] \left(\gamma^t(\mathcal{R}(s, a, s') - \alpha F[\pi_2(s)])\right)$. Consider the following two cases.
*Case 1:* $\mathbb{P}_\xi^{E,\pi_1}[S_t=s, A_t=a, S_{t+1}=s'] = 0$. In this case, because both sides are equal to 0:

$$\mathbb{P}_\xi^{E,\pi_1}[S_t=s, A_t=a, S_{t+1}=s'] \left(\gamma^t(\mathcal{R}(s, a, s') - \alpha F[\pi_1(s)])\right) = \dots$$
$$\mathbb{P}_\xi^{E,\pi_1}[S_t=s, A_t=a, S_{t+1}=s'] \left(\gamma^t(\mathcal{R}(s, a, s') - \alpha F[\pi_2(s)])\right)$$

*Case 2:* $\mathbb{P}_\xi^{E,\pi_1}[S_t = s, A_t = a, S_{t+1} = s'] > 0$. In this case, $\pi_1(s) = \pi_2(s)$ by Corollary B.17.3. Therefore, $\left(\gamma^t(\mathcal{R}(s, a, s') - \alpha F[\pi_1(s)])\right) = \left(\gamma^t(\mathcal{R}(s, a, s') - \alpha F[\pi_2(s)])\right)$, and by extension:

$$\mathbb{P}_\xi^{E,\pi_1}[S_t = s, A_t = a, S_{t+1} = s'] \left(\gamma^t(\mathcal{R}(s, a, s') - \alpha F[\pi_1(s)])\right) = \dots$$
$$\mathbb{P}_\xi^{E,\pi_1}[S_t = s, A_t = a, S_{t+1} = s'] \left(\gamma^t(\mathcal{R}(s, a, s') - \alpha F[\pi_2(s)])\right)$$

Since these cases are exhaustive, this concludes the proof of the lemma. $\qquad\square$

Finally, recall that:

$$J_{RRL}(\pi) := \sum_{t=0}^{\infty} \sum_{(s,a,s')\in\mathcal{S}\times\mathcal{A}\times\mathcal{S}} \mathbb{P}_\xi^{E,\pi}[S_t = s, A_t = a, S_{t+1} = s'] \left(\gamma^t(\mathcal{R}(s, a, s') - \alpha F[\pi(s)])\right)$$

By Lemma B.18, $L_{\pi_1} = L_{\pi_2} \implies J_{RRL}(\pi_1) = J_{RRL}(\pi_2)$. This means that the FTLR specification given by $f_{FTLR}(L_\pi) := J_{RRL}(\pi)$ is well-defined.

Both FTLR and RRL derive policy orderings directly from the policy evaluation functions, so this means the two specifications induce the same policy ordering. Therefore, we've shown that for any RRL specification in any environment, it is possible to construct an FTLR specification which expresses the same policy ordering. This concludes the proof of the proposition. $\square$

**Proposition B.19** ($FTLR \succeq_{EXPR} OMORL$). *Any policy ordering expressible with Outer Multi-Objective RL (OMORL) can be expressed with Functions from Trajectory Lotteries to Reals (FTLR).*

**Intuition:** Two policies generate the same trajectory lottery if and only if they are identical on all states that either policy ever visits with nonzero probability. If two policies are identical on all states visited with nonzero probability, then an OMORL specification must assign them the same value. Therefore, a well-defined function can take a trajectory lottery as input and output the value assigned by an OMORL specification to all policies that generate the given trajectory lottery.

*Proof.* Let $k, \mathcal{R}, f, \gamma$ be an arbitrary OMORL objective specification in an arbitrary environment. Then the following FTLR specification ($f_{FTLR}$) expresses the same policy ordering:

- $f_{FTLR}(L_\pi) := J_{OMORL}(\pi) := f\left(J_1(\pi), ..., J_k(\pi)\right)$, where
$$J_i(\pi) := \mathbb{E}_\xi \left[ \sum_{t=0}^\infty \gamma^t \mathcal{R}_i(s_t, a_t, s_{t+1}) \right]$$

Here, $L_\pi$ is the lottery over trajectories that is produced by policy $\pi$ in the environment. Since $J_{FTLR}(\pi) := f_{FTLR}(L_\pi)$, the equivalency of $J_{FTLR}$ and $J_{OMORL}$ is immediate: $J_{FTLR}(\pi) := f_{FTLR}(L_\pi) := J_{OMORL}(\pi)$. However, we must verify that this function from trajectory lotteries to reals is well-defined, i.e. that $L_{\pi_1} = L_{\pi_2} \implies J_{OMORL}(\pi_1) = J_{OMORL}(\pi_2)$. A trajectory lottery and an OMORL specification together fix all relevant quantities for computing $J_i(\pi) := \mathbb{E}_\xi \left[ \sum_{t=0}^\infty \gamma^t \mathcal{R}_i(s_t, a_t, s_{t+1}) \right]$ for all $i \in [k]$. Given all the $J_i$ values, the OMORL specification also fixes $f\left(J_1(\pi), ..., J_k(\pi)\right)$. Therefore, a trajectory lottery and an OMORL specification together uniquely specify a value of $J_{OMORL}(\pi) := f\left(J_1(\pi), ..., J_k(\pi)\right)$, and $f_{FTLR}$ above is well-defined.

Both FTLR and OMORL derive policy orderings directly from the policy evaluation functions, so this means the two specifications induce the same policy ordering. Therefore, we've shown that for any OMORL specification in any environment, it is possible to construct an FTLR specification which expresses the same policy ordering. $\square$

**Lemma B.20.** *For any environment E = (S,A,T,I), there exists a reward function $\mathcal{R} : \mathcal{S} \times \mathcal{A} \times \mathcal{S} \to \mathbb{R}$ and discount factor $\gamma \in [0,1)$ such that the trajectory return function $G(\xi) := \sum\limits_{t=0}^{\infty} \gamma^t \mathcal{R}(S_t, A_t, S_{t+1})$ is injective.*

*Proof.* Let $X$ be a finite set with $|X| > 1$ and let $\Xi = X^\omega$ be the set of infinite sequences of members of $X$. Let $\mathcal{R} : X \to \mathbb{R}$ be a reward function and let $\gamma \in [0,1)$ be a discount factor. Define $G_{\mathcal{R}} : \Xi \to \mathbb{R}$ such that $G(\xi) := \sum_{t=0}^{\infty} \gamma^t \mathcal{R}(x_t)$.

Let the reward function $\mathcal{R}$ be any injective function. Since $X$ is finite and $\mathcal{R}$ is injective, there exists a positive real number $M = \max_{x,y \in X}(|\mathcal{R}(x) - \mathcal{R}(y)|)$. Also, since $\mathcal{R}$ is injective, there exists a nonzero positive real number $m = \min_{x,y \in X}(|\mathcal{R}(x) - \mathcal{R}(y)|)$ (for $x \neq y$).

Now let $\xi_1$ and $\xi_2$ be two sequences that differ first at position $s$. Then:

$$G(\xi_1) - G(\xi_2) = \sum_{t=0}^{\infty} \gamma^t (\mathcal{R}(x_t) - \mathcal{R}(y_t)) = \gamma^s \sum_{r=0}^{\infty} \gamma^r (\mathcal{R}(x_r) - \mathcal{R}(y_r))$$

Since $\xi_1$ and $\xi_2$ must differ at position $s$ ($r = 0$), the difference of rewards at that position must be at least $m$. Subsequently, the difference of rewards can be at most $M$. So we can try to compensate for the difference of $m$ at $r = 0$ by subtracting differences of $M$ at positions $r > 0$.

We will never be able to fully compensate if $m > \frac{\gamma M}{1-\gamma} \Rightarrow m(1 - \gamma) > \gamma M \quad (*)$.

Note that since $M > m$, we must have $\gamma < 1/2$.

But also, the maximum difference between reward assignments must be at least $(|X| - 1)$ times the minimum difference, so we have the constraint: $M \geq m(|X| - 1)$. Combined with $(*)$, this entails that $\gamma < 1/|X|$.

Thus, for **any** $\gamma$ satisfying this constraint, simply take an injective reward function $\mathcal{R} : X \to \{0, m, ..., m(|X| - 1)\}$. Then for any two sequences $\xi_1$ and $\xi_2$ that differ anywhere, the difference between their corresponding trajectory returns $G(\xi_1)$ and $G(\xi_2)$ will be nonzero. Thus, we have chosen $\mathcal{R}$ and $\gamma$ such that $G : \Xi \to \mathbb{R}$ is injective (and we can always make it surjective by restricting the codomain to the image of $\Xi$ under $G$). $\square$

Note that this scheme will result in a very small $\gamma$ for large enviroments. Next we will use this lemma to prove that INMR, IMORL, and FTR are equivalent.

**Theorem B.21** ($INMR \sim_{EXPR} IMORL \sim_{EXPR} FTR$ (Theorem 3.1 in Section 3)). *Inner Nonlinear MR (INMR), Inner Multi-Objective RL (IMORL) and Functions from Trajectories to Reals (FTR) can each express every policy ordering on any environment that the other two formalisms can express*

*Proof.* It is trivial that IMORL can express INMR and that FTR can express IMORL and INMR, so if we prove that INMR can express FTR we have proved that the three formalisms are equivelent.

We need to show that, given any FTR objective specification $(f)$, there exists an INMR objective specification $(\mathcal{R}, \gamma, h)$ such that $f = h \circ G_{\mathcal{R}}$.

To show this, we use B.20.

If the state space $S$ and action space $A$ are both finite, then it is always possible to specify a reward function $\mathcal{R} : \mathcal{S} \times \mathcal{A} \times \mathcal{S} \to \mathbb{R}$ and a discount factor $\gamma \in [0,1)$ such that the function $G_{\mathcal{R},\gamma} : \Xi \to \mathbb{R}$ is an injective function.

Hence, choose $(\mathcal{R}, \gamma)$ such that $G_{\mathcal{R},\gamma}(\xi)$ is injective. We can also make it surjective by restricting the codomain $\mathbb{R}$ to the image of $\Xi$ under $G_{\mathcal{R},\gamma}$. Thus, we can make $G_{\mathcal{R},\gamma}$ invertible.

Then, construct $h : \mathbb{R} \to \mathbb{R}$ as: $h = f \circ G_{\mathcal{R},\gamma}^{-1}$. $\square$

As discussed previously this proof requires arbitrarily complex functions and infinite precision to hold in the general case. In many practical situations it would be reasonable to say that FTR$\succ_{EXPR}$ IMORL $\succ_{EXPR}$ INMR but it is still interesting that these formalisms are equivalent in the unrestricted mathematical sense.

**Lemma B.22.** *Let $\vec{J}_{OMORL}(\pi) = \langle J_1(\pi), ..., J_k(\pi) \rangle \in \mathbb{R}^k$ for an OMORL specification $\mathcal{O}_{OMORL} = (k, \mathcal{R}, f, \gamma)$. Then for any environment $E = (\mathcal{S}, \mathcal{A}, \mathcal{T}, \mathcal{I})$ and any FOMR specification $(f_{FOMR}, \gamma_{FOMR})$, there exists an OMORL specification $\mathcal{O}_{OMORL} = (k, \mathcal{R}, f, \gamma)$ such that $\vec{J}_{OMORL}(\pi) = \vec{m}(\pi)$.*

*Proof.* Since $\vec{m}(\pi) \in \mathbb{R}^{|\mathcal{S}||\mathcal{A}||\mathcal{S}|}$ and $\vec{J}_{OMORL}(\pi) \in \mathbb{R}^k$, $k$ must equal $|\mathcal{S}||\mathcal{A}||\mathcal{S}|$ in order to allow $\vec{J}_{OMORL}(\pi) = \vec{m}(\pi)$. Let us define $|\mathcal{S}| \times |\mathcal{A}| \times |\mathcal{S}|$ reward functions indexed by a $s, a, s'$ triple, as follows: $\mathcal{R}_{ijk}(s_l, a_m, s_n) = \delta_{il}\delta_{jm}\delta_{kn}$. Here, $\delta$ is the Kronecker delta function:

$$\delta_{ab} := \begin{cases} 1, & \text{if } a = b, \\ 0, & \text{otherwise.} \end{cases}$$

Essentially, reward function $\mathcal{R}_{ijk}$ provides reward 1 for the transition $(s_i, a_j, s_k)$ and reward 0 for all other transitions. Also, let $\gamma = \gamma_{FOMR}$. Then,

$$\begin{aligned} J_{ijk}(\pi) &= \sum_{l,m,n} \sum_t \gamma^t P(s_t = s_l, a_t = a_m, s_{t+1} = s_n) \mathcal{R}_{ijk}(s_l, a_m, s_n) \\ &= \sum_{l,m,n} \sum_t \gamma_{FOMR}^t P(s_t = s_l, a_t = a_m, s_{t+1} = s_n) \, \delta_{il}\delta_{jm}\delta_{kn} \\ &= \sum_t \gamma_{FOMR}^t P(s_t = s_i, a_t = a_j, s_{t+1} = s_k) \qquad =: \vec{m}(\pi)[s_i, a_j, s_k] \end{aligned}$$

Since all components of $\vec{J}_{OMORL}(\pi)$ and $\vec{m}(\pi)$ match, $\vec{J}_{OMORL}(\pi) = \vec{m}(\pi)$. $\qquad\square$

$S \times A \times S$ could be a very large number of reward functions and therefore in many practical situations FOMR $\succ_{EXPR}$ OMORL and OMO $\succ_{EXPR}$ GOMORL.

**Lemma B.23.** *For any environment $E = (\mathcal{S}, \mathcal{A}, \mathcal{T}, \mathcal{I})$ and discount factor $\gamma$, $\vec{m}(\pi_1) = \vec{m}(\pi_2) \iff L_{\pi_1} = L_{\pi_2}$.*

*Proof.* **Direction 1:** $L_{\pi_1} = L_{\pi_2} \implies \vec{m}(\pi_1) = \vec{m}(\pi_2)$

Consider the set $\Xi_k^* = \{\xi_k \in \Xi_k \mid s_k = s, a_k = a, s_{k+1} = s'\}$. Then,

$$P_\pi(s_k = s, a_k = a, s_{k+1} = s') = \sum_{\xi_k \in \Xi_k^*} L_{k,\pi}(\xi_k)$$

If $L_{\pi_1} = L_{\pi_2}$, then $L_{k,\pi_1} = L_{k,\pi_2} \; \forall k \in \mathbb{N}$. Thus, at every time-step $t$:

$$P_{\pi_1}(s_t = s, a_t = a, s_{t+1} = s') = \sum_{\xi_t \in \Xi_t^*} L_{t,\pi_1}(\xi_t) = \sum_{\xi_t \in \Xi_t^*} L_{t,\pi_2}(\xi_t) = P_{\pi_2}(s_t = s, a_t = a, s_{t+1} = s')$$

Therefore,

$$\vec{m}(\pi_1)[s, a, s'] = \sum_{t=0}^{\infty} \gamma^t P_{\pi_1}(s_t = s, a_t = a, s_{t+1} = s') = \sum_{t=0}^{\infty} \gamma^t P_{\pi_2}(s_t = s, a_t = a, s_{t+1} = s') = \vec{m}(\pi_2)[s, a, s']$$

**Direction 2:** $\vec{m}(\pi_1) = \vec{m}(\pi_2) \implies L_{\pi_1} = L_{\pi_2}$

To prove this direction we will show the contrapositive, i.e. $L_{\pi_1} \neq L_{\pi_2} \implies \vec{m}(\pi_1) \neq \vec{m}(\pi_2)$.

Suppose $L_{\pi_1} \neq L_{\pi_2}$. This means there exists some state $s$ and a subsequent state $s'$ that both policies visit with the same nonzero probability such that $\pi_1(s) \neq \pi_2(s)$. (This might be the initial state, if they diverge right away.) If they were identical on every visited state, they would produce the same trajectory lottery. There must be two actions, $a_1$ and $a_2$, such that $\pi_1(a_1|s) > \pi_2(a_1|s)$ and $\pi_2(a_2|s) > \pi_1(a_2|s)$. For both sets of probabilities to sum to one, $\pi_1$ can't assign a greater probability to all actions at $s$.

Now, considering the transition to $s'$:

$$\frac{\vec{m}(\pi_1)[s,a_1,s']}{\vec{m}(\pi_1)[s,a_2,s']} = \frac{\sum_t \gamma^t P_{\pi_1}(s_t = s, a_t = a_1, s_{t+1} = s')}{\sum_t \gamma^t P_{\pi_1}(s_t = s, a_t = a_2, s_{t+1} = s')}$$

$$= \frac{\sum_t \gamma^t P_{\pi_1}(s_t = s)\pi_1(a_t = a_1|s_t = s)T(s_{t+1} = s'|a_t = a_1, s_t = s)}{\sum_t \gamma^t P_{\pi_1}(s_t = s)\pi_1(a_t = a_2|s_t = s)T(s_{t+1} = s'|a_t = a_2, s_t = s)}$$

Since we're considering stationary policies only, $\pi_1(a_t = a|s_t = s) = \pi_1(a|s)$. The transition function is also time independent. Therefore:

$$\frac{\vec{m}(\pi_1)[s,a_1,s']}{\vec{m}(\pi_1)[s,a_2,s']} = \frac{\pi_1(a_1|s)T(s'|a_1,s)\sum_t \gamma^t P_{\pi_1}(s_t = s)}{\pi_1(a_2|s)T(s'|a_2,s)\sum_t \gamma^t P_{\pi_1}(s_t = s)} = \frac{\pi_1(a_1|s)T(s'|a_1,s)}{\pi_1(a_2|s)T(s'|a_2,s)}$$

Similarly,

$$\frac{\vec{m}(\pi_2)[s,a_1,s']}{\vec{m}(\pi_2)[s,a_2,s']} = \frac{\pi_2(a_1|s)T(s'|a_1,s)}{\pi_2(a_2|s)T(s'|a_2,s)}$$

From the inequalities $\pi_1(a_1|s) > \pi_2(a_1|s)$ and $\pi_2(a_2|s) > \pi_1(a_2|s)$, we deduce:

$$\frac{\pi_1(a_1|s)}{\pi_1(a_2|s)} > \frac{\pi_2(a_1|s)}{\pi_2(a_2|s)} \Rightarrow \frac{\vec{m}(\pi_1)[s,a_1,s']}{\vec{m}(\pi_1)[s,a_2,s']} > \frac{\vec{m}(\pi_2)[s,a_1,s']}{\vec{m}(\pi_2)[s,a_2,s']}$$

Thus, $\vec{m}(\pi_1) \neq \vec{m}(\pi_2)$.

$\square$

Note that this proof requires stationary policies; on non-stationary policies, the above will not necessarily hold.

Theorems B.24 and B.25 follow fairly straightforwardly from Lemmas B.22 and B.23.

**Theorem B.24** (OMORL $\sim_{EXPR}$ FOMR $\sim_{EXPR}$ FTLR (Theorem 3.3 in Section 3)). *Outer Multi-Objective RL (OMORL), Functions from Occupancy Measures to Reals (FOMR), and Functions from Trajectory Lotteries to Reals (FTLR) can each express every stationary policy ordering on any environment that the other two formalisms can express.*

*Proof.* From lemma B.22 we can choose the OMORL reward functions $\mathcal{R}_{ijk}$ such that the associated Markovian policy evaluation functions $J_{ijk}$ form the components of the policy's occupancy measure. If we apply the same function $f$ to both the vector of $J_{ijk}$ functions and the occupancy measure we get that $J_{OMORL}(\pi) = f(\vec{J}_{OMORL}(\pi)) = f(\vec{m}(\pi)) = J_{FOMR}(\pi)$. From lemma B.23 we have that occupancy measures and trajectory lotteries uniquely determine each other and as such we can choose corresponding functions $f_{FOMR}$ and $f_{FTLR}$ such that both FOMR and FTLR induce the same policy orderings. We have now proven that OMORL, FOMR, and FTLR are equally expressive. $\square$

**Theorem B.25** (GOMORL $\sim_{EXPR}$ OMO $\sim_{EXPR}$ TLO (Theorem 3.4 in Section 3)). *Generalised Outer Multi-Objective RL (GOMORL), Occupancy Measures Orderings (OMO), and Trajectory Lottery Orderings (TLO) can each express every stationary policy ordering on any environment that the other two formalisms can express.*

*Proof.* A GOMORL objective specification consists of an ordering on the set of policy evaluation functions $\{\vec{J}_{OMORL}(\pi)|\pi \in \Pi_E\}$, while an OMO objective specification consists of an ordering on the occupation measures $\{\vec{m}(\pi)|\pi \in \Pi_E\}$. As in Theorem B.24, Lemma B.22 says that we can choose our reward functions $\mathcal{R}_{ijk}$ such that $\vec{J}_{OMORL}(\pi) = \vec{m}(\pi), \forall \pi \in \Pi_E$. Naturally, if we apply the same ordering to both the policy evaluation vectors and to the occupancy measures, this then results in the same induced policy ordering. From Lemma B.23 we have that occupancy measures and trajectory lotteries uniquely determine each other, so any ordering of one uniquely determines an ordering of the other. Thus, orderings over occupation measures can induce all and only those policy orderings that can be induced by orderings over trajectory lotteries. Therefore, GOMORL, OMO, and TLO are equally expressive. □

**Proposition B.26** ($RM, MR \not\succeq_{EXPR} LAR, LTL$). *There is an environment and an ordering over policies in that environment that $LAR$ and $LTL$ can induce, but $RM$ and $MR$ cannot.*

*Proof by construction.* Note that since reward machines are strictly more expressive than Markov Rewards ($RM \succeq_{EXPR} MR$;), we only need to show that $RM \not\succeq_{EXPR} LAR$ and $RM \not\succeq_{EXPR} LTL$. However, the reason $RM$ and $MR$ cannot express $LTL$ and $LAR$ is the same: $LAR$ and $LTL$ can give $J^E_{\mathcal{O}}$ functions that are discontinuous in the policy space, but $RM$ and $MR$ cannot. The proof will proceed in three stages. First, we construct the environment $E$ and a subset of policy space $\Pi' \subset \Pi$. Second, Lemma B.27 shows that $LAR$ and $LTL$ can achieve a particular ordering over $\Pi'$. Third, Lemma B.28 demonstrates that $MR$ and $RM$ cannot achieve that policy ordering.

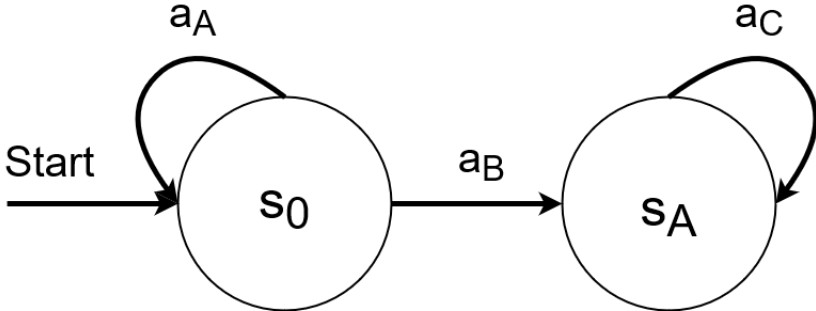

Figure 3: A simple environment consisting of two states $s_0$ and $s_A$ and three actions $a_A$ which leads from $s_0$ to itself ,$a_B$ which leads from $s_0$ to $s_A$ and $a_C$ which leads from $s_A$ to itself. The starting state is $s_0$

**Lemma B.27.** *Consider the following ordering over $\Pi'$ : $\pi_A \succ \pi_B \sim \pi_\alpha \quad \forall \alpha \ 0 < \alpha < 1$ Where:*

- *$\pi_A$ is the policy which takes action $a_A$ in state $s_0$*

- *$\pi_B$ is the policy which takes action $a_B$ in state $s_0$*

- *$\pi_\alpha$ is a policy which takes action $a_B$ with probability $\alpha$ and action $a_A$ with probability $1 - \alpha$*

*There exists an $LAR$ specification and an $LTL$ specification that each induces this ordering over $\Pi'$. In particular, they do so by setting $J^E_{\mathcal{O}}(\pi_\alpha) = \mathbb{1}[\alpha = 0]$.*

- *(A) $\mathcal{O}_{LAR} := (\mathcal{R}[LAR])$, where $\mathcal{R}[LAR](s, a, s') = \mathbb{1}[s = s_0]$, gives expected return $J^E_{\mathcal{O}_{LAR}}(\pi_\alpha) = \mathbb{1}[\alpha = 0]$. Therefore, $\succeq^E_{\mathcal{O}_{LAR}}$ induces the ordering.*

- *(B) $\mathcal{O}_{LTL} := (\psi)$, where $\psi(\xi) = \neg\Diamond s_A$, gives expected return $J^E_{\mathcal{O}_{LAR}}(\pi_\alpha) = \mathbb{1}[\alpha = 0]$. Therefore, $\succeq^E_{\mathcal{O}_{LAR}}$ induces the ordering.*

See lemma B.27 for proof of this. Intuitively, both $LAR$ and $LTL$ allow a discontinuity in $J^E_{\mathcal{O}}(\pi_\alpha)$ at $\alpha = 0$, allowing $\pi_A \succ \pi_\alpha$ for arbitrarily small, but nonzero, values of $\alpha$. Any $\alpha > 0$ gives *some* probability of transitioning to $s_A$ at each time-step, and therefore a certainty of transitioning and getting stuck at some point in an infinite trajectory. For $LAR$, the finite sequence of $\mathcal{R}[LAR](s_0, a_A, s_0) = 1$ rewards will be dominated by the infinite sequence of $\mathcal{R}[LAR](s_A, a_C, s_A) = 0$ rewards. For $LTL$, $\Diamond s_A$ will be true with probability 1, and therefore $\neg\Diamond s_A$ will be true with probability 0. If $\alpha = 0$, the agent will always loop at $s_0$ indefinitely, getting a consistent reward of 1 in $LAR$. Similarly, since there is 0 probability of entering $s_A$, $\neg\Diamond s_A$ is true.

To finish the counter-example, it remains to show that the ordering over $\Pi'$ cannot be captured by *any* objective specification $\mathcal{O}_{MR}$ or $\mathcal{O}_{RM}$. To do this, we first show that $J^E_{\mathcal{O}_{MR}}(\pi_\alpha)$ and $J^E_{\mathcal{O}_{RM}}(\pi_\alpha)$ must be continuous at $\alpha = 0$.

**Lemma B.28.** *For all $\mathcal{O}_{RM}$ and $\mathcal{O}_{MR}$, $\lim_{\alpha \to 0^+} J_{\mathcal{O}}^E(\pi_\alpha) = J_{\mathcal{O}}^E(\pi_A)$*

For a proof of this lemma, see appendix B.3.2. Intuitively, for any finite-time horizon, arbitrarily small values $\alpha$ lead to small probabilities of entering $s_A$. Since the infinite-horizon return for $MR$ and $RM$ is closely approximated by a finite-time horizon return, arbitrarily small values of $\alpha$ should lead to small changes to the expected return.

**Corollary B.28.1.** *For all $\mathcal{O}_{RM}$ and $\mathcal{O}_{MR}$, if $\forall \alpha \in (0, 1)$, $\pi_B \sim_{\mathcal{O}}^E \pi_\alpha$, then $\pi_B \sim \pi_A$.*

This corollary follows from the fact that $\pi_B \sim \pi_\alpha$ implies that $J_{\mathcal{O}}^E(\pi_\alpha) = J_{\mathcal{O}}^E(\pi_B)$ and therefore the limit is given by: $\lim_{\alpha \to 0^+} J_{\mathcal{O}}^E(\pi_\alpha) = J_{\mathcal{O}}^E(\pi_B)$. Since lemma B.28 also shows that $\lim_{\alpha \to 0^+} J_{\mathcal{O}}^E(\pi_\alpha) = J_{\mathcal{O}}^E(\pi_A)$, we have that $J_{\mathcal{O}}^E(\pi_A) = J_{\mathcal{O}}^E(\pi_B)$ for any $\mathcal{O}_{MR}$ or $\mathcal{O}_{RM}$.

Combining *lemma B.27* and *lemma B.28* tells us that, no $MR$ and $RM$ cannot induce an ordering with $\pi_A \succ \pi_B$, and $\pi_\alpha \sim \pi_B$, even though $LAR$ and $LTL$ can. Therefore, for this particular $E$, there is an ordering over $\Pi'$ that $LAR$ and $LTL$ can induce but $MR$ and $RM$ cannot. This is sufficient to show that $RM \not\succeq_{EXPR} LAR$, $RM \not\succeq_{EXPR} LTL$, $MR \not\succeq_{EXPR} LAR$, and $MR \not\succeq_{EXPR} LTL$.

$\square$

### B.3.1 PROOF OF LEMMA B.27

*Proof of lemma B.27 (A).* We will show that $J_{\mathcal{O}_{LAR}}^E(\pi_\alpha) = \mathbb{1}[\alpha = 0]$. First recall that $\mathcal{R}[LAR](s, a, s') \, s_0 := \mathbb{1}[s_t = s_0 \, s_0]$. Next, recall the definition of $J_{E, \mathcal{O}_{LAR}}$:

$$J_{E, \mathcal{O}_{LAR}}(\pi_\alpha) = \lim_{N \to \infty} \frac{1}{N} \mathop{\mathbb{E}}_{\xi}^{E, \pi} \left[ \sum_{t=0}^{N-1} \mathcal{R}[LAR](s_t, a_t, s_{t+1}) \right] \qquad \text{(by definition)}$$

$$= \lim_{N \to \infty} \frac{1}{N} \sum_{t=0}^{N-1} \mathop{\mathbb{E}}_{\xi}^{E, \pi} \left[ \mathbb{1}(s_t = s_0) \right] \qquad \text{(by linearity of expectation)}$$

$$= \lim_{N \to \infty} \frac{1}{N} \sum_{t=0}^{N-1} \mathop{\mathbb{P}}_{\xi}^{E, \pi} \left[ s_t = s_0 \right] \qquad \text{(Since } \mathbb{E}[\mathbb{1}(X)] = \mathbb{P}[X])$$

We consider two cases. First, if $\alpha = 0$, then $\mathbb{P}_{\xi}^{E, \pi} \left[ s_t = s_0 \right] = 1$ for all $t$. Therefore $J_{E, \mathcal{O}_{LAR}}(\pi_A) = \lim_{N \to \infty} \frac{1}{N} \sum_{t=0}^{N-1} 1 = 1$. Second, if $\alpha > 0$, then $\mathbb{P}_{\xi}^{E, \pi} \left[ s_t = s_0 \right] = (1-\alpha)^t \sum_{t=0}^{N-1} (1-\alpha)^t$ is the finite sum of a geometric series with $a = 1$ and $r = (1-\alpha) < 1$:

$$J_{E, \mathcal{O}_{LAR}}(\pi_\alpha) = \lim_{N \to \infty} \frac{1}{N} \sum_{t=0}^{N-1} (1-\alpha)^t = \lim_{N \to \infty} \frac{1}{N} \frac{1 - (1-\alpha)^N}{1 - (1-\alpha)} = \frac{1}{\alpha} \lim_{N \to \infty} \frac{1}{N} - \frac{(1-\alpha)^N}{N} = 0$$

$\square$

*Proof of lemma B.27 (B).* We will show that $J_{\mathcal{O}_{LTL}}^E(\pi_\alpha) = \mathbb{1}[\alpha = 0]$. First define $\mathcal{O}_{LTL} = (\psi)$ with $\psi(\xi) := \neg \Diamond s_A$. Next, recall the definition of $J_{\mathcal{O}_{LTL}}^E$:

$$J_{\mathcal{O}_{LTL}}^E(\pi_\alpha) = \mathop{\mathbb{E}}_{\xi}^{E, \pi} \left[ \psi(\xi) \right] \qquad \text{(by definition)}$$

$$= \mathop{\mathbb{P}}_{\xi}^{E, \pi} \left[ \neg \Diamond s_A \right]$$

$$= 1 - \mathop{\mathbb{P}}_{\xi}^{E, \pi} \left[ \Diamond s_A \right]$$

We consider two cases. First, if $\alpha = 0$, then $\mathbb{P}_\xi^{E,\pi}\left[\lozenge s_A\right] = 0$ since $a_B$ is never chosen. Second, if $\alpha > 0$, then $\mathbb{P}_\xi^{E,\pi}\left[\lozenge s_A\right] = 1$, since $a_B$ will eventually be chosen. $\qquad\square$

### B.3.2 Proof of lemma B.28

*Proof of lemma B.28.* We will show that

$$\lim_{\alpha \to 0^+} J_{\mathcal{O}}^E(\pi_\alpha)) = J_{\mathcal{O}}^E(\pi_0)$$

for all $\mathcal{O}_{RM}$ and $\mathcal{O}_{MR}$. Since $MR$ is a special case of $RM$ where $\mid U \mid = 1$, to prove the lemma, it will suffice to show that $\lim_{\alpha \to 0^+}(\mid J_{\mathcal{O}_{RM}}^E(\pi_\alpha) - J_{\mathcal{O}_{RM}}^E(\pi_A) \mid) = 0$. Again, recall the definition of a reward machine specification, $O_{RM} = (U, u_0, \delta_U, \delta_{\mathcal{R}}, \gamma)$ with return given by:

$$J_{\mathcal{O}_{RM}}^E(\pi) = \mathop{\mathbb{E}}_{\xi}^{E,\pi} \left[ \sum_{t=0}^{\infty} \gamma^t \mathcal{R}_t(s_t, a_t, s_{t+1}) \right]$$

Here $\mathcal{R}_t = \delta_{\mathcal{R}}(u_t, u_{t+1})$ and $u_{t+1} := \delta_u(u_t, s_t)$. Noting that $\mathcal{R}_t(s_t, a_t, s_{t+1})$ depends only on $s_t$, $a_t$, $s_{t+1}$ and $u_t$, we can rewrite it as $\mathcal{R}'(s_t, a_t, s_{t+1}, u_t) := \delta_{\mathcal{R}}(u_t, \delta_u(u_t, s_t))(s_t, a_t, s_{t+1})$. Furthermore, because we have bounded rewards the series is absolutely convergent meaning we can, we can rewrite the whole expression:

$$J_{\mathcal{O}_{RM}}^E(\pi) = \sum_{t=0}^{\infty} \gamma^t \mathop{\mathbb{E}}_{\xi}^{E,\pi} [\mathcal{R}'(s_t, a_t, s_{t+1}, u_t)]$$

We can then rewrite the expression in terms of the difference between expected rewards at a given timestep:

$$J_{\mathcal{O}_{RM}}^E(\pi_\alpha) - J_{\mathcal{O}_{RM}}^E(\pi_A) = \sum_{t=0}^{\infty} \gamma^t \left( \mathop{\mathbb{E}}_{\xi}^{E,\pi} [\mathcal{R}'(s_t, a_t, s_{t+1}, u_t)] - \mathop{\mathbb{E}}_{\xi}^{E,\pi} [\mathcal{R}'(s_t, a_t, s_{t+1}, u_t)] \right)$$

Label this difference $\delta_t := \mathbb{E}_{\xi}^{E,\pi} [\mathcal{R}'(s_t, a_t, s_{t+1}, u_t)] - \mathbb{E}_{\xi}^{E,\pi} [\mathcal{R}'(s_t, a_t, s_{t+1}, u_t)]$. We will find a particular expression for $\delta_t$. First, note that there are a finite number of transitions $(s, a, s')$. This allows us to marginalise the expectation for the reward at a particular timestep:

$$\mathop{\mathbb{E}}_{\xi}^{E,\pi} [\mathcal{R}'(s_t, a_t, s_{t+1}, u_t)] = \sum_{s,a,s'} p_{s,a,s'}^{t,\pi} r_{s,a,s'}^{t,\pi}$$

Where:

$$p_{s,a,s'}^{t,\pi} := \mathop{\mathbb{P}}_{\xi}^{E,\pi} [(s_t, a_t, s_{t+1}) = (s, a, s')]$$

$$r_{s,a,s'}^{t,\pi} := \mathop{\mathbb{E}}_{\xi}^{E,\pi} [\mathcal{R}'(s, a, s', u_t) \mid (s_t, a_t, s_{t+1}) = (s, a, s')]$$

Now, consider our particular $E$ and $\pi_\alpha$ for $\alpha \in [0, 1]$. At any timestep, there are only three possible transitions: $(s_0, a_A, s_0)$, $(s_0, a_B, s_A)$, and $(s_A, a_B, s_A)$. We can, therefore write out a case-by-case expression for $p_{s,a,s'}^{t,\pi_\alpha}$:

$$p_{s_0,a_A,s_0}^{t,\pi_\alpha} = (1 - \alpha)^{t+1}$$
$$p_{s_0,a_B,s_A}^{t,\pi_\alpha} = \alpha(1 - \alpha)^t$$
$$p_{s_0,a_B,s_A}^{t,\pi_\alpha} = 1 - (1 - \alpha)^t$$
$$p_{s,a,s'}^{t,\pi_\alpha} = 0 \text{ otherwise}$$

Noting that $p_{s_0,a_A,s_0}^{t,\pi_A} = 1$, we can write the difference between the expected rewards between $\pi_\alpha$ and $\pi_A$ at a given timestep as:

$$\delta_t = -(1 - p_{s_0,a_A,s_0}^{t,\pi_\alpha})r_{s_0,a_A,s_0}^{t,\pi} + p_{s_0,a_B,s_A}^{t,\pi_\alpha} r_{s_0,a_B,s_A}^{t,\pi_\alpha} + p_{s_0,a_B,s_A}^{t,\pi_\alpha} r_{s_0,a_B,s_A}^{t,\pi_\alpha} \tag{1}$$

Note that $\mathcal{R}'(s_t, a_t, s_{t+1}, u_t)$ is bounded, since $\mathcal{S}$, $\mathcal{A}$ and $U$ are finite. Therefore, there exists some $c$ such that $|\mathcal{R}'(s_t, a_t, s_{t+1}, u_t)| \leq c$. It follows that:

$$
\begin{aligned}
\mid \delta_t \mid &= \left| p_{s_0,a_B,s_A}^{t,\pi_\alpha} r_{s_0,a_B,s_A}^{t,\pi_\alpha} + p_{s_0,a_B,s_A}^{t,\pi_\alpha} r_{s_0,a_B,s_A}^{t,\pi_\alpha} - (1 - p_{s_0,a_A,s_0}^{t,\pi_\alpha})r_{s_0,a_A,s_0}^{t,\pi} \right| \\
&= p_{s_0,a_B,s_A}^{t,\pi_\alpha} \left| r_{s_0,a_B,s_A}^{t,\pi_\alpha} \right| + p_{s_0,a_B,s_A}^{t,\pi_\alpha} \left| r_{s_0,a_B,s_A}^{t,\pi_\alpha} \right| + (1 - p_{s_0,a_A,s_0}^{t,\pi_\alpha}) \left| r_{s_0,a_A,s_0}^{t,\pi} \right| \\
&\leq c \cdot \left( p_{s_0,a_B,s_A}^{t,\pi_\alpha} + p_{s_0,a_B,s_A}^{t,\pi_\alpha} + (1 - p_{s_0,a_A,s_0}^{t,\pi_\alpha}) \right) \\
&= c \cdot \left( 1 - (1-\alpha)^t + \alpha(1-\alpha)^t + (1 - (1-\alpha)^{t+1}) \right) \\
&= 2c \cdot \left( 1 - (1-\alpha)^t + \alpha(1-\alpha)^t \right)
\end{aligned}
$$

We can use this to bound the difference in $J_{\mathcal{O}}^E$ values.

$$
\begin{aligned}
\mid J_{\mathcal{O}_{RM}}^E(\pi_\alpha) - J_{\mathcal{O}_{RM}}^E(\pi_A) \mid &= \left| \sum_{t=0}^{\infty} \gamma^t \delta_t \right| \\
&\leq \sum_{t=0}^{\infty} \gamma^t \mid \delta_t \mid \\
&\leq \sum_{t=0}^{\infty} \gamma^t 2c \cdot \left( 1 - (1-\alpha)^t + \alpha(1-\alpha)^t \right) \\
&= 2c \left( \sum_{t=0}^{\infty} \gamma^t - \sum_{t=0}^{\infty} \gamma^t \cdot (1-\alpha)^t + \alpha \sum_{t=0}^{\infty} \gamma^t \cdot (1-\alpha)^t \right) \\
&= 2c \left( \frac{1}{1-\gamma} - \frac{1}{1-\gamma(1-\alpha)} + (\alpha)\frac{1}{1-\gamma(1-\alpha)} \right) \\
&:= Z
\end{aligned}
$$

If we have that $\mid J_{\mathcal{O}_{RM}}^E(\pi_\alpha) - J_{\mathcal{O}_{RM}}^E(\pi_A) \mid \leq Z$, then to show that $\lim_{\alpha \to 0^+}(J_{\mathcal{O}_{RM}}^E(\pi_\alpha)) = J_{\mathcal{O}_{RM}}^E(\pi_A)$ it suffices to show that $\lim_{\alpha \to 0^+}(Z) = 0$.

$$
\begin{aligned}
\lim_{\alpha \to 0^+}(Z) &= \lim_{\alpha \to 0^+} 2c \left( \frac{1}{1-\gamma} - \frac{1}{1-\gamma(1-\alpha)} + \frac{\alpha}{1-\gamma(1-\alpha)} \right) \\
&= 2c \left( \frac{1}{1-\gamma} - \lim_{\alpha \to 0^+} \frac{1}{1-\gamma(1-\alpha)} + \lim_{\alpha \to 0^+} \frac{\alpha}{1-\gamma(1-\alpha)} \right) \\
&= 2c \left( \frac{1}{1-\gamma} - \frac{1}{1-\gamma} + 0 \right) \\
&= 0
\end{aligned}
$$

$\square$

**Proposition B.29** ($LAR \not\preceq_{EXPR} MR, LTL$). *There is an environment and an ordering over policies in that environment that Markov Rewards (MR) and Linear Temporal Logic (LTL) can induce, but Limit Average Reward (LAR) cannot.*

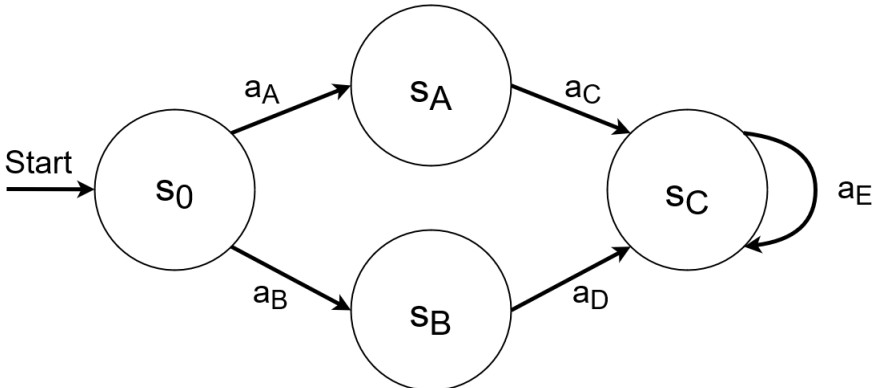

Figure 4: An environment consisting of four states $s_0, s_A, s_B, s_C$ and five actions $a_A, a_B, a_C, a_D, a_E$. The starting state is $s_0$.

*Proof by construction.* In the environment above there are two possible deterministic policies, $\pi_u$ taking the upper path through $s_A$ and $\pi_l$ taking the lower path through $s_B$. We argue that LAR cannot express the policy ordering

$$\pi_u \succ \pi_l$$

while MR and LTL can. First we will argue that LAR cannot express this policy ordering: Both policies take action $a_E$ an infinite number of times while they only take the other actions at most once.

$$
\begin{aligned}
J^E_{\mathcal{O}_{LAR}}(\pi_u) &= \lim_{N \to \infty} \left[ \frac{1}{N} \sum_{t=0}^{N-1} \mathcal{R}(s_t, a_t, s_{t+1}) \right] \\
&= \lim_{N \to \infty} \left[ \frac{1}{N} \left( \mathcal{R}(s_0, a_A, s_A) + \mathcal{R}(s_A, a_C, s_C) + \sum_{t=2}^{N-1} \mathcal{R}(s_C, a_E, s_C) \right) \right] \\
&= \mathcal{R}(s_C, a_E, s_C) \\
J^E_{\mathcal{O}_{LAR}}(\pi_l) &= \lim_{N \to \infty} \left[ \frac{1}{N} \sum_{t=0}^{N-1} \mathcal{R}(s_t, a_t, s_{t+1}) \right] \\
&= \lim_{N \to \infty} \left[ \frac{1}{N} \left( \mathcal{R}(s_0, a_B, s_B) + \mathcal{R}(s_B, a_D, s_B) + \sum_{t=2}^{N-1} \mathcal{R}(s_C, a_E, s_C) \right) \right] \\
&= \mathcal{R}(s_C, a_E, s_C)
\end{aligned}
$$

Meaning that $J^E_{\mathcal{O}_{LAR}}(\pi_u) = J^E_{\mathcal{O}_{LAR}}(\pi_l)$ and so $\pi_u \sim \pi_l$. Next we will show that MR can express this policy ordering: let $\mathcal{R}(s_0, a_A, s_A) := 1$ and all other rewards $:= 0$. Then:

$$
\begin{aligned}
J^E_{\mathcal{O}_{MR}}(\pi_u) &= 1 \\
J^E_{\mathcal{O}_{MR}}(\pi_l) &= 0
\end{aligned}
$$

resulting in our desired ordering.

Next we will show that LTL can express this policy ordering: Consider the LTL predicate $\Diamond s_A$ i.e. finally $s_A$ this will give reward 1 to all trajectories which include $s_A$ and 0 otherwise. This gives:

$$
\begin{aligned}
J^E_{\mathcal{O}_{LTL}}(\pi_u) &= 1 \\
J^E_{\mathcal{O}_{LTL}}(\pi_l) &= 0
\end{aligned}
$$

i.e. our desired ordering. □

**Proposition B.30** ($LTL \not\succeq_{EXPR} MR, LAR$). *There is an environment and an ordering over policies in that environment that Markov Rewards (MR) and Limit Average Reward (LAR) can induce, but Linear Temporal Logic (LTL) cannot.*

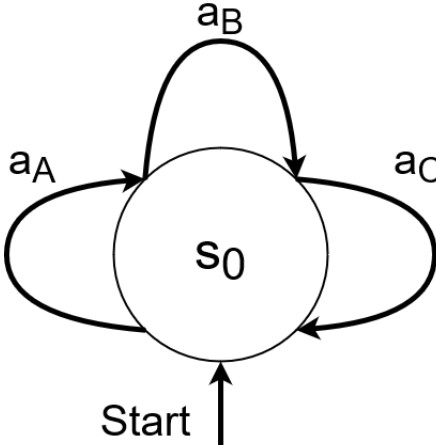

Figure 5: An environment with a single state $s_0$ with three actions $a_A, a_B$ and $a_C$ which all lead back to itself.

*Proof by construction.* Consider the deterministic policies, $\pi_A$, $\pi_B$ and $\pi_C$ corresponding to taking actions $a_A, a_B$ or $a_C$ respectively.

We argue that LTL cannot express the policy ordering:

$$\pi_A \succ \pi_B \succ \pi_C$$

while MR and LAR can.

First we will show that LTL cannot express this policy ordering:

$$\pi_A \succ \pi_B \implies \varphi\left(\xi_A\right) > \varphi\left(\xi_B\right)$$
$$\implies \varphi\left(\xi_A\right) = 1 \text{ and } \varphi\left(\xi_B\right) = 0 \qquad \text{(as deterministic environment)}$$

however,

$$\pi_B \succ \pi_C \implies \varphi\left(\xi_B\right) > \varphi\left(\xi_C\right)$$
$$\implies \varphi\left(\xi_B\right) = 1 \text{ and } \varphi\left(\xi_C\right) = 0 \qquad \text{(as deterministic environment)}$$

leading to a contradiction. For deterministic policies on deterministic environments LTL can only divide policies into two categories and as such cannot order 3 or more policies.

MR can clearly express this ordering by setting $\mathcal{R}(s_0, a_A, s_0) := 1$, $\mathcal{R}(s_0, a_B, s_0) := 0$ and $\mathcal{R}(s_0, a_B, s_0) := -1$ leading to:

$$J^E_{\mathcal{O}_{MR}}(\pi_A) = \frac{1}{1-\gamma}$$
$$J^E_{\mathcal{O}_{MR}}(\pi_B) = 0$$
$$J^E_{\mathcal{O}_{MR}}(\pi_C) = \frac{-1}{1-\gamma}$$

LAR can also clearly express this ordering by again setting $\mathcal{R}(s_0, a_A, s_0) := 1$, $\mathcal{R}(s_0, a_B, s_0) := 0$ and $\mathcal{R}(s_0, a_B, s_0) := -1$ which leads to

$$J^E_{\mathcal{O}_{LAR}}(\pi_A) = 1$$
$$J^E_{\mathcal{O}_{LAR}}(\pi_B) = 0$$
$$J^E_{\mathcal{O}_{LAR}}(\pi_C) = -1$$

$\square$

**Proposition B.31** ($MR, RRL \not\succeq_{EXPR} RM, ONMR, LTL$). *There is an environment and an ordering over policies in that environment that Reward Machines (RM), Outer Nonlinear MR (ONMR) and Linear Temporal Logic (LTL) can induce, but Markov Rewards (MR) and Regularised RL (RRL) cannot.*

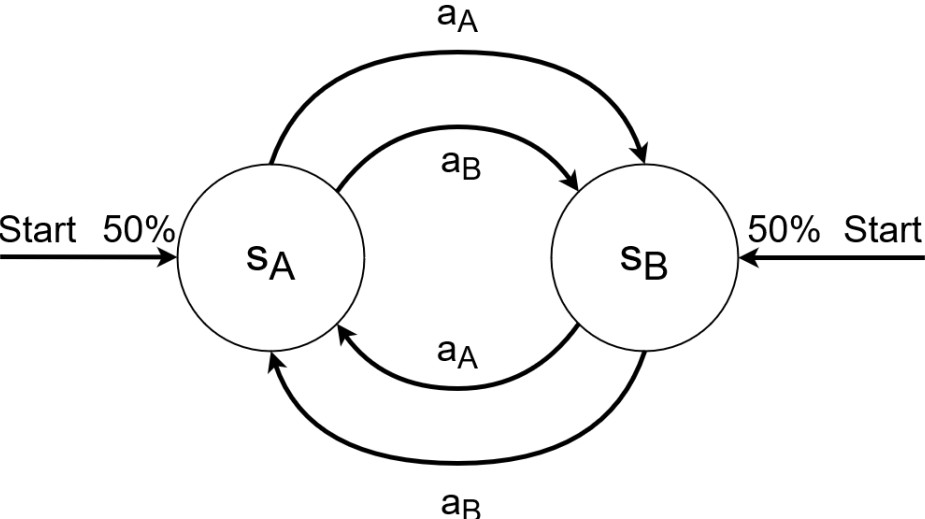

Figure 6: A two state environment with 2 actions $a_A$ and $a_B$ going from each state to the other state. The initial state is random, i.e. $s_A$ and $s_B$ with equal probability.

*Proof by construction.* The figure above shows a simple 2-state system where 2 different actions $a_A$ and $a_B$ are possible in each state. Let $\pi_{ij}$ denote the policy which takes action $i$ in state $s_A$ and action $j$ in state $s_B$. There are 4 possible trajectories following deterministic stationary policies, namely, $(s_A a_A s_B a_B)^*$, $(s_A a_B s_B a_A)^*$, $(s_A a_A s_B a_A)^*$ and $(s_A a_B s_B a_B)^*$ which correspond to policy $\pi_{AB}$, $\pi_{BA}$, $\pi_{AA}$ and $\pi_{BB}$ respectively.

We claim that MR and RRL cannot implement the XOR function, i.e. the policy ordering $\pi_{AB} \sim \pi_{BA} \succ \pi_{AA} \sim \pi_{BB}$ while RM, ONMR and LTL can.

First we will show that MR cannot express this policy ordering:

We want to describe the ordering $J^E_{\mathcal{O}_{MR}}(\pi_{AB}) = J^E_{\mathcal{O}_{MR}}(\pi_{BA}) > J^E_{\mathcal{O}_{MR}}(\pi_{AA}) = J^E_{\mathcal{O}_{MR}}(\pi_{BB})$. Each trajectory only has two unique state-action pairs, and since the initial state is random, in expectation each action a policy contains will be taken 50% of the time at each time step.

This means that:

$$J^E_{\mathcal{O}_{MR}}(\pi_{AB}) > J^E_{\mathcal{O}_{MR}}(\pi_{AA}) \implies$$
$$\frac{1}{2}\sum_{t=0}^{\infty}\gamma^t \mathcal{R}(s_A, a_A, s_B) + \frac{1}{2}\sum_{t=0}^{\infty}\gamma^t \mathcal{R}(s_B, a_B, s_A) >$$
$$\frac{1}{2}\sum_{t=0}^{\infty}\gamma^t \mathcal{R}(s_A, a_A, s_B) + \frac{1}{2}\sum_{t=0}^{\infty}\gamma^t \mathcal{R}(s_B, a_A, s_A) \implies$$
$$\sum_{t=0}^{\infty}\gamma^t \mathcal{R}(s_B, a_B, s_A) > \sum_{t=0}^{\infty}\gamma^t \mathcal{R}(s_B, a_A, s_A) \implies$$
$$\mathcal{R}(s_B, a_B, s_A) > \mathcal{R}(s_B, a_A, s_A).$$

However,

$$J^E_{\mathcal{O}_{MR}}(\pi_{BA}) > J^E_{\mathcal{O}_{MR}}(\pi_{BB}) \implies$$

$$\frac{1}{2}\sum_{t=0}^{\infty}\gamma^t\mathcal{R}(s_A, a_B, s_B) + \frac{1}{2}\sum_{t=0}^{\infty}\gamma^t\mathcal{R}(s_B, a_A, s_A) >$$

$$\frac{1}{2}\sum_{t=0}^{\infty}\gamma^t\mathcal{R}(s_A, a_B, s_B) + \frac{1}{2}\sum_{t=0}^{\infty}\gamma^t\mathcal{R}(s_B, a_B, s_A) \implies$$

$$\sum_{t=0}^{\infty}\gamma^t\mathcal{R}(s_B, a_A, s_A) > \sum_{t=0}^{\infty}\gamma^t\mathcal{R}(s_B, a_B, s_A) \implies$$

$$\mathcal{R}(s_B, a_A, s_A) > \mathcal{R}(s_B, a_B, s_A),$$

in contradiction with the previous relation.

Next we will show that Regularised RL cannot express this policy ordering either. Recall the policy evaluation function in Regularised RL:

$$J^E_{\mathcal{O}_{RRL}}(\pi) = \mathbb{E}_{\xi \sim \pi, T, I}[\sum_{t=0}^{\infty}\gamma^t(\mathcal{R}(s_t, a_t) + \alpha F[\pi(s_t)])]$$

In the case above, maintaining full generality we can write $F[\pi(s_t)] = f(P(a_A|s_t), P(a_B|s_t))$, for arbitrary $f : \Delta(A) \rightarrow \mathbb{R}$. Thus, for the four deterministic policies considered above, we have:

$$F[\pi_{AB}(s_A)] = f(1, 0);\ F[\pi_{AB}(s_B)] = f(0, 1)$$
$$F[\pi_{BA}(s_A)] = f(0, 1);\ F[\pi_{BA}(s_B)] = f(1, 0)$$
$$F[\pi_{AA}(s_A)] = f(1, 0);\ F[\pi_{AA}(s_B)] = f(1, 0)$$
$$F[\pi_{BB}(s_A)] = f(0, 1);\ F[\pi_{BB}(s_B)] = f(0, 1)$$

Following through the same argument from above, we derive the contradictory conditions:

$$J^E_{\mathcal{O}_{RRL}}(\pi_{AB}) > J^E_{\mathcal{O}_{RRL}}(\pi_{AA}) \implies \mathcal{R}(s_B, a_B, s_A) + \alpha f(0, 1) > \mathcal{R}(s_B, a_A, s_A) + \alpha f(1, 0) \quad (1)$$
$$J^E_{\mathcal{O}_{RRL}}(\pi_{BA}) > J^E_{\mathcal{O}_{RRL}}(\pi_{BB}) \implies \mathcal{R}(s_B, a_A, s_A) + \alpha f(1, 0) > \mathcal{R}(s_B, a_B, s_A) + \alpha f(0, 1) \quad (2)$$

Next we will show that ONMR can express this policy ordering. Recall that:

$$J^E_{\mathcal{O}_{ONMR}}(\pi) = f(\mathbb{E}_{\xi \sim \pi, T, I}[\sum_{t=0}^{\infty}\gamma^t\mathcal{R}(s_t, a_t, s'_t)])$$

Let:

$$\mathcal{R}(s_A, a_A, s_B) = -1$$
$$\mathcal{R}(s_A, a_B, s_B) = 1$$
$$\mathcal{R}(s_B, a_A, s_A) = 1$$
$$\mathcal{R}(s_B, a_B, s_A) = -1$$
$$f(x) = |x|$$

Now:

$$J^E_{\mathcal{O}_{ONMR}}(\pi_{AB}) = \left| \frac{1}{2} \sum_{t=0}^{\infty} \gamma^t + \frac{1}{2} \sum_{t=0}^{\infty} \gamma^t \right| = \left| \frac{1}{1-\gamma} \right| = \frac{1}{1-\gamma}$$

$$J^E_{\mathcal{O}_{ONMR}}(\pi_{BA}) = \left| \frac{1}{2} \sum_{t=0}^{\infty} -\gamma^t + \frac{1}{2} \sum_{t=0}^{\infty} -\gamma^t \right| = \left| \frac{-1}{1-\gamma} \right| = \frac{1}{1-\gamma}$$

$$J^E_{\mathcal{O}_{ONMR}}(\pi_{AA}) = \left| \frac{1}{2} \sum_{t=0}^{\infty} \gamma^t + \frac{1}{2} \sum_{t=0}^{\infty} -\gamma^t \right| = |0| = 0$$

$$J^E_{\mathcal{O}_{ONMR}}(\pi_{BB}) = \left| \frac{1}{2} \sum_{t=0}^{\infty} -\gamma^t + \frac{1}{2} \sum_{t=0}^{\infty} \gamma^t \right| = |0| = 0$$

giving us our desired policy ordering.

Next we will show that RM can do this policy ordering:

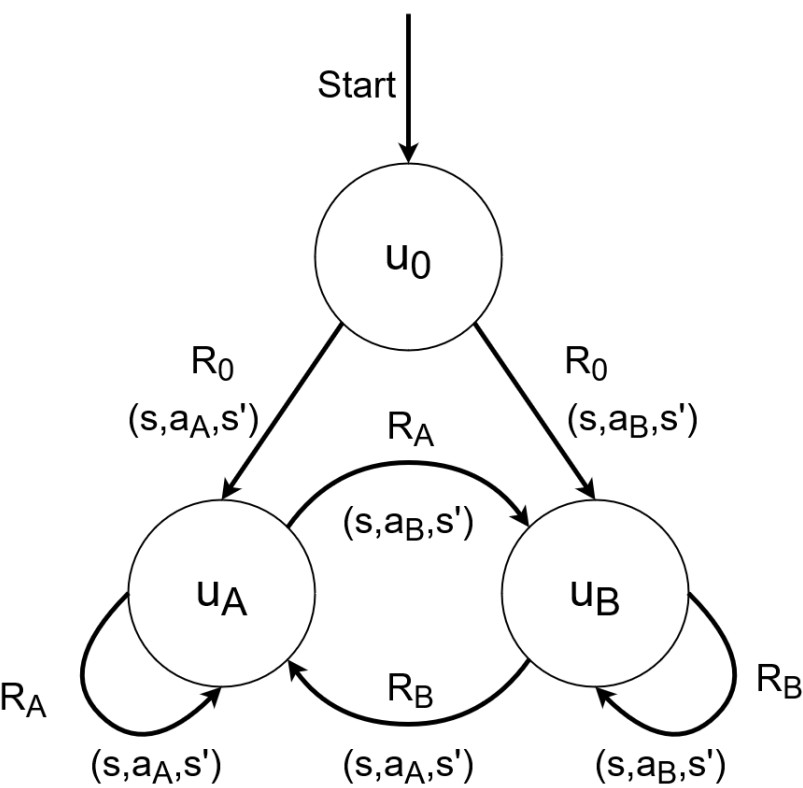

Figure 7: A reward machine which can express the desired policy ordering.

The reward machine looks as above. The reward functions are:

$$\mathcal{R}_1(s_0, a_A, s_A) = 0$$
$$\mathcal{R}_1(s_0, a_B, s_A) = 1$$
$$\mathcal{R}_1(s_A, a_A, s_0) = 0$$
$$\mathcal{R}_1(s_A, a_B, s_0) = 1$$

$$\mathcal{R}_2(s_0, a_A, s_A) = 1$$
$$\mathcal{R}_2(s_0, a_B, s_A) = 0$$
$$\mathcal{R}_2(s_A, a_A, s_0) = 1$$
$$\mathcal{R}_2(s_A, a_B, s_0) = 0$$

For $\pi_{AA}$ we first transition $\delta_u(u_0, s_0, a_A) = u_A$ or $\delta_u(u_0, s_A, a_A) = u_A$ depending on if we start at 0 or 1. For this $\delta_\mathcal{R} = \mathcal{R}_0$ so we get no reward. Next we will always transition $\delta_u(u_A, s_0, a_A) = u_A$ or $\delta_u(u_A, s_A, a_A) = u_A$ which gives $\delta_\mathcal{R} = \mathcal{R}_1$ which results in 0 reward, therefore

$$J^E_{\mathcal{O}_{RM}}(\pi_{AA}) = 0$$

For $\pi_{BB}$ we first transition $\delta_u(u_0, s_0, a_B) = u_B$ or $\delta_u(u_0, s_A, a_B) = u_B$ depending on if we start at 0 or 1. For this $\delta_\mathcal{R} = \mathcal{R}_0$ so we get no reward. Next we will always transition $\delta_u(u_A, s_0, a_B) = u_B$ or $\delta_u(u_A, s_A, a_B) = u_B$ which gives $\delta_\mathcal{R} = \mathcal{R}_2$ which results in 0 reward, therefore

$$J^E_{\mathcal{O}_{RM}}(\pi_{BB}) = 0$$

Now for $\pi_{12}$ we instead have $\delta_u(u_0, s_0, a_A) = u_A$ or $\delta_u(u_0, s_A, a_B) = u_B$ For this $\delta_\mathcal{R} = \mathcal{R}_0$ so we get no reward. Next we have $\delta_u(u_A, s_A, a_B) = u_B$ or $\delta_u(u_B, s_0, a_A) = u_A$ which give reward $\delta_\mathcal{R} = \mathcal{R}_1(s_A, a_B, s_0) = 1$ or $\delta_\mathcal{R} = \mathcal{R}_2(s_0, a_A, s_A) = 1$ meaning

$$J^E_{\mathcal{O}_{RM}}(\pi_{BB}) = \sum_{t=1}^{\infty} \gamma = \frac{\gamma}{1 - \gamma}$$

Similarly for $\pi_{21}$ we have $\delta_u(u_0, s_0, a_B) = u_B$ or $\delta_u(u_0, s_A, a_A) = u_A$ For this $\delta_\mathcal{R} = \mathcal{R}_0$ so we get no reward. Next we have $\delta_u(u_B, s_A, a_A) = u_A$ or $\delta_u(u_A, s_0, a_B) = u_B$ which give reward $\delta_\mathcal{R} = \mathcal{R}_2(s_0, a_B, s_A) = 1$ or $\delta_\mathcal{R} = \mathcal{R}_1(s_A, a_A, s_0) = 1$ meaning

$$J^E_{\mathcal{O}_{RM}}(\pi_{BB}) = \sum_{t=1}^{\infty} \gamma = \frac{\gamma}{1 - \gamma}$$

This gives us our desired policy ordering.

Next we will show that LTL can also express this policy ordering. We will use the shorthand of referring to atomic propositions (which we have defined to be transitions in $\mathcal{S} \times \mathcal{A} \times \mathcal{S}$) as actions; a shorthand atomic proposition $a$ evaluates to true at a time $t$ if and only if the transition at time $t$ includes $a$ as its action.

Consider the LTL formula $((a_A \to \bigcirc a_B) \wedge (a_B \to \bigcirc a_A))$, which reads "$a_A$ implies next $a_B$ and $a_B$ implies next $a_A$." Clearly this is false for $\pi_{AA}$ as the action taken after $a_A$ in not always $a_B$ and false for $\pi_{BB}$ as the action taken after $a_B$ is not always $a_A$. For $\pi_{AB}$ and $\pi_{BA}$ however, the formula is true as action $a_A$ is always followed by $a_B$ and action $a_B$ is always followed by $a_A$. Consequently,

$$J^E_{\mathcal{O}_{RM}}(\pi_{AA}) = 0$$
$$J^E_{\mathcal{O}_{RM}}(\pi_{BB}) = 0$$
$$J^E_{\mathcal{O}_{RM}}(\pi_{AB}) = 1$$
$$J^E_{\mathcal{O}_{RM}}(\pi_{BA}) = 1$$

which gives us the desired policy ordering. □

**Proposition B.32** ($ONMR \not\succeq_{EXPR} LAR$). *There is an environment and an ordering over policies in that environment that Limit Average Reward (LAR) can induce, but Outer Nonlinear MR (ONMR) cannot.*

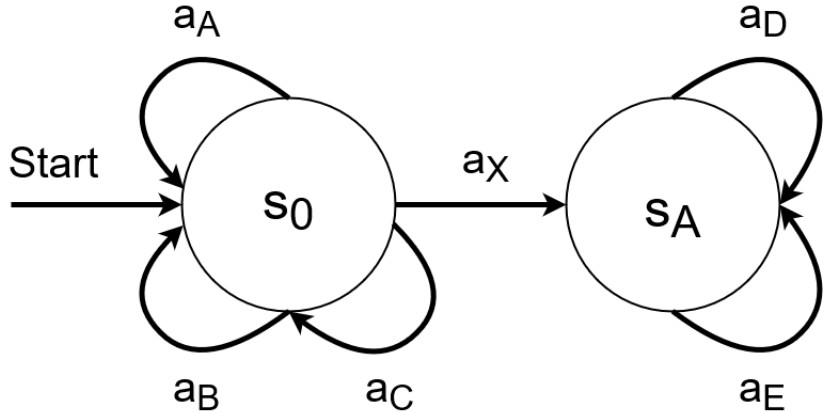

Figure 8: An environment with 2 states $s_0$ and $s_A$ with 4 possible actions in state $s_0$, namely $a_A, a_B, a_C$ and $a_X$ and 2 possible actions in state $s_A$, namely $a_D$ and $a_E$. The starting state is $s_0$.

*Proof by construction.* Consider the environment above. We start in state $s_0$ and have 5 deterministic policies, $\pi_A$, $\pi_B$ and $\pi_C$ which correspond to taking action $a_A$, $a_B$ or $a_C$ in state $s_0$ and policies $\pi_D$ and $\pi_E$ which correspond to taking action $a_x$ in state $s_0$ and action $a_D$ or $a_E$ in state $s_A$, respectively. We also have stochastic policies defined by the conditions

$$\pi_{\theta IJ}(a_I | s_0) = \theta$$
$$\pi_{\theta IJ}(a_X | s_0) = 1 - \theta$$
$$\pi_{\theta IJ}(a_J | s_A) = 1,$$

for $I \in \{A, B, C\}$ and $J \in \{D, E\}$.

We would like the policy ordering:

$$\pi_E \sim \pi_{\theta AE} \sim \pi_{\theta BE} \sim \pi_{\theta CE} \succ \pi_D \sim \pi_{\theta AD} \sim \pi_{\theta BD} \sim \pi_{\theta CD} \succ \pi_B \succ \pi_A$$

This can be done in LAR by setting $\mathcal{R}_E > \mathcal{R}_D > \mathcal{R}_C > \mathcal{R}_B > \mathcal{R}_A$ and $\mathcal{R}_X = 0$. For all policies which have a finite probability of taking $a_X$, $J^E_{\mathcal{O}_{LAR}}(\pi)$ only depends on what the the policy does in state two meaning that: $\pi_E \sim \pi_{\theta AE} \sim \pi_{\theta BE} \sim \pi_{\theta CE}$ and $\pi_D \sim \pi_{\theta AD} \sim \pi_{\theta BD} \sim \pi_{\theta CD}$. Setting the rewards as above completes the rest of the ordering.

Now we will show that ONMR cannot express this policy ordering:

$$J^E_{\mathcal{O}_{ONMR}}(\pi_{\theta IJ}) = f(\mathbb{E}_{\xi \sim \pi_\theta}[G(\xi)]) = f(J^E_{\mathcal{O}_{MR}}(\pi_{\theta IJ}))) = $$
$$f\left( \frac{\mathcal{R}_I}{1 - \theta\gamma} + \frac{1}{1 - (1-\theta)\gamma} \left( \mathcal{R}_x + \frac{\mathcal{R}_J}{1 - \gamma} \right) \right)$$

It is possible to choose $\mathcal{R}_x$ such that $J^E_{\mathcal{O}_{MR}}(\pi_{\theta IJ}) = J^E_{\mathcal{O}_{MR}}(\pi_J)$ however it is not possible to choose $\mathcal{R}_x$ such that both $J^E_{\mathcal{O}_{MR}}(\pi_{\theta ID}) = J^E_{\mathcal{O}_{MR}}(\pi_D)$ and $J^E_{\mathcal{O}_{MR}}(\pi_{\theta IE}) = J^E_{\mathcal{O}_{MR}}(\pi_E)$. Let us assume we have chosen $\mathcal{R}_x$ such that $J^E_{\mathcal{O}_{MR}}(\pi_{\theta IE}) = J^E_{\mathcal{O}_{MR}}(\pi_E)$ and we can therefore not make $J^E_{\mathcal{O}_{MR}}(\pi_{\theta ID}) = J^E_{\mathcal{O}_{MR}}(\pi_D)$. (Of course, an analogous proof goes through under the contrary assumption.)

We need $J^E_{\mathcal{O}_{ONMR}}(\pi_{\theta AD}) = J^E_{\mathcal{O}_{ONMR}}(\pi_{\theta BD}) = J^E_{\mathcal{O}_{ONMR}}(\pi_{\theta CD}) = J^E_{\mathcal{O}_{ONMR}}(\pi_D) \;\; \forall \theta \in [0, 1)$ so $f$ must map the ranges

$$\left(J^E_{\mathcal{O}_{MR}}(\pi_{\theta AD}), J^E_{\mathcal{O}_{MR}}(\pi_D)\right] \rightarrow f(J^E_{\mathcal{O}_{MR}}(\pi_D))$$
$$\left(J^E_{\mathcal{O}_{MR}}(\pi_{\theta BD}), J^E_{\mathcal{O}_{MR}}(\pi_D)\right] \rightarrow f(J^E_{\mathcal{O}_{MR}}(\pi_D))$$
$$\left(J^E_{\mathcal{O}_{MR}}(\pi_{\theta CD}), J^E_{\mathcal{O}_{MR}}(\pi_D)\right] \rightarrow f(J^E_{\mathcal{O}_{MR}}(\pi_D))$$

Either one or more of $J^E_{\mathcal{O}_{MR}}(\pi_A)$, $J^E_{\mathcal{O}_{MR}}(\pi_B)$ or $J^E_{\mathcal{O}_{MR}}(\pi_C)$ are equal which contradicts our desired policy ordering or one of $J^E_{\mathcal{O}_{MR}}(\pi_A)$, $J^E_{\mathcal{O}_{MR}}(\pi_B)$ or $J^E_{\mathcal{O}_{MR}}(\pi_C)$ must be within the range which is mapped to $J^E_{\mathcal{O}_{ONMR}}(\pi_D)$ above. This is also against our policy ordering. Therefore ONMR cannot express this policy ordering. $\qquad\square$

**Proposition B.33** ($ONMR \not\succeq_{EXPR} LTL$). *There is an environment and an ordering over policies in that environment that Linear Temporal Logic (LTL) can induce, but Outer Nonlinear MR (ONMR) cannot.*

*Proof by construction.* Consider the following environment, and the linear temporal logic (LTL) formula below.

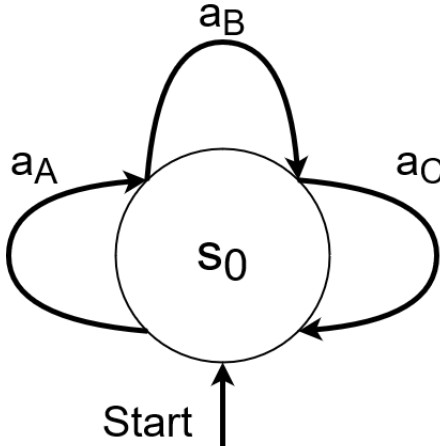

Figure 9: An environment with a single state $s_0$ with three actions $a_A, a_B$ and $a_C$ which all lead back to itself.

This environment has policies $\pi_{\alpha\beta\theta}$ where $\alpha \in [0,1]$, $\beta \in [0,1]$ and $\theta \in [0,1]$ represents the probability of taking action $a_A$, $a_B$ and $a_C$ respectively. We would like the policy ordering

$$\pi_{\alpha\beta\theta} \succ \pi_{\alpha'\beta'\theta'}$$

Where: $\alpha > 0 \wedge \beta > 0 \wedge \theta > 0$ and $\alpha' = 0 \vee \beta' = 0 \vee \theta' = 0$ I.e assign the same reward to all policies which have a positive probability of taking all three actions and the same lower reward otherwise. LTL can express this policy ordering with the following LTL formula:

$$\varphi = \diamond a_A \wedge \diamond a_B \wedge \diamond a_C$$

This LTL formula reads "Eventually $a_A$ and eventually $a_B$ and eventually $a_C$," and is satisfied by a trajectory if and only if the actions $a_A$, $a_B$, and $a_C$ are all taken at some point. Since our trajectories are infinite and we only have one state, eventually taking an action is equivalent to having positive probability of taking an action. Therefore LTL gives us:

$$J^E_{\mathcal{O}_{LTL}}(\pi_{\alpha\beta\theta}) = 1 \quad \forall\, \alpha > 0 \wedge \beta > 0 \wedge \theta > 0$$
$$J^E_{\mathcal{O}_{LTL}}(\pi_{\alpha\beta\theta}) = 0 \quad \forall\, \alpha = 0 \vee \beta = 0 \vee \theta = 0$$

Which is our desired policy ordering. Now let us show that ONMR cannot express the same policy ordering:

Let

$$\mathcal{R}_A = \mathcal{R}(s_0, a_A, s_0)$$
$$\mathcal{R}_B = \mathcal{R}(s_0, a_B, s_0)$$
$$\mathcal{R}_C = \mathcal{R}(s_0, a_C, s_0)$$

Either one or more of these rewards are equal (we will return to the equality case below) or one reward lies between the other two. To fulfill this policy ordering the ONMR function must map all MR rewards of policies which sometimes take all actions to one value, lets call this $\mathcal{R}_H$. The function

must also map all policies which do not do this to another value, $\mathcal{R}_L$. Specifically it must map all policies which only take two actions with any probability to the same value:

$$\left( \frac{\mathcal{R}_A}{1-\gamma}, \frac{\mathcal{R}_B}{1-\gamma} \right) \to \mathcal{R}_L$$

$$\left( \frac{\mathcal{R}_A}{1-\gamma}, \frac{\mathcal{R}_C}{1-\gamma} \right) \to \mathcal{R}_L$$

$$\left( \frac{\mathcal{R}_B}{1-\gamma}, \frac{\mathcal{R}_C}{1-\gamma} \right) \to \mathcal{R}_L$$

However, we also need to map the range:

$$\left( \frac{min(\mathcal{R}_A, \mathcal{R}_B, \mathcal{R}_C)}{1-\gamma}, \frac{max(\mathcal{R}_A, \mathcal{R}_B, \mathcal{R}_C)}{1-\gamma} \right) \to \mathcal{R}_H$$

These ranges clearly overlap and no function can map the same value to two different values. In the case that two or more rewards are the same the proof is even simpler. Call the rewards which are equal $\mathcal{R}_A$ and $\mathcal{R}_B$ with $\mathcal{R}_C$ being the other reward. The ONMR function must for instance map

$$\frac{1}{3}\mathcal{R}_A + \frac{1}{3}\mathcal{R}_B + \frac{1}{3}\mathcal{R}_C \to \mathcal{R}_H$$

$$\frac{2}{3}\mathcal{R}_A + \frac{1}{3}\mathcal{R}_C \to \mathcal{R}_L$$

But these two expressions are equal. Thus, we prove that ONMR cannot express this policy ordering.
$\square$

**Proposition B.34** ($FTR \not\succeq_{EXPR} ONMR$). *There is an environment and an ordering over policies in that environment that Outer Nonlinear MR (ONMR) can induce, but Functions from Trajectories to Reals (FTR) cannot.*

*Proof by construction.* Consider the environment depicted in the diagram below.

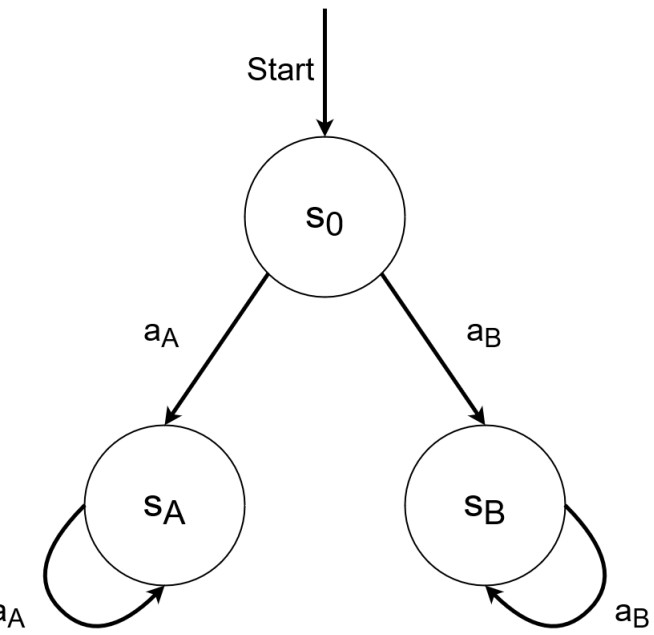

Figure 10: A three-state environment with two possible trajectories corresponding to choosing $a_A$ or $a_B$ in the starting state $s_0$.

We will show that in this environment, the following ONMR specification $(\mathcal{R}, f, \gamma)$ cannot be expressed with any FTR specification.

$$\mathcal{R}_A := \mathcal{R}(s_0, a_A, s_A) = 1, \mathcal{R}_B := \mathcal{R}(s_0, a_B, s_B) = 0 \text{ (as depicted above)}$$

$$f(x) = \mathbb{1}\left[x \geq \frac{1}{2}\right] = \begin{cases} 1 & \text{if } x \geq \frac{1}{2} \\ 0 & \text{if } x < \frac{1}{2} \end{cases}$$

$$\gamma = 0.99$$

With this specification:

$$J^E_{\mathcal{O}_{ONMR}}(\pi) := f\left(\mathop{\mathbb{E}}_{\xi \sim \pi, T, I}\left[\sum_{t=0}^{\infty} \gamma^t \mathcal{R}(s_t, a_t, s_{t+1})\right]\right) \qquad \text{(by definition)}$$

$$= f\left(\mathop{\mathbb{E}}_{\xi \sim \pi, T, I}[\mathcal{R}(s_{t=0}, a_{t=0}, s_{t=1})]\right) \qquad \text{(Rewards are 0 after the first step)}$$

$$= f\left(\pi(a_A|s_0)\mathcal{R}(s_0, a_A, s_A) + \pi(a_B|s_0)\mathcal{R}(s_0, a_B, s_B)\right)$$

$$= f\left(\pi(a_A|s_0)(1) + \pi(a_B|s_0)(0)\right)$$

$$= f\left(\pi(a_A|s_0)\right)$$

$$= \begin{cases} 1 & \text{if } \pi(a_A|s_0) \geq \frac{1}{2} \\ 0 & \text{if } \pi(a_A|s_0) < \frac{1}{2} \end{cases}$$

So this ONMR specification prefers policies that satisfy $\pi(a_A|s_0) \geq \frac{1}{2}$ to those which do not, and has no other preferences. Now, we will show that no FTR specification can express this policy ordering.

In this environment, there are only two possible trajectories: $\xi_A := (s_0, a_A, (s_A, S_A)^*)$ and $\xi_B := (s_0, a_B, (s_B, S_B)^*)$. Suppose there is an FTR specification $(f_{FTR})$ which induces the same policy ordering as the ONMR specification described above. One property this specification must satisfy is that the policy $\pi_A$ that takes action $a_A$ deterministically must be preferred to the policy $\pi_B$ that takes action $a_B$ deterministically.

$$J^E_{\mathcal{O}_{FTR}}(\pi_A) = \mathop{\mathbb{E}}_{\xi \sim \pi_A, T, I}[f_{FTR}(\xi)] > J^E_{\mathcal{O}_{FTR}}(\pi_B) = \mathop{\mathbb{E}}_{\xi \sim \pi_B, T, I}[f_{FTR}(\xi)]$$
$$\implies f_{FTR}(\xi_A) > f_{FTR}(\xi_B)$$

Additionally, if $\pi_{mix}(a_A|s_0) = \pi_{mix}(a_B|s_0) = \frac{1}{2}$, then to match the ONMR specification, the FTR specification should have no preference between $\pi_{mix}$ and $\pi_A$.

$$J^E_{\mathcal{O}_{FTR}}(\pi_A) = \mathop{\mathbb{E}}_{\xi \sim \pi_A, T, I}[f_{FTR}(\xi)] = J^E_{\mathcal{O}_{FTR}}(\pi_{mix}) = \mathop{\mathbb{E}}_{\xi \sim \pi_{mix}, T, I}[f_{FTR}(\xi)]$$
$$\implies f_{FTR}(\xi_A) = \frac{1}{2}f_{FTR}(\xi_A) + \frac{1}{2}f_{FTR}(\xi_B)$$
$$\implies \frac{1}{2}f_{FTR}(\xi_A) = \frac{1}{2}f_{FTR}(\xi_B)$$
$$\implies f_{FTR}(\xi_A) = f_{FTR}(\xi_B)$$

Since it cannot simultaneously be true that $f_{FTR}(\xi_A) > f_{FTR}(\xi_B)$ and $f_{FTR}(\xi_A) = f_{FTR}(\xi_B)$, this means there is no FTR specification which prefers $\pi_A$ to $\pi_B$ but has no preference between $\pi_A$ and $\pi_{mix}$. Therefore, FTR cannot express this policy ordering, concluding the proof. $\square$

This proof highlights that the expectation in the definition of FTR limits it to linearly interpolating between the possible trajectories that a policy can produce. Meanwhile, ONMR specifications can express more complex preferences over trajectory lotteries, such as setting a minimum acceptable probability for a trajectory.

**Proposition B.35** ($FTR \not\succeq_{EXPR} RRL$). *There is an environment and an ordering over policies in that environment that Regularised RL (RRL) can induce, but Functions from Trajectories to Reals (FTR) cannot.*

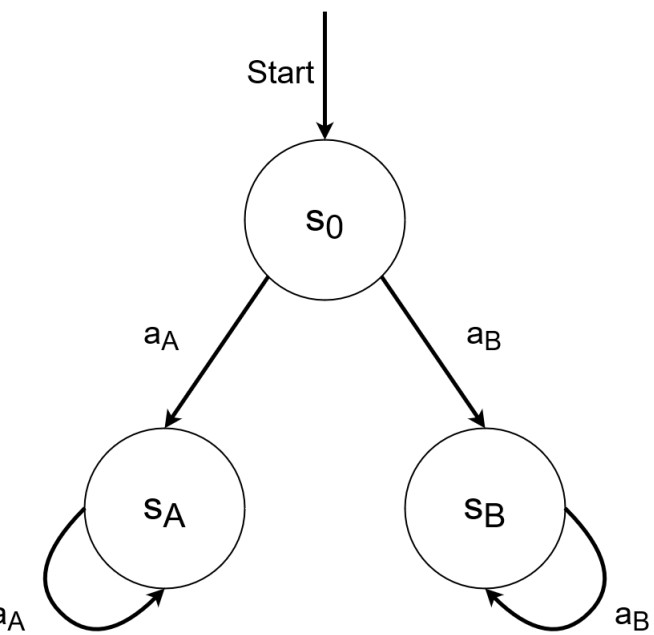

Figure 11: A three-state environment with two possible trajectories corresponding to choosing $a_A$ or $a_B$ in the starting state $s_0$.

*Proof by construction.* $J(\pi) = \mathbb{E}_{\xi \sim \pi}[\sum_{t=0}^{\infty} \gamma^t (\mathcal{R}(s_t, a_t) - \alpha F[\pi(s_t)])]$

where $F : \Delta(A) \to \mathbb{R}$ is a functional of the policy's distribution over actions at a given state.

We construct an example of a task that can be expressed in Regularised RL with $F[\pi(s)] = H[\pi(s)] = -\sum_i \pi(a_i|s) \log \pi(a_i|s)$ (the Shannon entropy), and then prove that it cannot be expressed by any FTR objective.

Suppose we have the environment above with only two trajectories $\xi_A, \xi_B$ through it, both of which can be reached by means of deterministic policies. Let us specify the reward function $\mathcal{R}$ so that both trajectories have the same discounted sum of rewards. Three possible policies are:

$\pi_L$: Deterministic policy that takes $\xi_A$ with probability 1

$\pi_R$: Deterministic policy that takes $\xi_B$ with probability 1

$\pi_m$: Indeterministic policy that takes $\xi_A$ with probability $p$ and $\xi_B$ with probability $1-p$

Since $H[\pi_L(s)] = H[\pi_R(s)] = 0$, and $G(\xi_A) = G(\xi_B)$ by stipulation, $J(\pi_L) = J(\pi_R)$.

Moreover, for $\alpha > 0$, it is easy to show that $J(\pi_m) < J(\pi_L) = J(\pi_R)$. Thus Regularised RL in this environment can express the task $\pi_m \prec \pi_L \sim \pi_R$.

Now consider FTR. The policy evaluation function is:

$J(\pi) = \mathbb{E}_{\xi \sim \pi}[f(\xi)] = P(\xi_A)f(\xi_A) + P(\xi_B)f(\xi_B)$

So:

$J(\pi_L) = f(\xi_A) = a$

$J(\pi_R) = f(\xi_B) = b$

$J(\pi_m) = pf(\xi_A) + (1-p)f(\xi_B)$

Since $p \in (0,1)$ the latter is a convex combination of the former and so we must have EITHER:

$\pi_R \prec \pi_m \prec \pi_L \qquad (a > b)$

$\pi_L \prec \pi_m \prec \pi_R \qquad (b > a)$

$\pi_L \sim \pi_m \sim \pi_R \qquad (a = b)$

None of these gives us the policy ordering induced by the Regularised RL specification above. $\quad\square$

**Proposition B.36** ($GOMORL \not\succeq_{EXPR} FPR$). *There is an environment and an ordering over policies in that environment that Functions from Policies to Reals (FPR) can induce, but Generalised Outer Multi-Objective RL (GOMORL) cannot.*

*Proof by construction.* The essential idea of this proof is that FPR can express preferences between policies that are only distinct on states that neither policy ever visits, while GOMORL cannot. Consider the following environment:

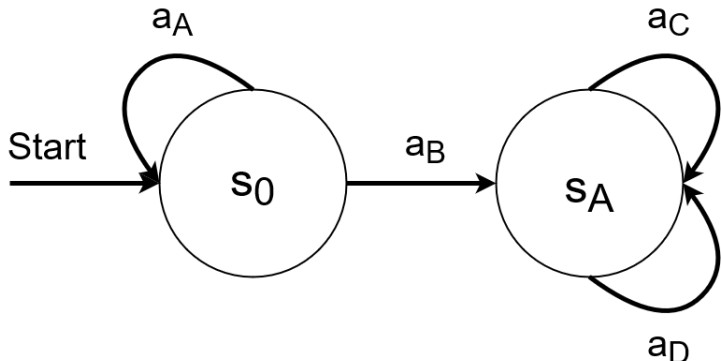

Figure 12: A two-state environment with two actions in the starting state $s_0$ taking $a_A$ back to itself and action $a_B$ leading to state $s_A$. In state $s_A$ we have two actions leading back to itself.

Now consider the following two policies within this environment, $\pi_1$ and $\pi_2$:

$\pi_1(a_A|s_0) = 1, \pi_1(a_B|s_0) = 0, \pi_1(a_C|s_A) = 1, \pi_1(a_D|s_A) = 0$. That is, $\pi_1$ deterministically chooses $a_A$ from $s_0$ and $a_C$ from $s_A$.

$\pi_2(a_A|s_0) = 1, \pi_2(a_B|s_0) = 0, \pi_2(a_C|s_A) = 0, \pi_2(a_D|s_A) = 1$. That is, $\pi_2$ deterministically chooses $a_A$ from $s_0$ and $a_D$ from $s_A$.

The policy ordering induced by any GOMORL specification $(k, \mathcal{R}, \gamma, \succeq_J)$ cannot have any preference between $\pi_1$ and $\pi_2$. To see this, first notice that both of these policies deterministically result in the trajectory $(s_0, a_A, s_0, a_A, s_0, ...)$. This means that each of the $k$ policy evaluation functions, $J_i(\pi)$ for all $i \in [k]$, returns the same value for both policies:

$$J_i(\pi_1) = \mathop{\mathbb{E}}_{\xi \sim \pi_1, T, I}[\sum_{t=0}^{\infty} \gamma^t \mathcal{R}_i(s_t, a_t, s_{t+1})] = \frac{\mathcal{R}_i(s_0, a_A, s_0)}{1 - \gamma} = J_i(\pi_2).$$

Thus, $\vec{J}(\pi_1) = \vec{J}(\pi_2)$. Since $\succeq_J$ must be reflexive as a weak order, $\vec{J}(\pi_1) \succeq_J \vec{J}(\pi_2)$ and $\vec{J}(\pi_2) \succeq_J \vec{J}(\pi_1)$. So in this environment, GOMORL must induce the ordering $\pi_1 \sim \pi_2$.

Clearly, an FPR specification $(J_{FPR})$ need not respect this equality. For instance, the specification could be:

$$J_{FPR}(\pi) = \begin{cases} 1 & \text{if } \pi = \pi_1 \\ 0 & \text{else} \end{cases}$$

The policy ordering this induces has $\pi_1 \succ \pi_2$, so cannot be expressed by GOMORL. This concludes the proof. $\qquad\square$

Recall that GOMORL has the same expressivity as Trajectory Lottery Orderings (TLO) and Occupancy Measure Orderings (OMO), so FPR is also not expressively dominated by either of those formalisms.

**Proposition B.37** ($FPR \not\succeq_{EXPR} GOMORL$)**.** *There is an environment and an ordering over policies in that environment that Generalised Outer Multi-Objective RL (GOMORL) can induce, but Functions from Policies to Reals (FPR) cannot.*

*Proof by construction.* Consider the following environment, and the GOMORL specification $(k, \mathcal{R}, \gamma, \succeq_J)$ below.

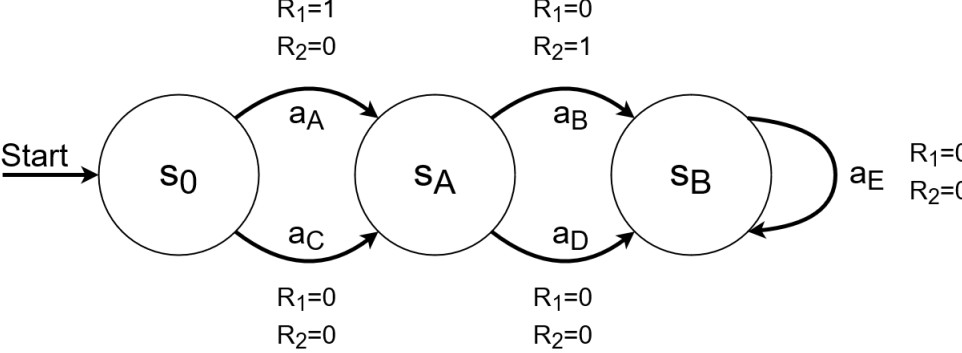

Figure 13: Three state system consisting of the starting state $s_0$ with actions $a_A$ and $a_C$ leading to $s_A$, the state $s_A$ which has two actions $a_B$ and $a_D$ which lead to $s_B$ and finally the state $s_B$ which has action $a_E$ leading to itself. The figure also shows the reward functions $\mathcal{R}_1$ which give 1 reward to action $a_A$ and 0 to everything else and $\mathcal{R}_2$ which give 1 reward to action $a_B$ and 0 to everything else.

- $k = 2$

- $\mathcal{R}_1$ and $\mathcal{R}_2$ are included in the diagram above. $\mathcal{R}_1$ only rewards taking action $a_A$, and $\mathcal{R}_2$ only rewards taking action $a_B$.

- $\gamma = 0.99$ (though this particular value is not important)

- Let $\vec{J}_1 = \langle J_{11}, J_{12} \rangle \in \mathbb{R}^2$ and let $\vec{J}_2 = \langle J_{21}, J_{22} \rangle$. $\succeq_J$ is specified such that $\vec{J}_1 \succeq_J \vec{J}_2$ if and only if $J_{11} > J_{21}$, or $J_{11} = J_{21}$ and $J_{12} \geq J_{22}$. This is a lexicographic ordering on $\mathbb{R}^2$.

Let us clarify the policy ordering expressed by this specification, starting by writing out the policy evaluation functions.

$$
\begin{aligned}
\vec{J}(\pi) &= \left\langle \mathop{\mathbb{E}}_{\xi}^{E,\pi} \left[ \sum_{t=0}^{\infty} \gamma^t \mathcal{R}_1(s_t, a_t, s_{t+1}) \right], ..., \mathop{\mathbb{E}}_{\xi}^{E,\pi} \left[ \sum_{t=0}^{\infty} \gamma^t \mathcal{R}_k(s_t, a_t, s_{t+1}) \right] \right\rangle \\
&= \left\langle \mathop{\mathbb{E}}_{\xi}^{E,\pi} \left[ \sum_{t=0}^{\infty} \gamma^t \mathcal{R}_1(s_t, a_t, s_{t+1}) \right], \mathop{\mathbb{E}}_{\xi}^{E,\pi} \left[ \sum_{t=0}^{\infty} \gamma^t \mathcal{R}_2(s_t, a_t, s_{t+1}) \right] \right\rangle \\
&= \langle \pi(a_A|s_0), \gamma\pi(a_B|s_A) \rangle
\end{aligned}
$$

The last line is true because the only transition that $\mathcal{R}_1$ gives nonzero reward to is $(s_0, a_A, s_A)$, and the only transition that $\mathcal{R}_2$ gives nonzero reward to is $(s_A, a_A, s_B)$ (which can be taken after one step, so must be discounted by $\gamma$).

Now we can look at the policy ordering in terms of the probabilities assigned to $a_A$ and $a_B$. The lexicographic ordering selected turns into the following: $\pi_1 \succeq \pi_2$ if and only if $\vec{J}(\pi_1) \succeq_J \vec{J}(\pi_2)$ if and only if $\pi_1(a_A|s_0) > \pi_2(a_A|s_0)$, or $\pi_1(a_A|s_0) = \pi_2(a_A|s_0)$ and $\pi_1(a_B|s_A) \geq \pi_2(a_B|s_A)$.

Next, let us show that FPR cannot express this policy ordering.

Suppose towards contradiction that $(J_{FPR})$ is an FPR specification that expresses the same lexicographic policy ordering. First, note that if $\vec{J}(\pi_1) = \vec{J}(\pi_2)$, then $\vec{J}(\pi_1) \succeq_J \vec{J}(\pi_2)$ and $\vec{J}(\pi_2) \succeq_J \vec{J}(\pi_1)$, so $\pi_1 \sim \pi_2$ according to the GOMORL specification above. Therefore, for

$(J_{FPR})$ to express the same policy ordering, it must be the case that $\vec{J}(\pi_1) = \vec{J}(\pi_2)$ implies $J_{FPR}(\pi_1) = J_{FPR}(\pi_2)$. This means that $J_{FPR}$, which was have assumed expresses this lexicographic policy ordering, can be decomposed into the GOMORL function from policies to $J$ vectors in $\mathbb{R}^2$, $\vec{J} : \Pi \to \mathbb{R}^2$, and a function from $J$ vectors in $\mathbb{R}^2$ to reals, $f_J : \mathbb{R}^2 \to \mathbb{R}$. So $J_{FPR} = f_J \circ \vec{J}$. Now we will show that no mapping $f_J : \mathbb{R}^2 \to \mathbb{R}$ can express the lexicographic policy ordering given by $\succeq_J$ above, so the supposed $J_{FPR}$ which expresses this policy ordering cannot exist.

For $f_J$ to express the same policy ordering as $\succeq_J$, it must be true that $f_J(\vec{J}(\pi_1)) \geq f_J(\vec{J}(\pi_2)) \iff \vec{J}(\pi_1) \succeq_J \vec{J}(\pi_2)$. Next, let us establish a few notations and facts which will allow us to demonstrate that there is no $f_J$ which satisfies this requirement.

1. Let $B_a$ be the set of possible values for $J_2$ that a policy can have if it has $J_1(\pi) = a$. That is, let $B_a = \{b \in \mathbb{R} : \exists \pi \in \Pi \text{ s.t. } J_1(\pi) = a, J_2(\pi) = b\}$, where $J_1$ and $J_2$ are the policy evaluation functions associated with $\mathcal{R}_1$ and $\mathcal{R}_2$ from the GOMORL specification above.

   In the environment we have specified, we have established that $J_1(\pi) = \pi(a_A|s_0)$ and $J_2(\pi) = \gamma\pi(a_B|s_A)$. So $J_1(\pi) = \pi(a_A|s_0) \in [0,1]$, $J_2(\pi) = \gamma\pi(a_B|s_A) \in [0,\gamma]$, and since the policy's behavior in $s_A$ is not constrained by its behavior in $s_0$, $\vec{J}(\Pi) = [0,1] \times [0,\gamma]$. Therefore, $B_a = [0,\gamma]$ for all $a \in [0,1]$. Since $\gamma = 0.99 > 0$, $B_a = [0, 0.99]$ is an uncountable set. Importantly for this proof, $|B_a| > 1$ for all $a \in J_1(\Pi)$. (Although in this case $B_a$ is the same for all values of $a$, we keep the notation $B_a$, as this makes the proof applicable to any environment in which there are uncountable values of $a$ such that $|B_a| > 1$.)

2. Suppose $a, a' \in J_1(\Pi)$ with $a' > a$. To induce the lexicographic policy ordering through $f_J$, it is necessary that
$$f_J(a', b') > f_J(a, b) \quad \forall b \in B_a, \, b' \in B_{a'}.$$

3. Suppose $b, b' \in B_a$ with $b' > b$. To induce the lexicographic policy ordering, it is necessary that for any $a \in J_1(\Pi)$
$$f_J(a, b') > f_J(a, b).$$

4. Define $f_{J,a} : B_a \to \mathbb{R}$ as $f_{J,a}(b) := f_J(a, b)$. Considering the range of $f_{J,a}$, denoted as $f_{J,a}(B_a)$, let
$$m_{1,a} = \inf(f_{J,a}(B_a)) \quad \text{and} \quad m_{2,a} = \sup(f_{J,a}(B_a)).$$

   Let $I_{f_{J,a}} = [m_{1,a}, m_{2,a}]$, a subset of the reals such that $f_J(a, b)$ lies within this range for all $b \in B_a$. Given point 3 and the fact that $|B_a| > 1$ for all $a \in J_1(\Pi)$, we have $m_{2,a} > m_{1,a}$.

5. For $a' \neq a$, point 2 implies that $f_J(a', b) \notin I_{f_{J,a}}$ for any $b \in B_a$. By extension, $a' \neq a \implies I_{f_{J,a}} \cap I_{f_{J,a'}} = \emptyset$.

6. Let $\mathbb{I} = \{I_{f_{J,a}} : a \in J_1(\Pi)\}$. We define a function $Int : J_1(\Pi) \to \mathbb{I}$ as
$$Int(a) = I_{f_{J,a}},$$

   where "Int" stands for interval. Note that the function $Int$ is injective.

7. Since the rationals are dense in the reals, every closed interval of reals containing at least two distinct real numbers also contains at least one rational. Therefore, we can define a function $\sigma : \mathbb{I} \to \mathbb{Q}$ which, when given an interval as input, outputs a rational within that interval. Owing to the disjoint nature of all $I_{f_{J,a}}$ (from point 5), $\sigma$ is injective.

Recall that $J_1(\Pi) = [0,1]$ is uncountable. Given the assumption that an FPR specification induces the lexicographic policy ordering, we have constructed injective functions $Int : J_1(\Pi) \to \mathbb{I}$ and $\sigma : \mathbb{I} \to \mathbb{Q}$. Composing these, we get an injective function
$$\sigma \circ Int : J_1(\Pi) \to \mathbb{Q}.$$

This leads to a contradiction: $J_1(\Pi)$ is uncountable and $\mathbb{Q}$ is countable, so there cannot be an injective function from $J_1(\Pi)$ to $\mathbb{Q}$. Thus, FPR cannot express this lexicographic policy ordering, and GOMORL can represent policy orderings which FPR cannot. $\qquad\square$

Note that this proof does not rely heavily on the specific environment depicted above or the specific reward functions used by GOMORL; the only features which are relevant to the proof are that there are uncountable values of $J_1(\pi)$ that each have at least 2 values of $J_2(\pi)$. This proof also extends to lexicographic preferences with more than 2 reward functions.

**Proposition B.38** ($ONMR \not\geq_{EXPR} RRL$). *There is an environment and an ordering over policies in that environment that Regularised RL (RRL) can induce, but Outer Nonlinear Markov Reward (ONMR) cannot.*

*Proof by construction.* Consider the following environment.

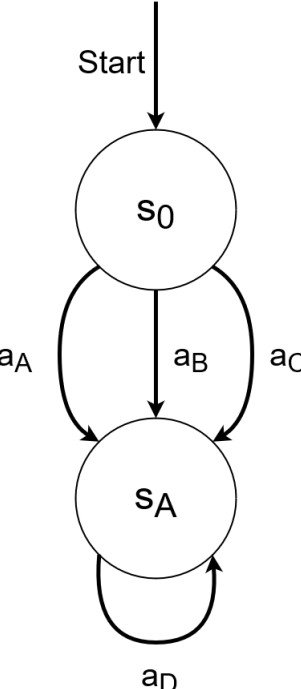

Figure 14: A four-state environment with three possible trajectories corresponding to the choice of $a_A$, $a_B$ or $a_C$ in the starting state $s_0$.

Consider the Regularised RL (RRL) specification given by:

- $\forall (s, a, s') \in \mathcal{S} \times \mathcal{A} \times \mathcal{S} : \quad \mathcal{R}(s, a, s') = 0$

- $\alpha = -1$

- $F[\pi(s)] = |\{a_i : \pi(a_i|s) = 0\}|$. (This $F$ and $\alpha$ incentivise assigning probability 0 to as many actions as possible, which is one way of incentivising more deterministic behavior.)

- $\gamma = 0.9$ (although this is unimportant)

After the first step, all policies deterministically select action $a_D$. Let $F_D$ be the value of $F[\pi(s)]$ if $\pi$ deterministically selects $a_D$ from $s$. Then with this RRL specification:

$$J^E_{\mathcal{O}_{RRL}}(\pi) = \mathbb{E}_{\xi \sim \pi, T, I} \left[ \sum_{t=0}^{\infty} \gamma^t \left( \mathcal{R}(s_t, a_t, s_{t+1}) - \alpha F[\pi(s_t)] \right) \right]$$

$$= \mathbb{E}_{\xi \sim \pi, T, I} \left[ \gamma^0 \left( 0 - (-1)F[\pi(s_0)] \right) + \frac{\gamma F_D}{1 - \gamma} \right]$$

$$= \mathbb{E}_{\xi \sim \pi, T, I} \left[ F[\pi(s_0)] + \frac{\gamma F_D}{1 - \gamma} \right]$$

$$= F[\pi(s_0)] + \frac{\gamma F_D}{1 - \gamma}$$

$$= |\{a_i : \pi(a_i|s_0) = 0\}| + \frac{\gamma F_D}{1 - \gamma}$$

$$= \begin{cases} 2 + \frac{\gamma F_D}{1-\gamma} & \text{if } \pi \text{ selects any one action deterministically} \\ 1 + \frac{\gamma F_D}{1-\gamma} & \text{if } \pi \text{ assigns nonzero probability to exactly two actions} \\ \frac{\gamma F_D}{1-\gamma} & \text{if } \pi \text{ assigns nonzero probability to all three actions} \end{cases}$$

Now let us show that ONMR cannot express the same policy ordering.

Conceptually, the limitation of ONMR that this proof will exploit is that if there are three different possible trajectories, each receiving a different trajectory return, then ONMR cannot distinguish between a policy that deterministically takes this intermediate trajectory and a policy that produces an appropriate probabilistic mixture of the other two.

Suppose towards contradiction that $(\mathcal{R}, f, \gamma)$ is an ONMR specification that expresses this policy ordering. Let $\mathcal{R}_A = \mathcal{R}(s_0, a_A, s_A)$, $\mathcal{R}_B = \mathcal{R}(s_0, a_B, s_B)$, $\mathcal{R}_C = \mathcal{R}(s_0, a_C, s_C)$, and $\mathcal{R}_D = \mathcal{R}(s_A, a_D, s_A)$. Without loss of generality, we can assume that we have numbered the states and actions such that $\mathcal{R}_A \leq \mathcal{R}_B \leq \mathcal{R}_C$. For now, assume that $\mathcal{R}_A < \mathcal{R}_B < \mathcal{R}_C$; we will return to the equality case below. This means that $\mathcal{R}_B = q\mathcal{R}_A + (1 - q)\mathcal{R}_C$ for some real number $q \in (0, 1)$. (Specifically, $q = \frac{\mathcal{R}_C - \mathcal{R}_B}{\mathcal{R}_C - \mathcal{R}_A}$.)

Now suppose we focus our attention on a class of policies of the following form:

$$\pi(a_B|s_0) = \theta, \pi(a_A|s_0) = (1 - \theta)q, \pi(a_C|s_0) = (1 - \theta)(1 - q)$$

Note that an ONMR specification has fixed $q$ in advance, so the only variable is $\theta$. Now, we will show that the ONMR specification must assign the same value to all policies of this form for any $\theta \in [0, 1]$. First, recall that $J^E_{\mathcal{O}_{ONMR}}(\pi_\theta) = f(J^E_{\mathcal{O}_{MR}}(\pi_\theta))$. Therefore, we can analyse $J^E_{\mathcal{O}_{MR}}(\pi_\theta)$ before returning to $J^E_{\mathcal{O}_{ONMR}}(\pi_\theta)$:

$$J^E_{\mathcal{O}_{MR}}(\pi_\theta) = \overset{E,\pi}{\underset{\xi}{\mathbb{E}}} \left[ \sum_{t=0}^{\infty} \gamma^t \mathcal{R}(s_t, A_t, s_{t+1}) \right]$$

$$= \gamma^0 \left( \theta \mathcal{R}_B + (1 - \theta)q\mathcal{R}_A + (1 - \theta)(1 - q)\mathcal{R}_C \right) + \frac{\gamma}{1 - \gamma}\mathcal{R}_D$$

$$= \theta \mathcal{R}_B + (1 - \theta)(q\mathcal{R}_A + (1 - q)\mathcal{R}_C) + \frac{\gamma}{1 - \gamma}\mathcal{R}_D$$

$$= \theta \mathcal{R}_B + (1 - \theta)\mathcal{R}_B + \frac{\gamma}{1 - \gamma}\mathcal{R}_D$$

$$= \mathcal{R}_B + \frac{\gamma}{1 - \gamma}\mathcal{R}_D$$

Connecting this analysis back to ONMR, we get:

$$J^E_{\mathcal{O}_{ONMR}}(\pi_\theta) = f(J^E_{\mathcal{O}_{MR}}(\pi_\theta))$$

$$= f\left( \mathcal{R}_B + \frac{\gamma}{1 - \gamma}\mathcal{R}_D \right)$$

Since this is independent of $\theta$, any ONMR specification that uses three distinct reward values for the three transitions must not have any preference between $\pi_{\theta=1}$ and $\pi_{\theta=0.5}$. However, the RRL specification above does have a preference: $\pi_{\theta=1}$ deterministically selects $a_B$, while $\pi_{\theta=0.5}$ assigns nonzero probability to all three actions. So $J^E_{\mathcal{O}_{RRL}}(\pi_{\theta=1}) = 2 + \frac{\gamma F_D}{1-\gamma} > \frac{\gamma F_D}{1-\gamma} = J^E_{\mathcal{O}_{RRL}}(\pi_{\theta=0.5})$.

The only case left to address is when the reward function $\mathcal{R}$ in the ONMR specification assigns the same reward to any two transitions. That is, $\mathcal{R}_\alpha = \mathcal{R}_\beta$ for some $\alpha, \beta \in \{A, B, C\}$, $\alpha \neq \beta$. No such ONMR specification can express this RRL policy ordering either, because no such specification can distinguish deterministically selecting action $a_\alpha$ or $a_\beta$ from assigning nonzero probability to both. Let $\pi_\alpha$ be a policy that selects $a_\alpha$ deterministically and let $\pi_{\alpha,\beta}$ be a policy that selects each of $a_\alpha$ and $a_\beta$ with probability 0.5.

$$
\begin{aligned}
J^E_{\mathcal{O}_{MR}}(\pi_{\alpha,\beta}) &= \mathop{\mathbb{E}}_{\xi}^{E,\pi_{\alpha,\beta}} \left[ \sum_{t=0}^{\infty} \gamma^t \mathcal{R}(s_t, A_t, s_{t+1}) \right] \\
&= \gamma^0 (0.5(\mathcal{R}_\alpha) + 0.5(\mathcal{R}_\beta)) + \frac{\gamma}{1-\gamma} \mathcal{R}_D \\
&= \mathcal{R}_\alpha + \frac{\gamma}{1-\gamma} \mathcal{R}_D \\
&= J^E_{\mathcal{O}_{MR}}(\pi_\alpha)
\end{aligned}
$$

Thus,
$$
J^E_{\mathcal{O}_{ONMR}}(\pi_{\alpha,\beta}) = f(J^E_{\mathcal{O}_{MR}}(\pi_{\alpha,\beta})) = f(J^E_{\mathcal{O}_{MR}}(\pi_\alpha)) = J^E_{\mathcal{O}_{ONMR}}(\pi_\alpha)
$$

So any ONMR specification that assigns the same reward to two transitions cannot express a preference between $\pi_\alpha$ and $\pi_{\alpha,\beta}$. Meanwhile, the RRL specification above prefers $\pi_\alpha$ to $\pi_{\alpha,\beta}$, since $\pi_\alpha$ deterministically selects $a_\alpha$ while $\pi_{\alpha,\beta}$ assigns nonzero probability to both $a_\alpha$ and $a_\beta$. Therefore, no ONMR specification (whether it assigns equal rewards to two transitions or not) can express the policy ordering expressed by this RRL specification. $\square$

**Proposition B.39** ($RRL \npreceq_{EXPR} LAR$). *There is an environment and an ordering over policies in that environment that Limit Average Reward (LAR) can induce, but Regularised RL (RRL) cannot.*

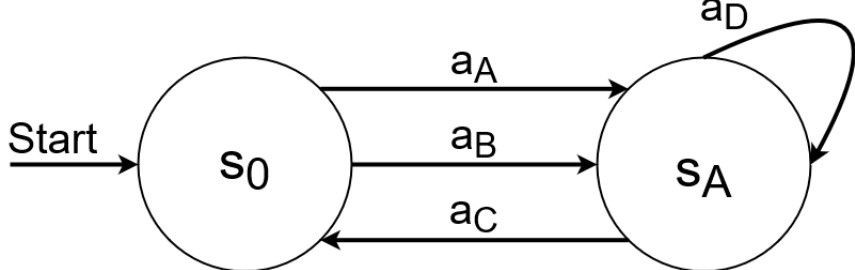

Figure 15: A 2 state environment with a starting state $s_0$ which has 2 actions $a_A$ and $a_B$ leading to $s_A$ and state $s_A$ which has an action $a_C$ leading to $s_0$ and an action $a_D$ leading to itself.

*Proof by construction.* In the figure above we have two states $s_0$ and $s_A$. $s_0$ has actions $a_A$ and $a_B$ which both lead to $s_A$ with reward $\mathcal{R}_A$ and $\mathcal{R}_B$ respectively. $s_A$ has actions $a_D$ which takes it to itself and $a_C$ which leads to $s_0$, these have reward $\mathcal{R}_D$ and $\mathcal{R}_C$. Let the initial state be $s_0$ and $s_A$ with equal probability $\frac{1}{2}$. There are 3 possible deterministic limit cycles here:

- Going from state 1 to state 1 forever via $a_D$

- Alternating between the states via $a_A$ and $a_C$

- Alternating between the states via $a_B$ and $a_C$

There are 4 possible deterministic policies which we will call $\pi_{ij}$ corresponding to taking action $i$ in state 0 and action $j$ in state 1. We would like to express the policy ordering:

$$\pi_{AD} \sim \pi_{BD} > \pi_{AC} > \pi_{BC}$$

This is easy to do in LAR, $J^E_{\mathcal{O}_{LAR}}$ for the different policies is:

$$J^E_{\mathcal{O}_{LAR}}(\pi_{AD}) = J^E_{\mathcal{O}_{LAR}}(\pi_{BD}) = \mathcal{R}_D$$

$$J^E_{\mathcal{O}_{LAR}}(\pi_{AC}) = \frac{\mathcal{R}_A + \mathcal{R}_C}{2}$$

$$J^E_{\mathcal{O}_{LAR}}(\pi_{BC}) = \frac{\mathcal{R}_B + \mathcal{R}_C}{2}$$

We can simply set $\mathcal{R}_D > \frac{\mathcal{R}_A + \mathcal{R}_C}{2}$ and $\mathcal{R}_A > \mathcal{R}_B$ leading to the desired ordering.

It is not possible to express this policy ordering in MR however. Looking at $J^E_{\mathcal{O}_{MR}}$ of the different policies we get:

$$J^E_{\mathcal{O}_{MR}}(\pi_{AD}) = \mathbb{E}_{\xi \sim \pi_{AD}, I}\left[G_{MR}(\xi)\right] = \frac{1}{2}\mathbb{E}_{\xi \sim \pi_{AD}, s_0}\left[G_{MR}(\xi)\right] + \frac{1}{2}\mathbb{E}_{\xi \sim \pi_{AD}, s_A}\left[G_{MR}(\xi)\right] =$$

$$\frac{1}{2}\mathcal{R}_A - \frac{1}{2}\mathcal{R}_D + \sum_{t=0}^{\infty}\gamma^t\mathcal{R}_D$$

$$J^E_{\mathcal{O}_{MR}}(\pi_{BD}) = \frac{1}{2}\mathcal{R}_B - \frac{1}{2}\mathcal{R}_D + \sum_{t=0}^{\infty}\gamma^t\mathcal{R}_D$$

Setting these two to be equal we get:

$$J^E_{\mathcal{O}_{MR}}(\pi_{AD}) = J^E_{\mathcal{O}_{MR}}(\pi_{BD}) \implies \mathcal{R}_A = \mathcal{R}_B$$

However,

$$J^E_{\mathcal{O}_{MR}}(\pi_{AC}) = \frac{1}{2}\sum_{t=0}^{\infty}\gamma^{2t}\mathcal{R}_A + \frac{1}{2}\sum_{t=0}^{\infty}\gamma^{2t+1}\mathcal{R}_C + \frac{1}{2}\sum_{t=0}^{\infty}\gamma^{2t}\mathcal{R}_C + \frac{1}{2}\sum_{t=0}^{\infty}\gamma^{2t+1}\mathcal{R}_A =$$
$$\frac{1}{2}\sum_{t=0}^{\infty}\gamma^{t}\mathcal{R}_A + \frac{1}{2}\sum_{t=0}^{\infty}\gamma^{t}\mathcal{R}_C$$

Similarly:

$$J^E_{\mathcal{O}_{MR}}(\pi_{BC}) = \frac{1}{2}\sum_{t=0}^{\infty}\gamma^{t}\mathcal{R}_B + \frac{1}{2}\sum_{t=0}^{\infty}\gamma^{t}\mathcal{R}_C$$

$$J^E_{\mathcal{O}_{MR}}(\pi_{AC}) > J^E_{\mathcal{O}_{MR}}(\pi_{BC}) \implies \frac{1}{2}\sum_{t=0}^{\infty}\gamma^{t}\mathcal{R}_A + \frac{1}{2}\sum_{t=0}^{\infty}\gamma^{t}\mathcal{R}_C > \frac{1}{2}\sum_{t=0}^{\infty}\gamma^{t}\mathcal{R}_B + \frac{1}{2}\sum_{t=0}^{\infty}\gamma^{t}\mathcal{R}_C \implies$$
$$\frac{1}{2}\sum_{t=0}^{\infty}\gamma^{t}\mathcal{R}_A > \frac{1}{2}\sum_{t=0}^{\infty}\gamma^{t}\mathcal{R}_B \implies \mathcal{R}_A > \mathcal{R}_B$$

This contradicts $\mathcal{R}_A = \mathcal{R}_B$. Therefore MR cannot express this policy ordering meaning that MR cannot express all tasks LAR can.

With a minor modification, this proof also shows that LAR can express some tasks that Regularised RL cannot. Recall the Regularised RL policy evaluation is defined as:

$$J^E_{\mathcal{O}_{RRL}}(\pi) = \mathbb{E}_{\xi\sim\pi,T,I}[\sum_{t=0}^{\infty}\gamma^{t}(\mathcal{R}(s_t, a_t) + \alpha F[\pi(s_t)])]$$

In this environment we have only four actions $(a_D, a_A, a_B, a_C)$, and so we can write $F[\pi(s_t)]$ as an arbitrary function of four probabilities:

$$\alpha F[\pi(s_t)] = f(P(a_D|s_t), P(a_A|s_t), P(a_B|s_t), P(a_C|s_t))$$

Then we have a closed-form expression for each of our four policies under $\mathcal{O}_{RRL}$:

$$J^E_{\mathcal{O}_{RRL}}(\pi_{AD}) = \frac{1}{2}(\mathcal{R}_A + f(0,1,0,0)) - \frac{1}{2}(\mathcal{R}_D + f(1,0,0,0)) + \sum_{t=0}^{\infty}\gamma^{t}(\mathcal{R}_D + f(1,0,0,0))$$

$$J^E_{\mathcal{O}_{RRL}}(\pi_{BD}) = \frac{1}{2}(\mathcal{R}_B + f(0,0,1,0)) - \frac{1}{2}(\mathcal{R}_D + f(1,0,0,0)) + \sum_{t=0}^{\infty}\gamma^{t}(\mathcal{R}_D + f(1,0,0,0))$$

$$J^E_{\mathcal{O}_{RRL}}(\pi_{AC}) = \frac{1}{2}\sum_{t=0}^{\infty}\gamma^{t}(\mathcal{R}_A + f(0,1,0,0)) + \frac{1}{2}\sum_{t=0}^{\infty}\gamma^{t}(\mathcal{R}_C + f(0,0,0,1))$$

$$J^E_{\mathcal{O}_{RRL}}(\pi_{BC}) = \frac{1}{2}\sum_{t=0}^{\infty}\gamma^{t}(\mathcal{R}_B + f(0,0,1,0)) + \frac{1}{2}\sum_{t=0}^{\infty}\gamma^{t}(\mathcal{R}_C + f(0,0,0,1))$$

Using the first two equations allows us to get:

$$\pi_{AD} \sim \pi_{BD} \Rightarrow J^E_{\mathcal{O}_{RRL}}(\pi_{AD}) = J^E_{\mathcal{O}_{RRL}}(\pi_{BD})$$
$$\Rightarrow \mathcal{R}_A + f(0,1,0,0) = \mathcal{R}_B + f(0,0,1,0)$$

This contradicts our assumption when combined with the second equations:

$$\pi_{AC} \succ \pi_{BC} \Rightarrow J^E_{\mathcal{O}_{RRL}}(\pi_{AC}) = J^E_{\mathcal{O}_{RRL}}(\pi_{BC})$$
$$\Rightarrow \mathcal{R}_A + f(0, 1, 0, 0) > \mathcal{R}_B + f(0, 0, 1, 0)$$

So Regularised RL cannot express the policy ordering

$$\pi_{AD} \sim \pi_{BD} \succ \pi_{AC} \succ \pi_{BC}$$

$\square$

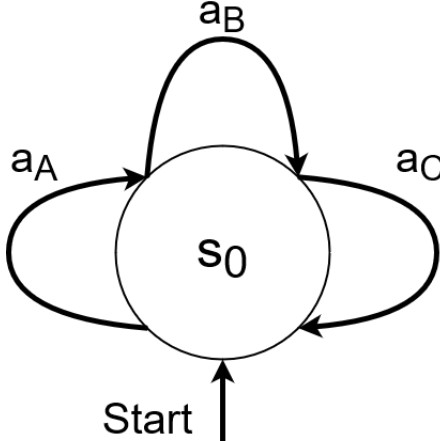

Figure 16: An environment with a single state $s_0$ with three actions $a_A, a_B$ and $a_C$ which all lead back to itself.

**Proposition B.40** ($ONMR \not\sqsupseteq_{EXPR} RM$). *There is an environment and an ordering over policies in that environment that Reward Machines (RM) can induce, but Outer Nonlinear Markov Reward (ONMR) cannot.*

*Proof by construction.* Consider the following environment, and the Reward Machine (RM) specification $(U, u_0, \delta_U, \delta_\mathcal{R}, \gamma)$:

- The environment $E$ is detailed in Figure 16.

- See Figure 17 below for a specification of $U, u_0$, and $\delta_U$. The transitions are labeled with actions as shorthand notation, enabled by the fact that this transition function $\delta_U$ depends only on the action. For instance, the arrow from $u_0$ to $u_A$ labeled with $a_A$ indicates that $\delta_U(u_0, s, a_A, s') = u_A$ for all $s$ and $s'$.

- $\delta_\mathcal{R}(u, u')(s, a, s') = \begin{cases} 1 & \text{if } u' = u_{ABC}, \\ 0 & \text{otherwise.} \end{cases}$

- $\gamma = 0.99$

Intuitively, this RM specification gives reward exactly 1 in a trajectory if and only if all three actions are taken in the trajectory, because that is when machine state $u_{ABC}$ is visited. After giving a reward once, the machine enters state $u_{end}$ and can never give further rewards. The states in the reward machine keep track of which distinct actions have been taken so far (and are named accordingly). The cumulative discounted reward is 0 if any of the actions are never taken, and nonzero if all three actions are taken.

With this specification, a policy that takes all three actions with nonzero probability has probability 1 of eventually taking all three actions and receiving some nonzero discounted reward when the third distinct action is taken, while all policies that assign 0 probability to at least one action are guaranteed to receive 0 cumulative discounted reward. Therefore, a policy that takes all three actions with nonzero probability is preferred to all policies that assign 0 probability to at least one action. We can show that it is impossible to express a policy ordering that satisfies this property with ONMR. Note that much of this proof is very similar to the proof above that ONMR cannot express Regularised RL (RRL).

Suppose towards contradiction that $(\mathcal{R}, f, \gamma)$ is an ONMR specification that induces a policy ordering in which any policy that takes all three actions with nonzero probability is preferred to all policies that assign 0 probability to at least one action. Let $\mathcal{R}_A = \mathcal{R}(s_0, a_A, s_0)$, $\mathcal{R}_B = \mathcal{R}(s_0, a_B, s_0)$, and $\mathcal{R}_C = \mathcal{R}(s_0, a_C, s_0)$. Without loss of generality, we can assume that we have labeled the actions

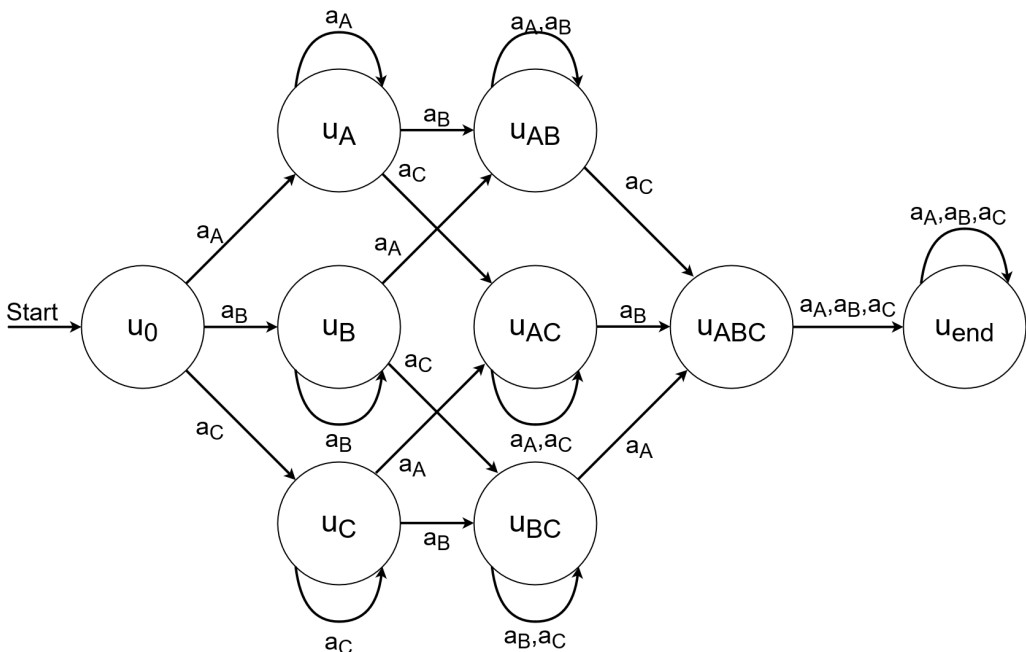

Figure 17: A reward machine which keeps track of which of the three states $s_A$, $s_B$ amd $s_C$ have been visited.

such that $\mathcal{R}_A \leq \mathcal{R}_B \leq \mathcal{R}_C$. For now, assume that $\mathcal{R}_A < \mathcal{R}_B < \mathcal{R}_C$; we will return to the equality case below. This means $\mathcal{R}_B = q\mathcal{R}_A + (1-q)\mathcal{R}_C$ for some real number $q \in (0,1)$.

Now suppose we focus our attention on a class of policies of the following form:

$$\pi(a_B|s_0) = \theta, \pi(a_A|s_0) = (1-\theta)q, \pi(a_C|s_0) = (1-\theta)(1-q)$$

Note that an ONMR specification has fixed $q$ in advance, so the only variable is $\theta$. Now, we will show that the ONMR specification must assign the same value to all policies of this form for any $\theta \in [0,1]$. First, recall that $J^E_{\mathcal{O}_{ONMR}}(\pi_\theta) = f(J^E_{\mathcal{O}_{MR}}(\pi_\theta))$. Therefore, we can analyse $J^E_{\mathcal{O}_{MR}}(\pi_\theta)$ before returning to $J^E_{\mathcal{O}_{ONMR}}(\pi_\theta)$:

$$J^E_{\mathcal{O}_{MR}}(\pi_\theta) = \mathop{\mathbb{E}}_{\xi}^{E,\pi} \left[ \sum_{t=0}^{\infty} \gamma^t \mathcal{R}(s_t, A_t, s_{t+1}) \right]$$

$$J^E_{\mathcal{O}_{MR}}(\pi_\theta) = \gamma^0 \left( \theta\mathcal{R}_B + (1-\theta)q\mathcal{R}_A + (1-\theta)(1-q)\mathcal{R}_C \right)$$

$$J^E_{\mathcal{O}_{MR}}(\pi_\theta) = \theta\mathcal{R}_B + (1-\theta)(q\mathcal{R}_A + (1-q)\mathcal{R}_C)$$

$$J^E_{\mathcal{O}_{MR}}(\pi_\theta) = \theta\mathcal{R}_B + (1-\theta)\mathcal{R}_B$$

$$J^E_{\mathcal{O}_{MR}}(\pi_\theta) = \mathcal{R}_B$$

Connecting this analysis back to ONMR, we get:

$$J^E_{\mathcal{O}_{ONMR}}(\pi_\theta) = f(J^E_{\mathcal{O}_{MR}}(\pi_\theta))$$

$$J^E_{\mathcal{O}_{ONMR}}(\pi_\theta) = f(\mathcal{R}_B)$$

Since $J^E_{\mathcal{O}_{ONMR}}(\pi_\theta) = f(\mathcal{R}_B)$ for all $\theta \in [0,1]$, any ONMR specification with 3 different reward values for the three transitions must not have any preference between $\pi_{\theta=1}$ and $\pi_{\theta=0.5}$. However, the RM specification above does have a preference between these policies. $\pi_{\theta=1}$ deterministically selects $a_B$, while $\pi_{\theta=0.5}$ assigns nonzero probability to all three actions. So $J^E_{\mathcal{O}_{RM}}(\pi_{\theta=0.5}) > J^E_{\mathcal{O}_{RM}}(\pi_{\theta=1})$.

The only case left to address is when the reward function $\mathcal{R}$ in the ONMR specification assigns the same reward to any two transitions. That is, $\mathcal{R}_\alpha = \mathcal{R}_\beta$ for some $\alpha, \beta \in \{A, B, C\}$, $\alpha \neq \beta$. No such

ONMR specification can express this RM policy ordering either, because no such specification can distinguish between policies that assign the same total probability to $a_\alpha$ and $a_\beta$ together, but divide that probability across the two actions differently. Let $\pi_{\alpha,\epsilon}$ be a policy that assigns probability 0.5 to $a_\alpha$, probability 0 to $a_\beta$, and probability 0.5 to the third action (call it $a_\epsilon$). Let $\pi_{\alpha,\beta,\epsilon}$ be a policy that selects each of $a_\alpha$ and $a_\beta$ with probability 0.25 and assigns probability 0.5 to $a_\epsilon$. Again, we can begin by analysing $J^E_{\mathcal{O}_{MR}}$ for these policies.

$$
\begin{aligned}
J^E_{\mathcal{O}_{MR}}(\pi_{\alpha,\beta,\epsilon}) &= \overset{E,\pi_{\alpha,\beta,\epsilon}}{\underset{\xi}{\mathbb{E}}} \left[ \sum_{t=0}^{\infty} \gamma^t \mathcal{R}(s_t, A_t, s_{t+1}) \right] \\
&= \left[ \sum_{t=0}^{\infty} \gamma^t (0.5\mathcal{R}_\epsilon + 0.25\mathcal{R}_\alpha + 0.25\mathcal{R}_\beta)) \right] \\
&= \left[ \sum_{t=0}^{\infty} \gamma^t (0.5\mathcal{R}_\epsilon + 0.5\mathcal{R}_\alpha)) \right] \\
&= \overset{E,\pi_{\alpha,\epsilon}}{\underset{\xi}{\mathbb{E}}} \left[ \sum_{t=0}^{\infty} \gamma^t \mathcal{R}(s_t, A_t, s_{t+1}) \right] \\
&= J^E_{\mathcal{O}_{MR}}(\pi_{\alpha,\epsilon})
\end{aligned}
$$

Returning to the ONMR policy evaluation function and using the fact that $J^E_{\mathcal{O}_{MR}}(\pi_{\alpha,\beta,\epsilon}) = J^E_{\mathcal{O}_{MR}}(\pi_{\alpha,\epsilon})$, we get:

$$
\begin{aligned}
J^E_{\mathcal{O}_{ONMR}}(\pi_{\alpha,\beta,\epsilon}) &= f(J^E_{\mathcal{O}_{MR}}(\pi_{\alpha,\beta,\epsilon})) \\
&= f(J^E_{\mathcal{O}_{MR}}(\pi_{\alpha,\epsilon})) \\
&= J^E_{\mathcal{O}_{ONMR}}(\pi_{\alpha,\epsilon}))
\end{aligned}
$$

So any ONMR specification that assigns the same reward to two transitions cannot express a preference between $\pi_{\alpha,\epsilon}$ and $\pi_{\alpha,\beta,\epsilon}$. Meanwhile, the RM specification above prefers $\pi_{\alpha,\beta,\epsilon}$ to $\pi_{\alpha,\epsilon}$, since $\pi_{\alpha,\beta,\epsilon}$ assigns nonzero probability to all three actions while $\pi_{\alpha,\epsilon}$ assigns probability 0 to $a_\beta$. Therefore, no ONMR specification (whether it assigns equal rewards to two transitions or not) can express the policy ordering expressed by this RM specification. $\square$

