# OpenReview forum: "On the Expressivity of Objective-Specification Formalisms in Reinforcement Learning"
_ICLR.cc/2024/Conference — ICLR 2024 poster_

### Official Review · Reviewer_EiAB · 2023-10-26

**Soundness:** 3 good
**Presentation:** 2 fair
**Contribution:** 2 fair
**Rating:** 3
**Confidence:** 3

**Summary:**

The paper aims to provide a comprehensive picture of the expressivity of a pletora of objective specifications in RL, which include reward machines, trajectory feedbacks, multi-objective formulations among the others. The expressive power of an objective specification is measured in terms of the orderings it can induce over stationary Markovian policies in every possible environment, which is formally defined through a set of states, a set of actions, a transition kernel, and an initial state distribution.

**Strengths:**

- (Significance) The paper tackles a very important question on how to specify the objective in RL. Given the recent results showing the limitations of Markovian rewards (e.g., Abel et al., 2022) and the growing popularity of alternative feedback models, such as RL from human feedback, regularized RL, and convex RL to name a few, I think this is a very relevant research line.
- (Originality) I am not aware of previous effort in exploring the expressivity of such a comprehensive set of objective specifications.
- (Limitations) The paper is upfront in reporting the limitations of the contribution (see Section 4.1).

**Weaknesses:**

- (Arbitrary analysis) The works makes a series of arbitrary choices in defining the expressivity of an objective specification (e.g., limiting the analysis to ordering of stationary policies) that are not sufficiently motivated in my opinion.
- (Implications) While the paper provide lots of results in terms of expressivity (as defined in the paper) of the objective specifications, most of the implications of their analysis are left as future work.
- (Related work) Given the nature of the work, which is in part a survey of objective specifications in RL, the discussion of the related literature is insufficient.
- (Clarity) While the main messages of the paper are clear and easily accessible, some notation choices are somewhat confusing, and they also depart the objective specifications from how they are formulated in the literature in some cases.

GENERAL COMMENT

The paper tackles a very important research problem, which can be coarsely rephrased as "how can we specify the objective in RL to obtain the behaviour that we want as output of the learning process"? However, it falls somewhat short of answering this questions, providing an analysis of a set of objective specifications in a very limited setting (full ordering of stationary policies), and without addressing the implications of their results. Some of the reported results can still be valuable, although it is not fully clear what is novel and what is already known, but I believe that a better motivation of the choices on how to evaluate expressivity is necessary to clear the bar for acceptance. Thus, I am currently providing a negative evaluation, but I will consider updating my score if the authors can convince me on why comparing objective specifications in terms of full ordering of stationary policies matters.

DEFINITION OF EXPRESSIVITY

(C1. Policy ordering) The paper makes clear that the expressivity of the objective specifications is evaluated only in terms of policy ordering, but I am wondering whether this is the most natural notion of task, and how can be motivated in practice. Arguably, in most of the real-world problems, what really matters is the outcome of the learning process, i.e., what policy is ultimately learned. In this view, the SOAP looks more natural (terminology from Abel et al., 2022). I think that a soft-SOAP task notion would make for a stronger case in the expressivity evaluation. Anyway, the choice of task notion has a strong impact on the expressivity overview and it shall be motivated carefully in my opinion.

(C2. Stationary policies) Another weak point I see in the analysis is restricting the policy class to stationary Markovian policies. It is well-known that a Markovian reward is expressive enough to specify objectives over deterministic stationary Markovian policies, which leaves only the set of stochastic policies to the other specifications. I am wondering why one would want a stochastic policy in the first place. Typical motivations are robustness to model misspecifications and adversaries, or perhaps to make the learning problem more efficient (e.g., regularized objectives). For some of the reported objective specifications, I think that the main selling point is that they cannot be optimized by Markovian policies. For instance, FTR allows to extract optimal history-dependent policies, which can be valuable, although in the analysis turns out to be less expressive than FOMR, which does not look that useful.

IMPLICATIONS

(C3. Expressivity against tractability) Whereas the Section 4.1 clarifies that tractability is not considered in the analysis, this may defeat the purpose of the analysis itself. Very general models have been proposed in the past, but they intractability makes them less appealing for practitioners. For instance, the POMDP model arguably subsumes all of the objective specifications reported in the paper, but it is intractable in general.

RELATED WORK

(C4. Missing references) This paper is surveying a lot of objective specifications but does little to relate those with pre-existing literature. Various previous works considered extension to the standard RL model (based on Markovian rewards), such as:
- RL with general utilities (e.g., Zhang et al., Variational policy gradient method for reinforcement learning with general utilities, 2020)
- Convex RL (e.g., Hazan et al., Provably efficient maximum entropy exploration, 2019; Zahavy et al., Reward is enough for convex MDPs, 2021; Geist et al., Concave utility reinforcement learning: the mean-field game viewpoint, 2021; Mutti et al., Challenging common assumptions in convex reinforcement learning, 2022; Mutti et al., Convex reinforcement learning in finite trials, 2023)
- Vectorial rewards (e.g., Cheung, Regret minimization for reinforcement learning with vectorial feedback and complex objectives, 2019)
- Trajectory feedback (e.g., Chatterji et al., On the theory of reinforcement learning with once-per-episode feedback, 2021)
- Duelling feedback (e.g., Saha et al., Dueling RL: Reinforcement learning with trajectory preferences, 2023)
- Preference feedback (e.g., Xu et al., Preference-based reinforcement learning with finite-time guarantees, 2020)

**Questions:**

- Can the authors address the comment reported above?

- Can the authors clarify which of the reported result is new and not directly implied by previous works?

- How can be OMO more expressive than FOMR? I am probably missing something, but why we cannot take for any ordering over occupancy measures $m_1 > m_2 > ... > m_n$ a function $f: M \to \mathbb{R}$ such that $f(m_1) > f (m_2) > ... > f (m_3)$?

---

> ### Author Response · Authors · 2023-11-18
>
> Thank you for your feedback! Our responses are as follows:
>
> (C1. Policy ordering) We compare the expressivities of formalisms in terms of policy orderings because the policy ordering is the most fine-grained unit of analysis for this purpose. SOAPs are constructed by taking a policy ordering, choosing a cutoff point to divide the policies into two classes (i.e., those above and below the cutoff), and then discarding all information about how the policies are ordered within each class. Thus, any difference in expressivity that is captured by an equal-SOAP or range-SOAP (also terminology from Abel et al., 2022) is captured by the policy ordering task definition. We will edit the paper to discuss this and other motivations for using policy orderings in more detail.
>
> (C2. Stationary policies) Markov reward can make any deterministic stationary policy optimal, but cannot express all orderings of deterministic stationary policies; for example, see the proof of Proposition A.31 in our appendix. Also, note that many of our results directly carry over to history-dependent policies; if a formalism A can express a stationary policy ordering which is not expressible by formalism B, then clearly A can express an ordering over history-dependent policies that B cannot. So every negative expressivity result carries over directly. Further, several of our positive proofs do not assume that we are considering only stationary policies. We will make edits to underscore this point and indicate what work remains to be done to round out the results for history-dependent policies. While many results carry over, rounding out these results would still be a significant undertaking which is out of scope for our work. This is an interesting direction for future research, especially given your point that some formalisms are well-suited to expressing tasks where history-dependent policies are essential. We hope that by adding further details about which of our results carry over to the history-dependent case we make our contribution more significant and lessen the amount of work remaining to complete the history-dependent analysis.
>
> (C3. Expressivity against tractability) Tractability is crucial, but analysing the tractability of each formalism would be a massive undertaking. Much of the existing RL literature can be construed as an attempt to establish the practical tractability of the MDP formalism. It is therefore not feasible to tackle this question in our paper. Many formalisms we compare are studied in the literature, and past work we reference does give an indication of the tractability of these formalisms. We hope that the tractability of the formalisms in our comparison will become clearer over time as new algorithms are proposed and applied to more problems. We also think our expressivity comparison is a valuable contribution by itself. For instance, our work can help an RL practitioner identify a formalism that can express their task of interest and seek out an appropriate algorithm. If no existing algorithms are suitable for the task, this can prompt further research. The Skalse et al., 2022 paper “Lexicographic Multi-Objective Reinforcement Learning” is an example of research of this sort: lexicographic objectives are expressible with GOMORL but not with Markovian rewards, and the paper provides algorithms for solving them. We will edit the paper to better convey a) the reasons we are unable to provide a thorough account of tractability and b) why our work is still a valuable step that future work can build on.
>
> (C4. Missing references) Thank you for providing this list of references. We construed Related Work as a section to discuss the formalisms that we were studying, but after seeing your comment, we believe it is sensible to point to formalisms beyond those that we consider to highlight that we haven’t considered all interesting formalisms. We will make these edits.
>
> (Clarity) Notational departures from the literature are motivated by a need to use comparable notations for the formalisms that we are comparing. We certainly want to make our results as clear and accessible as possible, so any pointers to specific notational choices that are difficult to parse would be appreciated.
>
> Questions:
> 1. We attempt to address your comments above.
> 2. Most of our results are novel, unless mentioned in the Related Work. Unfortunately, it is difficult to fully disentangle original work from previous work since our proofs often rely on a combination of existing and novel ideas.
> 3. In brief, you could have something akin to a lexicographic order over a set of occupancy measures, and there would be no function from occupancy measures to reals that preserves this order. See Proposition A.37 in our appendix (from which this results follows) for details.
>
> Please let us know if our response and planned edits address your concerns and if you have any other suggestions. We hope that you will consider increasing your score.

---

> > ### Comment · Reviewer_EiAB · 2023-11-21
> >
> > I want to thank the authors for their detailed replies and for taking my concerns in serious consideration.
> >
> > Unfortunately, I am still doubtful over the motivation for policy orderings (C1) and stationary policies (C2) and I am worried that  those choices have large impact on the reported results, potentially diminishing the value of the contribution. While I think that extending the analysis to history-dependent policies may clear the bar for acceptance, I cannot judge those changes without looking at them. For these reasons, I am planning to keep my original evaluation. However, I can see that other reviewers are highly positive on the paper: I will engage in the discussion to understand why they are not concerned on the problem formulation as I am.
> >
> > > (C1. Policy ordering) We compare the expressivities of formalisms in terms of policy orderings because the policy ordering is the most fine-grained unit of analysis for this purpose. SOAPs are constructed by taking a policy ordering, choosing a cutoff point to divide the policies into two classes (i.e., those above and below the cutoff), and then discarding all information about how the policies are ordered within each class. Thus, any difference in expressivity that is captured by an equal-SOAP or range-SOAP (also terminology from Abel et al., 2022) is captured by the policy ordering task definition. We will edit the paper to discuss this and other motivations for using policy orderings in more detail.
> >
> > I see that policy ordering is more general than SOAP formulation, but I am wondering whether it is unnecessarily general. Can the authors name a practical application where a policy ordering is needed? Arguably, in most real-world problems, one is satisfied with getting a "good" (if not optimal) policy out of the learning process, for which SOAP seems to be enough. To uphold this claim, I would also not that for several decades Markovian rewards have been the standard task specification, which, as the authors rightly noted, cannot induce all the policy orderings. Of course any difference in expressivity over SOAP is captured by policy ordering, but the opposite is not true.

---

> ### Author Response · Authors · 2023-11-23
>
> > I want to thank the authors for their detailed replies and for taking my concerns in serious consideration.
>
> We likewise thank you for your engagement, and for your detailed review of the paper!
>
> > While I think that extending the analysis to history-dependent policies may clear the bar for acceptance, I cannot judge those changes without looking at them.
>
> We have submitted a modified version of the paper that includes our partial analysis of history-dependent policies. The most relevant parts are a paragraph in the Limitations and Future Work subsection (4.1), Appendix A.3 (referenced in Section 4.1), and Table 3 (referenced in Appendix A.3). We will add that we actually have more results for history-dependent policies prepared than appear in the updated paper, since some additional results rely on proofs that we do not have time to type up before the end of the discussion period. We appreciate that you cannot weight this very heavily given that it is not included in the current submission, but there are only 12 white boxes (representing missing results) in our most up-to-date internal version of Table 3, and we can certainly make that adjustment later if the paper is accepted.
>
> > Can the authors name a practical application where a policy ordering is needed?
>
> The discussion of policy orderings is complicated, and we will provide two strands of argument in favor of our choice.
>
> First: In most real-world situations, we cannot find a globally optimal policy. For example, this is typically not feasible in robotics tasks, or when optimising a language model to exhibit some behaviour (such as politeness, etc). For this reason, if we want a task specification to *robustly* incentivise desirable behaviour, then it is important that this task specification expresses the correct preferences between the (sub-optimal) policies that the policy optimisation algorithm in fact may generate. One way to see the practical importance of this consideration is to note that MORL is very common in practice. One of the main reasons for this is that the MORL formulation makes it easier to describe how certain trade-offs should be made, and this is captured by the fact that MORL can express policy orderings which scalar Markov rewards cannot. For this reason, we think it is informative and relevant to consider the policy orderings which a task specification method is able to capture.
>
> Second: We appreciate that by choosing policy orderings as our task notion there is the possibility of finding a greater number of expressive differences between formalisms than would have been found with the use of SOAPs, and this gives rise to the concern that these “extra” expressive differences may lack practical significance. However, we believe that the negative results we provide in the paper point to interesting limitations of the different formalisms, and one of the following must be true: either those expressive differences would have also been seen under the SOAP task notion, in which case the use of policy orderings makes no difference to the discovered expressivity relationship; or those differences would have been invisible to the SOAP task notion, in which case use of SOAPs would have concealed from us the interesting limitations that we discovered. Insofar as you agree that the proofs of our negative results reveal meaningful insights about the expressive limitations of the different formalisms, the use of POs rather than SOAPs is at worst a harmless choice and at best even the superior choice. (Note also that many interesting expressive limitations of Markovian rewards are already visible under a SOAP formulation — so that is not merely a quirk of our choice of policy orderings as the task notion.)
>
> In any case, while we agree that other problem formulations can also be relevant, we do not think that this fact alone is sufficient ground for rejection of the paper.

---

### Official Review · Reviewer_Re36 · 2023-10-30

**Soundness:** 3 good
**Presentation:** 3 good
**Contribution:** 3 good
**Rating:** 6
**Confidence:** 5

**Summary:**

This paper surveys a wide range of objective-specification formalisms in reinforcement learning setting and establishes the theoretical expressive relationship among these formalisms. These results facilitate the choice of objective formalism for RL practitioners.

**Strengths:**

1. This work connects 17 objective-specification formalisms, compares their expressivity, and present results using a Hasse diagram. This contribution is clear.
2. The discussion and review of each formalism is helpful for RL practitioners to choose specifications.

**Weaknesses:**

1. There are some other formal language-based specification formalisms for RL, such as Signal Temporal Logic, which is more powerful than LTL because of its robustness property, and there are a wide range of literature in this direction, see e.g., [1], [2]. The authors are encouraged to study STL formalism with others to make the comparison more thorough.

2. As mentioned in Sec. 4.1, there is no tractability comparison among various formalisms, which is however very important for researchers/practitioners to decide which one they want to use.

3. The writing can be further improved. For instance, some notations can be be more precise (e.g., in Def. 2.7, vector in R^|S||A||S| should be replaced by R^|S| \times R^|A| \times R^|S|).

[1] Balakrishnan, A. and Deshmukh, J.V., 2019, November. Structured reward shaping using signal temporal logic specifications. In 2019 IEEE/RSJ International Conference on Intelligent Robots and Systems (IROS) (pp. 3481-3486). IEEE.
[2] Wang, J., Yang, S., An, Z., Han, S., Zhang, Z., Mangharam, R., Ma, M. and Miao, F., 2023. Multi-Agent Reinforcement Learning Guided by Signal Temporal Logic Specifications. arXiv preprint arXiv:2306.06808.

**Questions:**

I have no question.`

---

> ### Author Response · Authors · 2023-11-18
>
> First of all, we would like to thank reviewer Re36 for their feedback! Our responses to your comments are as follows:
> 1. The main reason for why we have not considered STL in this work is that we are focusing on discrete environments (i.e., environments with discrete time, where the set of states and the set of actions are both finite). Most of the objective specification methods that we consider are best suited for discrete environments, and in such environments, STL is equivalent to LTL. A generalisation to environments with continuous time would be interesting, but we consider it to be out of scope for this work. We will however amend the text to provide references to recent work on STL in RL.
> 2. Tractability is crucial, but analysing the tractability of each formalism would be a massive undertaking. Much of the existing RL literature can be construed as an attempt to establish the practical tractability of the MDP formalism. It is therefore not feasible to tackle this question in our paper. Many formalisms we compare are studied in the literature, and past work we reference does give an indication of the tractability of these formalisms. We hope that the tractability of the formalisms in our comparison will become clearer over time as new algorithms are proposed and applied to more problems. We also think our expressivity comparison is a valuable contribution by itself. For instance, our work can help an RL practitioner identify a formalism that can express their task of interest and seek out an appropriate algorithm. If no existing algorithms are suitable for the task, this can prompt further research. The Skalse et al., 2022 paper “Lexicographic Multi-Objective Reinforcement Learning” is an example of research of this sort: lexicographic objectives are expressible with GOMORL but not with Markovian rewards, and the paper provides algorithms for solving them. We will edit the paper to better convey a) the reasons we are unable to provide a thorough account of tractability and b) why our work is still a valuable step that future work can build on.
> 3. We are certainly keen to improve the presentation and the notation if you could give us examples of places where this is unclear. Note that $\mathbb{R}^{|S||A||S|}$ is isomorphic to $\mathbb{R}^{|S|} \times \mathbb{R}^{|A|} \times \mathbb{R}^{|S|}$, so there is no mathematical reason to substitute these.
>
> We hope that this clarifies our contributions and the choices that we made. Please let us know if our response and planned edits address your concerns and if you have any other suggestions. We also hope that you will consider increasing your score.

---

> > ### Comment · Reviewer_Re36 · 2023-11-19
> >
> > Thanks for your clarifications and they addressed my concerns. I improved my score accordingly.

---

### Official Review · Reviewer_XNit · 2023-10-31

**Soundness:** 4 excellent
**Presentation:** 3 good
**Contribution:** 4 excellent
**Rating:** 10
**Confidence:** 3

**Summary:**

This paper compares the relative expressivity of different task specification mechanisms for reinforcement learning agents. The authors set out to compare the relative expressivity of Markov rewards, limit average rewards, reward machines, linear temporal logic, regularized RL, and outer non-linear Markov rewards, and prove that each of these formalisms have a non-zero intersection, but also a non-zero exclusive component. In doing so the authors define 17 different task specification formalisms, and posit that a complete ordering over all possible policies of an agent is the most expressive (if impractical) tasks specification mechanism available for RL agents.

The authors primarily provide expressivity result as a Hasse diagram where a directed edge represents subsumption of the task specification formalism.

**Strengths:**

1. **Timely and major significance**: The authors assertion that each of the popular reward mechanisms do not completely express all aspects of other reward formalisms is an important result, and the conclusions are supported by extensive background materials, and proofs. I also believe that the recommendation of the authors that researchers be more aware of reward specification frameworks, and algorithms catering to them is also well received.

2. **Comprehensive survey of reward mechanisms**: The authors were systematic in their coverage of task specification formalisms, and have exhaustively documented each pairwise dependency. Although for presentational clarity, the proofs had to be relegated to the supplementary materials. The paper seems well indexed.

**Weaknesses:**

1. The authors center the discussion around ordering of stationary policies, and while it is justified in the context of the paper, and where the research effort has been directed historically, there are many instances (especially with partial observability), where non-stationary policy over observations must be implemented.

**Questions:**

1. I am curious if the authors only considered a fragment of LTL, or the entirety of LTL that allows for accpetability definitions over infinitely long trajectories
2. I am also uncertain if ordering over stationary policies is enough to describe an LTL task that requires a memory dependent policy (usually implemented through a cross product MDP)?

---

> ### Author Response · Authors · 2023-11-18
>
> First of all, we would like to thank reviewer XNit for their review! Our responses to your comments and questions are as follows:
>
> Weaknesses:
> 1. We aim to address your concern about nonstationary policies below in our response to your second question.
>
> Questions:
> 1. We consider the full version of LTL, i.e. the version that considers infinitely long trajectories.
> 2. In our paper we define a task to be an ordering of stationary policies, and so stationary policy orderings are (trivially) capable of expressing any LTL task so defined. That is why the formalism of stationary policy orderings (PO) expressively dominates LTL in our results. However, as you rightly indicate, certain objective specifications in LTL (and other formalisms) are of interest primarily for their behaviour on memory-dependent policies, and it is regrettable that we were unable to consider orderings over the set of memory-dependent policies as well as over the set of Markovian policies. We chose to focus on Markovian policies because these are assumed by many common RL algorithms. Also, it should be noted that many of our results straightforwardly carry over to memory-dependent policies. In particular, all our negative expressivity results carry over directly (if a formalism X can express an ordering over Markovian policies that is not expressible in a formalism Y, then this trivially implies that X can express an ordering over the broader class of memory-dependent policies that is not expressible in Y), and many of our proofs establishing positive expressivity relationships apply equally well when the class of policies is expanded to include memory-dependent policies. We will edit the paper to underscore this point, and indicate what work remains to be done to round out the results for memory-dependent policies (which is, however, outside the scope of our work).
>
> We hope that this answers your questions!

---

### Official Review · Reviewer_mz4B · 2023-11-01

**Soundness:** 4 excellent
**Presentation:** 4 excellent
**Contribution:** 4 excellent
**Rating:** 8
**Confidence:** 3

**Summary:**

Um. Wow.

This paper is remarkable in many ways. The paper contributes a comprehensive set of proofs designed to show which reward formalisms in RL are distinctive and which can be subsumed by others; the central contribution of the paper is a lattice that diagrams these relationships. One conclusion from the work is that many formalisms offer distinctive capabilities, and that no single formalism is dominant.

The paper really does seem to be an incredible, detailed set of work. It is is highly theoretical, and is likely to be useful only to a select subset of researchers who are deeply steeped in RL research.

I am not capable of evaluating the proofs in any sort of detail, even though I have spent many years in RL. However, the way that the paper is written, combined with the parts that I was able to evaluate, suggests a serious contribution to the literature.

The question is: is ICLR the right venue for this work? While this is probably the magnum opus of this sort of work, it's unclear how much the general ICLR community would benefit from including it in the conference proceedings.

Would a journal be a better outlet?

**Strengths:**

+ Comprehensive evaluation
+ Clearly summarized results
+ Interesting resulting insights
+ Well-written, despite extensive technical detail

**Weaknesses:**

- Likely to be useful to a very small subset of people
- Unclear if ICLR is the right venue
- Dense notation / writing

**Questions:**

Why do you think that ICLR is the right venue for this work, as opposed to an RL-specific conference or a journal?

---

> ### Author Response · Authors · 2023-11-18
>
> First of all, we would like to thank reviewer mz4B for their review! Our responses to your comments and question are below.
>
> We believe our work is potentially relevant to anyone interested in training AI systems to solve novel sequential decision-making tasks, since the strengths and limitations we identify for formalisms may be relevant for a variety of such tasks. Therefore, we hope that our results are of interest to a substantial portion of the general ICLR community. However, depending on the reception of the paper, we may consider extending it to a journal submission.
>
> We appreciate that the writing can be somewhat dense in places, though this is commensurate with the density of formal results in our paper, which makes this style of writing difficult to avoid. However, we will work on making the text more accessible by incorporating the feedback from the reviewers.

---

### Author Response · Authors · 2023-11-23

In our updated paper submission, we have made most of the adjustments discussed with reviewers. We have a couple of additional comments on our changes:

1. Due to the page limit, some additional discussion on topics from reviewer feedback was added to the appendices, along with sentences in the main body of the text that refer to those appendix sections.
2. We have not yet had time to add the related works pointed out to us by the reviewers and discuss their relationships with the formalisms we already studied. We will be sure to complete these revisions before the camera-ready submission if our paper is accepted.

---

### Meta-Review · Area_Chair_N3o6 · 2023-12-09

**Metareview:**

This paper compares the relative expressivity of different task specification mechanisms for RL agents. It compares the relative expressivity of Markov rewards, limit average rewards, reward machines, linear temporal logic, regularized RL, and outer non-linear Markov rewards, and prove that each of these formalisms have a non-zero intersection, but also a non-zero exclusive component. It also defines 17 different task specification formalisms, provides a complete ordering over all possible policies of an agent in tasks specification mechanism available for RL agents.

Paper is technically sound, with significant contribution to the RL community on reward formalism and task specification design mechanisms. Paper is clearly written and well-explained, with most technical proofs provided in supplementary materials.

**Justification For Why Not Higher Score:**

Reviewers expressed skepticism of technical details such as choice of stationary Markovian policies (over the more general history dependent policy class), and specifications of policy ordering, and questioned about the usefulness of policy ordering in applications. These questions are still remained unaddressed.

**Justification For Why Not Lower Score:**

This paper contributes a comprehensive set of proofs designed to show which reward formalisms in RL are distinctive. Theoretical contribution is novel and is likely to be useful to researchers who are deeply steeped in RL research.

---

### Decision · Program_Chairs · 2024-01-16

Accept (poster)